**Subject Category:**
Biology (whole organism)

biomaterials/evolution/behaviour

Chelonia, *Trachemys scripta*, *Pseudemys concinna*, nanostructure, pigments, chromatophores

**Author for correspondence:**
Jindřich Brejcha
e-mail: brejcha@natur.cuni.cz

# Body coloration and mechanisms of colour production in Archelosauria: the case of deirocheline turtles

Jindřich Brejcha[1,2,3], José Vicente Bataller[4],
Zuzana Bosáková[5], Jan Geryk[6], Martina Havlíková[5],
Karel Kleisner[1], Petr Maršík[7] and Enrique Font[8]

[1]Department of Philosophy and History of Science, Faculty of Science, Charles University, Viničná 7, Prague 2, 128 00, Czech Republic
[2]Department of Zoology, Natural History Museum, National Museum, Václavské nám. 68, Prague 1, 110 00, Czech Republic
[3]Department of Biophysical Chemistry, J. Heyrovský Institute of Physical Chemistry of the Czech Academy of Sciences, Dolejškova 3, Prague 8, 18223, Czech Republic
[4]Centro de Conservación de Especies Dulceacuícolas de la Comunidad Valenciana. VAERSA-Generalitat Valenciana, El Palmar, València, 46012, Spain
[5]Department of Analytical Chemistry, Faculty of Science, Charles University, Hlavova 8, Prague 2, 128 43, Czech Republic
[6]Department of Biology and Medical Genetics, 2nd Faculty of Medicine, Charles University and University Hospital Motol, V Úvalu 84, 150 06 Prague, Czech Republic
[7]Department of Food Science, Faculty of Agrobiology, Food, and Natural Resources, Czech University of Life Sciences Prague, Kamýcká 129, Prague 6, 165 00, Czech Republic
[8]Ethology Lab, Cavanilles Institute of Biodiversity and Evolutionary Biology, University of Valencia, C/ Catedrátic José Beltrán Martinez 2, Paterna, València, 46980, Spain

JB, 0000-0002-2164-6375

Animal body coloration is a complex trait resulting from the interplay of multiple mechanisms. While many studies address the functions of animal coloration, the mechanisms of colour production still remain unknown in most taxa. Here we compare reflectance spectra, cellular, ultra- and nano-structure of colour-producing elements, and pigment types in two freshwater turtles with contrasting courtship behaviour, *Trachemys scripta* and *Pseudemys concinna*. The two species differ in the distribution of pigment cell-types and in pigment diversity. We found xanthophores, melanocytes, abundant iridophores and dermal collagen fibres in stripes of both species. The yellow chin and forelimb stripes of both *P. concinna* and *T. scripta* contain xanthophores and iridophores, but the post-orbital regions of the two species

differ in cell-type distribution. The yellow post-orbital region of *P. concinna* contains both xanthophores and iridophores, while *T. scripta* has only xanthophores in the yellow-red postorbital/zygomatic regions. Moreover, in both species, the xanthophores colouring the yellow-red skin contain carotenoids, pterins and riboflavin, but *T. scripta* has a higher diversity of pigments than *P. concinna*. *Trachemys s. elegans* is sexually dichromatic. Differences in the distribution of pigment cell types across body regions in the two species may be related to visual signalling but do not match predictions based on courtship position. Our results demonstrate that archelosaurs share some colour production mechanisms with amphibians and lepidosaurs (i.e. vertical layering/stacking of different pigment cell types and interplay of carotenoids and pterins), but also employ novel mechanisms (i.e. nano-organization of dermal collagen) shared with mammals.

## 1. Introduction

Identifying the evolutionary origins and the selective processes responsible for the maintenance of complex morphological traits remains a persistent challenge in biology. Animal coloration is a complex morphological trait resulting from the interaction of multiple elements that evolve for various roles and only later are co-opted to produce colour [1,2]. Variation in body coloration during evolution is the result of two processes [3]: the integration of colour-producing elements to produce a given colour [4], and the independent development and evolution of those same colour-producing elements [5]. The intricate relationships between colour-producing elements and their functional roles stress the need to study the proximate causes of colour production [6]. While there has been progress in our understanding of the functions of animal coloration [7,8], our knowledge of colour production mechanisms in most groups of animals is still scarce.

Animal colours are produced by ultrastructural elements of the integument interacting with incident light, by light-absorbing pigments, or by a combination of both. In vertebrates, the majority of compounds responsible for the coloration of the integument are found in pigment cells which are derived from the neural crest [9–11]. Pigment cells are classified into five types: reflecting dermal pigment cells (iridophores, leucophores), absorbing dermal pigment cells (xanthophores, erythrophores, cyanophores), melanocytes with non-motile organelles (dermal melanocytes), melanocytes with motile organelles (dermal melanophores), and organelle transferring melanocytes (interfollicular and follicular epidermal melanocytes) [12].

Iridophores contain reflecting platelets of mainly guanine, while absorbing dermal pigment cells contain pterins in pterinosomes, or carotenoids dissolved in lipids in carotenoid vesicles. Melanocytes and melanophores contain melanins in melanosomes. Single cells with characteristics of two or more pigment cell types, e.g. xanthophore and iridophore, are occasionally found and are known as mosaic chromatophores [13,14]. Often, pigment cells of the same type are organized in more or less continuous layers that are roughly parallel to the skin surface. The vertical layering and relative thickness of the different pigment cells, thickness, orientation, number, and geometry of reflecting elements, as well as characteristics of the pigments within the cells determine the final colour of the skin [8,15,16]. Ectodermal keratinocytes (epidermis) and mesenchymal fibroblasts (dermis) also participate in colour production via reflection of light on the intra/extracellular protein matrix these cells produce [17–21].

Turtles are an early-diverging clade of Archelosauria, the evolutionary lineage of tetrapods leading to crocodiles and birds [22]. Although many turtles have a uniform dull colour, conspicuous striped and spotted patterns are common in all major lineages (for a comprehensive collection of photographs see [23–26]). These conspicuous colour patterns may be present in the hard-horny skin of shells, and/or in the soft skin of the head, limbs or tail. The dark areas of the skin of turtles may have a threefold origin consisting either of dermal, epidermal, or both epidermal and dermal melanocytes. Colourful bright regions are thought to be the result of the presence of xanthophores in the dermis [27] and their interplay with dermal melanophores [28]. Iridophores have not been shown to play a role in the coloration of turtles [27,29].

Pigment-bearing xanthophores were first described in the dermis of the Chinese softshell turtle (*Pelodiscus sinensis*) [29]. Xanthophores have also been found sporadically in the dermis of the spiny softshell turtle *Apalone spinifera*, the Murray river turtle (*Emydura macquarii*) and in the painted turtle (*Chrysemys picta*) [27]. Such scarcity of carotenoid/pterin-containing cells is in contrast with chemical analyses of the yellow and red regions of the integument in *C. picta* and in the red-eared slider

(*Trachemys scripta elegans*) [30,31]. Two major classes of carotenoids have been described in the integument of these turtles: short wavelength-absorbing apocarotenoids and longer wavelength-absorbing ketocarotenoids [30]. In addition to carotenoids, Steffen *et al*. [30] speculated that small quantities of some pterin could play a minor role in colour production in the integument of turtles. In support of this hypothesis, Alibardi [27] observed pterinosomes with characteristic concentric lamellae in the xanthophores of turtles. In the skin of the Japanese pond turtle (*Mauremys japonica*), Odate *et al*. [32] found two colourless pterins (biopterin and pterin-6-carboxylic acid) but did not find pterin, the precursor of yellowish xanthopterin, or isoxanthopterin. Thus, to this day there is no conclusive evidence of pterins playing a role in colour production in the skin of turtles, or in the skin of any other archelosaur.

Deirochelyinae are freshwater turtles of the family Emydidae, all of which, except one species (the diamond-back terrapin, *Malaclemys terrapin*), display a pattern of conspicuous stripes on the skin of the head and legs [33,34]. This striped pattern usually consists of alternating yellow and black regions, but red regions are also found in some species. The coloration of deirocheline turtles is thought to play a role in mate choice [30,35]. Some species are sexually dichromatic, with males often displaying the brightest, most conspicuous colours (reviewed in [19]); however sexual dichromatism is limited to particular regions of the body [36,37]. It has been suggested that some of the colourful regions found particularly in males may potentially be involved in intersexual communication and mate choice [36,38]. Colour ornaments are designed to maximize their conspicuousness only in certain, ethologically relevant contexts [39], e.g. when exposed to rivals during contests or to mates during courtship [40,41]. Courtship in freshwater turtles takes place underwater and males adopt characteristic positions relative to the female [42]. Deirocheline turtles are characterized by elaborate courtship behaviour including a complex forelimb display known as titillation [42]. Males in most species of deirocheline turtles perform the titillation display facing the female while swimming backwards in front of her, close to the water surface; however, males of the genus *Pseudemys* titillate while swimming above and parallel to the female, facing the same direction as her [42–47]. Thus, depending on the position adopted during courtship (face-to-face versus swim above) males of different species expose different body areas to females. By comparing different deirocheline species, it may be possible to examine whether there is a relationship between colour and colour-producing mechanisms and courtship position in these turtles.

In this study we compare the structure and pigment types of striped skin in two species of deirocheline freshwater turtles, the pond slider (*Trachemys scripta*) and the river cooter (*Pseudemys concinna*), with contrasting courtship behaviour. Males of *T. scripta* court females in a face-to-face position while males of *P. concinna* court females in a swim-above position. To determine if colour production mechanisms differ depending on the species-specific male position during courtship, we describe the cellular and ultrastructural make-up of those body regions that likely play a role during courtship behaviour (head, limbs) and the chemical nature of pigments involved in colour production. We combine multiple approaches to study the mechanisms of colour production: (i) reflectance spectrophotometry to objectively determine the colour of the turtles' body surfaces, (ii) transmission electron microscopy and image analyses to study the structure of the skin and the ultrastructure of colour-producing elements, and (iii) liquid chromatography coupled with mass spectrometry to determine the contribution of carotenoids/pterins to the turtles' coloration. We predict that regions of the skin that males expose during courtship to females should be more conspicuous and richer in pigment types than regions not exposed during courtship. Specifically, in *P. concinna* we expect the yellow chin to be sexually dichromatic, conspicuous and rich in pigment types compared to other skin surfaces. In *T. scripta*, on the other hand, we expect all regions exposed during courtship, i.e. dorsal and ventral head and limbs, to be sexually dichromatic, conspicuous and rich in pigment types.

# 2. Material and methods

## 2.1. Animals and handling

Two species of freshwater turtles, *Trachemys scripta* and *Pseudemys concinna*, were collected for purposes of control of invasive freshwater turtles as a follow-up to project 'LIFE-Trachemys' (LIFE09 NAT/ ES000529) to the Servicio de Vida Silvestre, D.G. Medi Natural (Generalitat Valenciana). Turtles (*P. concinna*: 4 males, 8 females; *T. s. elegans*: 42 males, 39 females; *T. s. scripta*: 1 male, 2 females) were captured during 2014–2017 with floating or hoop traps in coastal marshes or artificial ponds at several

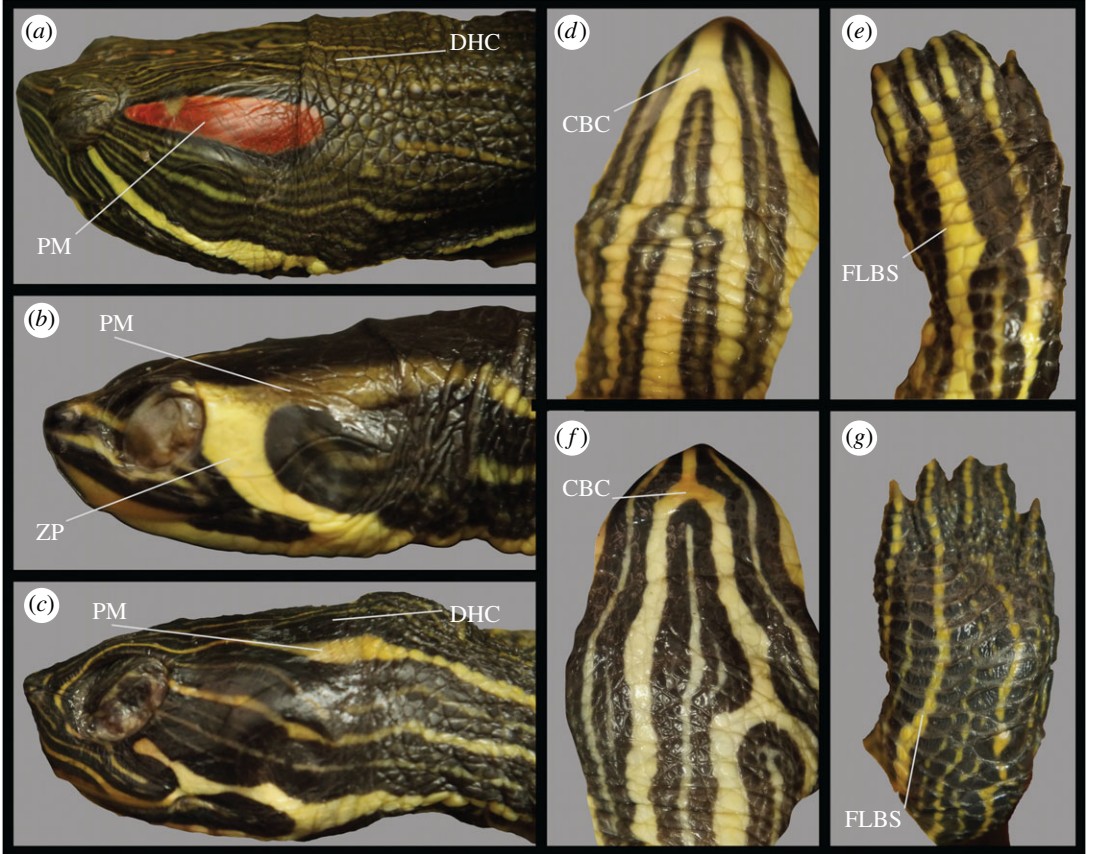

**Figure 1.** Photographs of skin regions examined in this study. (*a*) Lateral view of the head of *Trachemys scripta elegans*; (*b*) lateral view of the head of female of *Trachemys scripta scripta*; (*c*) lateral view of head of female *Pseudemys concinna*; (*d*) ventral side of head of female of *T. scripta*; (*e*) front limb of *T. scripta*; (*f*) ventral side of *P. concinna*; (*g*) front limb of *P. concinna*. CBC: main median chin yellow stripe; DHC: dorsal head background coloration; FLBS: main bright stripe of the fore limb; PM: postorbital marking; ZP: yellow zygomatic patch.

localities in the Comunidad Valenciana (Spain). Turtles were euthanized with an overdose of sodium pentobarbital (Eutanax, Fatro Ibérica, S.L) by authorized personnel. As all applicable international, national and/or institutional guidelines for the care and use of animals were followed (see ethical statement) no other permissions were required prior to conducting this research.

## 2.2. Spectral reflectance measurements

*Pseudemys concinna* and the yellow-bellied slider (*T. s. scripta*) have only yellow stripes on the head, while *T. s. elegans* also has red markings in the postorbital region. To quantify colour, we measured the spectral reflectance of turtle skin between 300 and 700 nm using a spectrophotometer. Reflectance spectra were taken before euthanasia with an Ocean Optics Jaz System (QR400-7-SR-BX: 400 µm reflection probe attached to JAZ-UV-VIS: Deuterium-Tungsten light source module) and a SONY Vaio computer running SpectraSuite software (Ocean Optics, Dunedin, Florida). The internal trigger was set to 30, integration time to 30, scans average to 30 and boxcar width to 10. The system was calibrated using a Spectralon WS-1 white diffuse reflectance standard, also from Ocean Optics. Reflectance spectra were taken in a darkened room using a hand-held probe oriented perpendicular to the skin surface (i.e. coincident normal measuring geometry [48]). An entomological pin (with acrylic head down) attached to the side of the probe allowed us to maintain a fixed 5 mm distance between the probe and the skin surface. Turtles were kept alive in clean water inside black plastic tanks and the skin surface was dried with a piece of cloth just before taking reflectance measurements. The main median chin yellow stripe (CBC), dorsal head background coloration (DHC), main bright/yellow forelimb stripe (FLBS) and the postorbital marking on the left side (PM) were measured in all specimens of *P. concinna* and *T. s. elegans* (figure 1). Unfortunately, as *T. s. scripta* is rarely found in the Valencia region, reflectance spectra could be obtained from only one specimen (a male). The yellow zygomatic patch (ZP) instead

of the PM was measured in *T. s. scripta* (figure 1*b*). Adult males of *T. s. elegans* become melanistic as they get older [49]; therefore, we have not included in our analyses reflectance spectra of 12 males determined as melanistic.

## 2.3. Processing and statistical analyses of reflectance spectra

Reflectance spectra binned by 0.37 nm were restricted to the range between 300 and 700 nm. Calculations were performed in R 3.3.2 [50] using the package PAVO [51]. Spectra were processed by smoothing (span = 0.3) and negative values were fixed to be zero. Processed reflectance spectra from each region of each individual of each taxon were summarized by the following variables: luminance or brightness (B1: sum of the relative reflectance over the entire spectral range), hue (H1: wavelength of maximum reflectance), and chroma (relative contribution of a spectral range to total brightness). We calculated five measures of chroma based on the wavelength ranges normally associated with specific colour hues: 300–400 nm (S1.UV), 400–510 nm (S1.blue), 510–605 nm (S1.green), 550–625 nm (S1.yellow), 605–700 nm (S1.red) [51–53]. Reflectance spectra of *T. s. scripta* were not included in the statistical analyses due to the small sample size available for this lineage. To analyse the contribution of summary variables to differences among species as well as sexes, we performed redundancy analyses (RDA) using the package vegan in R [54]. To overcome difficulties with different scales and variances of summary variables we standardized the data by scaling to zero mean and unit variance [55]. Species identity was the first constraining variable in the RDA to compare *P. concinna* and *T. s. elegans*. While testing for differences among species we also controlled for sex in the RDA as a second constraining variable to test for sexual dichromatism. Significance of results was assessed by ANOVA-like permutation tests (9999 permutations) for redundancy analysis (anova.cca) using the vegan package. Partition of variation with respect to explanatory variables (species identity and sex) was assessed using the varpart function also from the vegan package. We also analysed species separately; for this we performed RDA for each species, tested for its significance by anova.cca, and compared means of summary variables between the sexes by ANOVA with Holm's correction for multiple comparisons (see R script of all the analyses at [56]).

## 2.4. Microscopy

To investigate the cellular composition of the turtles' integument we took samples from CBC, FLBS and PM regions of *P. concinna* (females = 2) and CBC, DHC, FLBS and PM regions of *T. s. elegans* (normal males = 3, melanistic males = 2, females = 3). We also took samples of the CBC, PM and ZP of *T. s. scripta* (female = 2). Pieces of integument measuring *ca* 1 cm$^2$ from target regions were excised immediately after sacrifice and placed in Karnovsky's fixative (2% paraformaldehyde, 2.5% glutaraldehyde in PB buffer), left overnight at room temperature, and stored in the refrigerator at 4°C for 7 days. Smaller pieces (*ca* 2 mm$^3$) were cut from the original pieces of tissue. These were washed with 0.1 M PB, postfixed with 2% osmium tetra oxide (in 0.1 M PB solution), dehydrated in an increasing ethanol series, washed with 2% uranyl acetate in 70% ethanol, washed in propylenoxide and transferred to resin (Durcupan, Sigma). Polymerized resin blocks were cut on a Leica UCT Ultracut ultramicrotome. Semi-thin sections were stained with toluidine blue and observed under bright field and phase contrast in a Nikon Eclipse E800 photomicroscope. Ultra-thin sections were stained with lead citrate and observed and photographed using a FEI Tecnai Spirit G2 TEM equipped with a Morada digital camera (Soft Imaging System, Münster, Germany) and image capture software (TIA 4.7 SP3, FEI Tecnai). Magnification ranged from 1250× to 43 000× depending on the structures observed; intensity of the electron beam was adjusted to be in the optimal range for different magnifications.

## 2.5. Reflecting platelets of iridophores

To infer the role of iridophores in colour production we analysed the geometric properties of intact reflecting platelets as well as of the empty holes that remain after the reflecting platelets dissolve due to the embedding process. For analyses we used multiple electromicrographs of 4500–9900× magnification: *P. concinna*: PM (*N* = 8), CBC (*N* = 8), FLBS (*N* = 8); *T. s. elegans*: PM (*N* = 19), CBC (*N* = 15), FLBS (*N* = 9). Unfortunately, measurements of iridophores from the FLBS of *P. concinna* are not included in the results because the platelets were not well preserved. The scale of each image in nm/pixel was set based on the number of pixels in the scale bar of the original electromicrographs.

The length of the minor axis (height) and of the major axis (width) of intact reflecting platelets was measured directly from electromicrographs using the digital callipers utility in ImageJ (1.52a; [57]). Empty holes were analysed automatically by greyscale thresholding and subsequent measurement tools in ImageJ [16,58]. Briefly, electromicrographs were optimized for contrast and ellipses were fitted inside the platelets' bodies. Geometric parameters such as length of major axis of ellipse, length of minor axis, and angle between major axis of the ellipse and $x$-axis of electromicrographs were calculated (see electromicrographs of reflecting platelets in [56]).

Based on the length of the minor axis of both intact platelets and empty holes we calculated putative reflective wavelengths of reflecting platelets following equation (2) in Morrison [59] as applied by Haisten et al. [60] with refractive index of reflecting platelets 1.83 [61]. However, this model of colour production applies only to reflecting platelets that produce colour through thin-film interference [62], which is characterized by reflecting platelets organized in planes perpendicular to the direction of propagation of incoming light (i.e. parallel to the skin surface). Disorganized reflecting platelets on the other hand act as broadband reflectors producing incoherent scattering and reflect light across the entire wavelength spectrum with little influence on the hue of the specific skin region [16]. Following Saenko et al. [16] we calculated the $A/y_0$ ratio to determine the relative amount of reflecting platelets parallel to the skin surface from the Gaussian curve fitted to the distribution of the density of different angles of the ellipses. $A$ is the amplitude of the Gaussian curve above the background of randomly oriented platelets, and $y_0$ is the background level of randomly oriented platelets, i.e. intersection of the Gaussian curve with $y$-axis. The lower the $A/y_0$ ratio the lower the portion of platelets parallel to the skin surface, with 0 corresponding to either no platelets parallel to the skin surface, or completely disorganized platelets [16]. High $A$ and low $y_0$ indicate predominant orientation of platelets parallel to the surface and low density of disordered platelets. Full width of half maximum (FWHM) of the peak of the Gaussian curve was computed to characterize the spread of the Gaussian curve fitted to the data, i.e. width of angular distribution, as in Saenko et al. [16].

## 2.6. Fourier analysis of spatial distribution of dermal collagen arrays

To determine the role of the abundant extracellular collagen fibres in colour production we performed two dimensional discrete Fourier analyses of the spatial distribution of dermal collagen arrays in the superficial layers of the dermis using a Fourier tool [63] in MATLAB R2018a [64] kindly provided by Dr Richard Prum (https://prumlab.yale.edu/research/fourier-tool-analysis-coherent-scattering-biological-nanostructures). Because the tool was originally developed for earlier versions of MATLAB, script syntax had to be updated to secure compatibility with the current version of MATLAB (see description of changes in [56]).

There are two main outputs resulting from two-dimensional discrete Fourier analysis of spatial distribution of dermal collagen fibres: the 2-D Fourier power spectrum, and the radial averages of the power spectra. The colour of each pixel in the 2-D Fourier power spectrum represent the magnitude (the Fourier power) of a given spatial frequency (values on the $x$- and $y$-axes) in a given direction in the image [63]. Thus, a narrow ring pattern in the 2-D Fourier power spectrum is observed when the periodicity of the collagen fibres varies in a narrow range of spatial frequencies in all directions from any point in the image [63]. On the other hand, if the 2-D Fourier power spectrum would show two dots symmetrically located above and below the centre of the plot in one straight line, the interpretation would be that there is periodicity predominantly in one direction from each point of the image only and that the collagen fibres have a laminar organization [63]. Completely disorganized collagen fibres would be represented by random noise in the 2-D Fourier power spectrum plot.

The radial averages of the power spectra are derived from 2-D Fourier power spectra and summarize the distribution of the Fourier power of different spatial frequencies, i.e. they are averages of values of the Fourier power for each concentric ring (the frequency bin) in the 2-D Fourier power spectra. As these radial averages are normalized to total Fourier power of each single 2-D Fourier power spectra, it is possible to calculate the average of each frequency bin from cumulative analyses of multiple electromicrographs and to plot the average periodicity of collagen fibres across different samples.

The Fourier analyses followed published procedures [18,63]. In each micrograph (9900× magnification) of cross-sections of the collagen fibre arrays we selected a standardized square portion of the array (800 pixels$^2$), and optimized and standardized contrast in Adobe Photoshop (CS3, [65]). The scale of the image in nm/pixel was set based on the number of pixels in the scale bar of the original electromicrographs. The darker pixels of collagen fibres and lighter pixels of mucopolysaccharide were assigned representative refractive indices of 1.42 and 1.35, respectively [66]. We used the Fourier tool with the 2-D fast Fourier transform (FFT2) algorithm [67], resulting in a 2-D

Fourier power spectrum. Cumulative analyses were performed using multiple images for each region examined (*P. concinna*: PM, $N = 8$; CBC, $N = 8$; FLBS, $N = 8$; *T. s. elegans*: PM, $N = 19$; CBC, $N = 9$; FLBS, $N = 9$; *T. s. scripta*: CBC, $N = 6$; ZP, $N = 11$) (see electromicrographs of collagen fibres at [56]) to obtain the average radial averages of the power spectra for each region. For results we pooled images of the CBC region of both *T. scripta* subspecies to increase the sample size for this skin region.

The predicted reflectivity curve derives from radial averages of the power spectra of each electromicrograph analysed. Predicted reflectivity of the region is calculated by each frequency bin separately. First the inverse value of radial average for each spatial frequency is multiplied by twice the average refractive index and expressed as wavelength (nm) [63,68]. Thus, lower spatial frequencies in the radial averages of the power spectra represent longer wavelengths of predicted reflectivity. The predicted reflectivity in visible spectra was calculated for 51 concentric bins of the 2-D Fourier power spectra (corresponding to 51 10 nm wide wavelength intervals from 300 to 800 nm) for each region as described in Prum & Torres [63]. To visualize the composite predicted shape of reflectivity from multiple measurements of collagen fibre arrays per region we fitted a smoothed curve (span = 0.3) to the data using the local fitting loess function in R.

## 2.7. Pigment analyses

Integument from *P. concinna* ($N = 3$), *T. s. elegans* ($N = 2$) and *T. s. scripta* ($N = 2$) was used for carotenoid analyses. Sampled regions were CBC, FLBS, PM and ZP. Small pieces of integument were removed using micro-scissors and tweezers, cleaned mechanically with scalpel and tweezers, washed briefly with distilled water to get rid of potential contamination from muscles or body fluids and frozen at −20°C. Samples were then transported on ice to Prague (Czech Republic) by plane, where they were stored frozen at −20°C until analyses, but never longer than one month. Samples were extracted with 0.5 ml ethyl acetate. The vials containing the extracts were stored for 3 days at room temperature in complete darkness. The extracts were then evaporated to dryness by stream of nitrogen at 27°C and stored at −18°C. Immediately prior to the analyses samples were diluted in 200 µl of ethyl acetate (EtOAc).

Standards of astaxanthin, canthaxanthin, lutein and zeaxanthin were purchased from Sigma Aldrich (Munich, Germany). Stock solutions of the external standards were prepared at a concentration of 0.1 mg ml$^{-1}$ by dissolving in EtOAc. The working solution of the mixture of all the studied carotenoids in EtOAc was prepared at concentrations of 50, 100, 200, 500 and 1000 ng ml$^{-1}$ from the stock solutions.

Carotenoids were determined using UPLC system Dionex Ultimate 3000 (Thermo Fischer, USA) coupled with photodiode array (PDA) detector followed by ultra-high resolution accurate mass (HRAM) Q-TOF mass spectrometer (IMPACT II, Bruker Daltonik, Germany). Carotenoid separation was performed on a Kinetex C18 RP column (2.6 µm, 150 × 2.1 mm; Phenomenex, USA) maintained at 35°C using acetonitrile (A), methanol/water 1:1 $v/v$ (B) and a mixture of tert-butyl methyl ether/acetonitrile/methanol—86:86:8 $v/v/v$ (C) as mobile phases for gradient elution (electronic supplementary material, table S1) with constant flow rate of 0.2 ml/min. Chromatograms were monitored at 445 and 472 nm. The identity of the particular carotenoids was confirmed by HRAM mass spectrometry. Ions were detected in positive mode with electrospray (ESI) as well as atmospheric pressure chemical ionization (APCI) (see details in electronic supplementary material, table S2). A chromatogram of the mixture of all carotenoid standards (1000 ng of each standard dissolved in 1 ml EtOAc) is shown as electronic supplementary material, figure S1a. Chromatograms of the individual carotenoid standards with their absorbance spectra are shown in electronic supplementary material, figure S1c–f.

For analyses of pterins we used integument from one individual of each of the examined taxa. Sampled regions were CBC, FLBS, PM and ZP. Samples of integument were removed, treated and transported as for carotenoid analyses. Samples were extracted with dimethyl sulfoxide (DMSO) following a previously published procedure [69].

Standards of 6-biopterin, D-neopterin, leucopterin, pterin, pterin-6-carboxylic acid (pterin-6-COOH), and riboflavin were purchased from Sigma Aldrich (Munich, Germany). Isoxanthopterin and xanthopterin were obtained from Fluka (Buchs, Switzerland). L-sepiapterin was purchased from the Cayman Chemical Company (Ann Arbor, MI, USA). Erythropterin was kindly provided by Dr Ron Rutowski. Drosopterin was prepared according to Ferré *et al.* [70]. Stock solutions of the external standards (except drosopterin) were prepared at a concentration of 0.1 mg ml$^{-1}$ by dissolving in DMSO. The working solution of their mixture in DMSO was prepared at a concentration of 0.01 mg ml$^{-1}$ from the stock solutions.

All chromatographic measurements were carried out in a HPLC system (Agilent series 1290 coupled with a Triple Quad 6460 tandem mass spectrometer, Agilent Technologies, Waldbronn, Germany). For data acquisition, the Mass Hunter Workstation software was used. A ZIC®-HILIC (4.6 mm × 150 mm, 3.5 μm) column, based on zwitterionic sulfobetaine groups, was chosen (Merck, Darmstadt, Germany). The chromatographic conditions were adapted from Kozlík et al. [71] and Andrade et al. [72]. Separation of all studied compounds (with the exception of drosopterin) was achieved through isocratic elution at a flow rate of 0.5 ml min$^{-1}$ with the mobile phase consisting of acetonitrile/5 mM ammonium acetate, pH 6.80 at a volume ratio of 85 : 15 (v/v). Since drosopterin exhibits higher polarity compared to other pterins, it was analysed using acetonitrile/10 mM ammonium acetate, pH 6.80 at a volume ratio of 55 : 45 (v/v) and flow rate of 0.8 ml min$^{-1}$. Tandem mass spectrometric (MS/MS) measurements were performed in the selected reaction monitoring mode (SRM) with positive ionization. SRM conditions used for MS/MS determination are listed in electronic supplementary material, table S3. Pterins eluted with the following retention times (min): L-sepiapterin $t = 7.00$, pterin $t = 9.20$, isoxanthopterin $t = 12.5$, 6-biopterin $t = 12.6$, xanthopterin $t = 16.4$, leucopterin $t = 18.4$, erythropterin $t = 25.0$, pterin-6-COOH $t = 28.0$ and D-neopterin $t = 28.8$. Riboflavin eluted with retention time $t = 5.9$ min., drosopterin eluted with retention time $t = 10.2$ min. A chromatogram of the mixture of all standards (except drosopterin) in DMSO at a concentration of 0.01 mg ml$^{-1}$, measured in the TIC (total ion current) mode is shown as electronic supplementary material, figure S2a; SRM chromatograms of the individual compounds are shown as electronic supplementary material, figure S2b,c.

# 3. Results

## 3.1. Spectral reflectance

Average standardized reflectance spectra of *Pseudemys concinna* and *Trachemys scripta* are shown in figure 2 (see reflectance data of each individual at [56]). The dorsal head background coloration (DHC) of both *T. scripta* and *P. concinna* has a flat reflectance spectrum and low overall reflectance. Moreover, the reflectance of the DHC is further reduced beyond 550 nm. The red postorbital marking (PM) of *T. s. elegans* also has a relatively flat reflectance spectrum, with low overall reflectance but, in contrast to the DHC, shows increased reflectance at wavelengths longer than 550 nm giving rise to a plateau between 600–700 nm. All reflectance spectra from yellow regions (i.e. main median chin stripe—CBC, and main forelimb stripe—FLBS, of both species, and the PM of *P. concinna* as well as the zygomatic patch—ZP of *T. s. scripta*) are similar in that they show two peaks, one minor peak in the UV part of the spectrum between 300 and 400 nm and a larger, primary peak between 500 and 600 nm. The width, shape and height of the primary peak differ slightly among the yellow regions.

Means of summary variables derived from reflectance spectra and their standard deviations for each region of each taxon are shown in table 1 (see summary variables of each individual at [56]). Summary variables derived from reflectance spectra of one individual *T. s. scripta* overlap with summary variables (mean ± standard deviation) of *T. s. elegans* but fall outside the values (mean ± standard deviation) of the summary variables for *P. concinna*. Therefore, the colour of the different regions varies more between species than between subspecies of the same species. For most of the regions *T. s. elegans* and *T. s. scripta* can be treated as a single taxon *T. scripta*.

The ordination plot resulting from redundancy analyses (RDA) based on summary variables of both *T. s elegans* and *P. concinna* is shown in figure 3a. The axis constrained by species identity (RDA1) explains 18% and the axis constrained by sex (RDA2) explains 4% of total variance. The first residual axis (PC1) explains 18% and the second residual axis (PC2) explains 13% of the total variance. The differences between *P. concinna* and *T. s. elegans* and between sexes are statistically significant (ANOVA-like permutation test; species identity—$F = 17.16$, $p < 0.001$; sex—$F = 4.66$, $p < 0.001$). Effect size decreased when the model included sex as a single explanatory variable (adj $R^2 = 0.038$) compared with a model with both species identity and sex as explanatory variables (adj $R^2 = 0.2$) or a model with species identity as a single explanatory variable (adj $R^2 = 0.163$). Thus, compared to species identity, the contribution of sex to colour variation among homologous body regions is relatively minor.

Interpretation of the RDA1 axis, and therefore of differences between both species, seems to be region specific. *Pseudemys concinna* shows increased brightness (B1) along the RDA1 axis in CBC and PM, but decreased brightness in FLBS compared to *T. s. elegans*. The yellow CBC and FLBS of *P. concinna* show

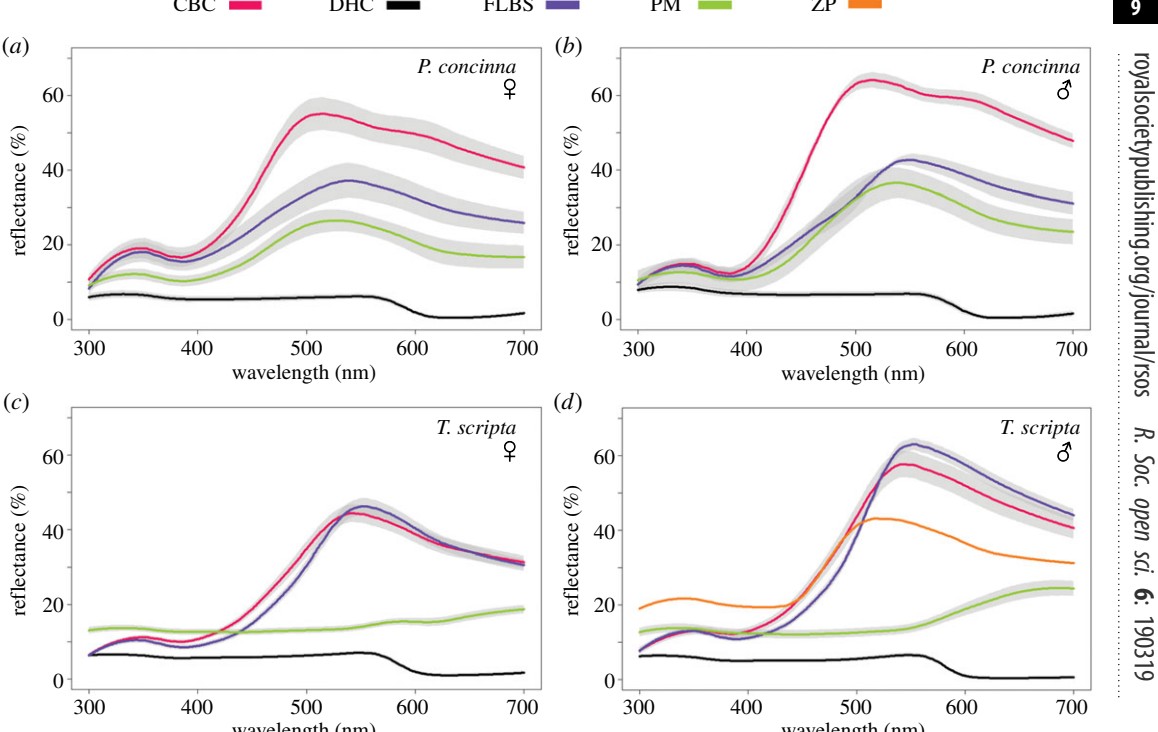

**Figure 2.** Mean (±s.e.m.) reflectance spectra of focal regions of (*a*) females of *Pseudemys concinna* (*N* = 8), (*b*) males of *P. concinna* (*N* = 4), (*c*) females of *Trachemys scripta elegans* (*N* = 39), and (*d*) males of *T. s. scripta elegans* (*N* = 30). CBC: main median chin yellow stripe; DHC: dorsal head background coloration; FLBS: main bright stripe of the fore limb; PM: postorbital marking. The yellow zygomatic patch (ZP) of *T. s. scripta* (*N* = 1) is shown in (*d*).

higher chroma (S1) in the blue and UV segments of reflectance spectra, but the yellow CBC and FLBS *T. s. elegans* have higher chroma (S1) in the yellow, red and green segments. The yellow PM of *P. concinna* shows higher chroma in the green, yellow and blue segments of reflectance spectra, while the red PM of *T. s. elegans* shows higher chroma in the red segment, but also in the UV segment of reflectance spectra (figure 3*b*). The increased chroma in the UV segment of the PM in *T. s. elegans* is a consequence of the overall low reflectance of this region rather than exceptionally high reflectance in the UV. Reported patterns of interspecific differences did not change when the PM region was not included in the analyses (data not shown here but see R script together with input data at [56]). Variables pertaining to background colour (DHC) are associated with PC1 rather than with RDA1 (compare values of RDA1 scores of the DHC to scores of summary variables of other regions—figure 3*b*) and contribute very little to colour differences between both species.

When species are treated separately, RDA shows no difference between sexes in summary variables of *P. concinna* (ANOVA-like permutation test; *F* = 1.97, *p* = 0.091). Also pairwise comparisons among summary variables of *P. concinna* show no significant sex differences (ANOVA with Holm's correction for multiple comparisons) (electronic supplementary material, table S4). On the other hand, differences in summary variables between sexes of *T. s. elegans* shown by RDA are significant (ANOVA-like permutation test; *F* = 3.90, *p* < 0.001). Axis constrained by sex of *T. s. elegans* explains 5.5% of total variance in colour of skin regions. The first residual axis explains 21% and the second residual axis explains 17% of the total variance. Pairwise comparisons (ANOVA with Holm's correction) show significant differences in B1CBC (*F* = 13.22, adj *p* = 0.014), B1FLBS (*F* = 12.98, adj *p* = 0.015), S1.blueFLBS (*F* = 10.89, adj *p* < 0.001) and S1.redFLBS (*F* = 17.22, adj *p* < 0.001) between males and females of *T. s. elegans* (see all *F* and *p* values in electronic supplementary material, table S4).

In short, the major differences in colour between *P. concinna* and *T. s. elegans* are: (i) shorter wavelengths contribute more to yellow colour of the CBC and FLBS in *P. concinna* compared to *T. s. elegans*; (ii) CBC and PM are brighter in *P. concinna*, but the FLBS is brighter in *T. s. elegans*; (iii) *P. concinna* shows little sexual dichromatism, but males of *T. s. elegans* have distinctly brighter CBC, and brighter FLBS with more blue and red chroma than females.

**Table 1.** Summary colour variables for each taxon involved in this study. For *P. concinna* and *T. scripta* summary variables are shown as means for both sexes. Asterisk denotes variables that differ significantly between the sexes. B1—sum of the relative reflectance over the entire spectral range; H1—wavelength of maximum reflectance; relative contributions of a spectral ranges to the total brightness: 300–400 nm (S1.UV), 400–510 nm (S1.blue), 510–605 nm (S1.green), 550–625 nm (S1.yellow), 605–700 nm (S1.red). CBC: main median chin yellow stripe; DHC: dorsal head background coloration; FLBS: main bright stripe of the fore limb; PM: postorbital marking; ZP: yellow zygomatic patch. Note that for *T. s. scripta* the values of summary variables resulted from measurements of only one male.

| taxon (number of individuals) | region | sex | variable (s.d.) | | | | | | |
| --- | --- | --- | --- | --- | --- | --- | --- | --- | --- |
| | | | B1 | S1.UV | S1.blue | S1.green | S1.yellow | S1.red | H1 |
| *P. concinna* | CBC | f | 15002.9 (3159.58) | 0.113 (0.036) | 0.273 (0.021) | 0.334 (0.024) | 0.255 (0.021) | 0.287 (0.027) | 514.75 (12.406) |
| (females, *N* = 8) | | m | 16744.947 (391.134) | 0.080 (0.017) | 0.271 (0.019) | 0.035 (0.017) | 0.27 (0.013) | 0.306 (0.019) | 513.25 (4.856) |
| (males, *N* = 4) | DHC | f | 1781.986 (752.728) | 0.347 (0.054) | 0.348 (0.013) | 0.28 (0.043) | 0.144 (0.028) | 0.032 (0.037) | 384.125 (101.226) |
| | | m | 2149.312 (651.104) | 0.376 (0.016) | 0.348 (0.017) | 0.0257 (0.012) | 0.135 (0.012) | 0.026 (0.035) | 327.25 (8.382) |
| | FLBS | f | 10394.211 (3696.107) | 0.152 (0.038) | 0.266 (0.02) | 0.328 (0.026) | 0.246 (0.022) | 0.262 (0.024) | 539.375 (10.954) |
| | | m | 10928.878 (1214.563) | 0.116 (0.024) | 0.234 (0.038) | 0.358 (0.026) | 0.278 (0.024) | 0.299 (0.32) | 541.75 (16.998) |
| | PM | f | 7067.994 (2582.48) | 0.16 (0.025) | 0.282 (0.025) | 0.338 (0.027) | 0.238 (0.016) | 0.228 (0.038) | 526.25 (12.702) |
| | | m | 9190.255 (2912.302) | 0.123 (0.025) | 0.250 (0.02) | 0.365 (0.029) | 0.268 (0.019) | 0.269 (0.013) | 536.75 (5.91) |
| *T. s. elegans* | CBC | f | 10705.771 (3632.844)* | 0.1 (0.04) | 0.225 (0.029) | 0.378 (0.041) | 0.287 (0.030) | 0.305 (0.03) | 550.073 (34.71) |
| (females, *N* = 39) | | m | 14258.388 (4376.958)* | 0.088 (0.025) | 0.214 (0.021) | 0.389 (0.025) | 0.299 (0.018) | 0.317 (0.027) | 544.556 (5.493) |
| (males, *N* = 30) | DHC | f | 1903.384 (1084.486) | 0.346 (0.046) | 0.356 (0.028) | 0.277 (0.023) | 0.137 (0.033) | 0.028 (0.061) | 419.609 (119.7) |
| | | m | 1623.636 (1044.952) | 0.363 (0.033) | 0.350 (0.025) | 0.285 (0.036) | 0.129 (0.027) | 0.008 (0.035) | 451.037 (119.264) |
| | FLBS | f | 11002.409 (3036.267)* | 0.09 (0.024) | 0.195 (0.024)* | 0.404 (0.025) | 0.313 (0.022) | 0.319 (0.025)* | 550.975 (7.837) |
| | | m | 13580.557 (2640.995)* | 0.08 (0.024) | 0.174 (0.026)* | 0.407 (0.024) | 0.324 (0.022) | 0.346 (0.029)* | 555.852 (12.63) |
| | PM | f | 5625.108 (2062.488) | 0.23 (0.035) | 0.242 (0.042) | 0.240 (0.024) | 0.203 (0.029) | 0.294 (0.084) | 602.488 (143.451) |
| | | m | 6548.252 (2844.586) | 0.21 (0.037) | 0.213 (0.047) | 0.233 (0.036) | 0.213 (0.035) | 0.35 (0.089) | 652.593 (77.594) |
| *T. s. scripta* | CBC | m | 11467.67 | 0.094 | 0.229 | 0.375 | 0.28 | 0.309 | 518 |
| (male, *N* = 1) | DHC | m | 801.896 | 0.29 | 0.37 | 0.346 | 0.159 | 0 | 549 |
| | FLBS | m | 10980.87 | 0.078 | 0.192 | 0.405 | 0.309 | 0.334 | 541 |
| | ZP | m | 12102.164 | 0.172 | 0.252 | 0.323 | 0.239 | 0.261 | 518 |

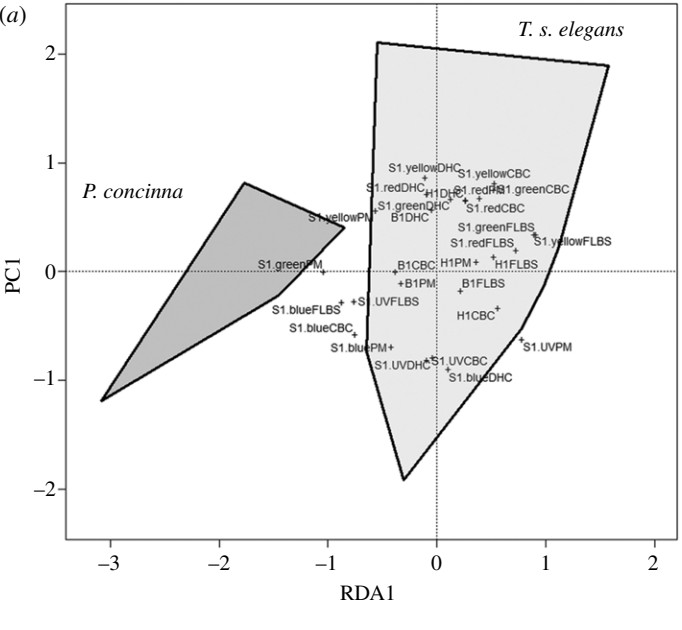

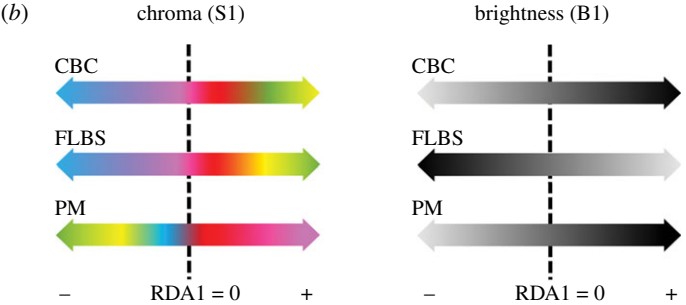

(c) scores of summary variables of colour regions on constrained axis (RDA1)

| CBC | | |
| --- | --- | --- |
| | region | RDA1 |
| s1.blueCBC | cbc | −0.75289 |
| B1CBC | cbc | −0.37918 |
| s1.UVCBC | cbc | −0.09259 |
| s1.redCBC | cbc | 0.26518 |
| s1.greenCBC | cbc | 0.524 |
| s1.yellowCBC | cbc | 0.52967 |
| H1CBC | cbc | 0.55835 |

| DHC | | |
| --- | --- | --- |
| | region | RDA1 |
| s1.yellowDHC | dhc | −0.1107 |
| s1.redDHC | dhc | −0.09387 |
| B1DHC | dhc | −0.05111 |
| S1.UVDHC | dhc | −0.04114 |
| S1.blueDHC | dhc | 0.10764 |
| S1.greenDHC | dhc | 0.12507 |
| H1DHC | dhc | 0.2659 |

| FLBS | | |
| --- | --- | --- |
| | region | RDA1 |
| s1.blueFLBS | flbs | −0.87266 |
| s1.UVFLBS | flbs | −0.76201 |
| B1FLBS | flbs | 0.21535 |
| H1FLBS | flbs | 0.51805 |
| S1.redFLBS | flbs | 0.72465 |
| S1.yellowFLBS | flbs | 0.89408 |
| S1.greenFLBS | flbs | 0.90804 |

| PM | | |
| --- | --- | --- |
| | region | RDA1 |
| S1.greenPM | pm | −1.04123 |
| S1.yellowPM | pm | −0.56383 |
| S1.bluePM | pm | −0.41629 |
| B1PM | pm | −0.32662 |
| H1PM | pm | 0.364 |
| S1.redPM | pm | 0.39256 |
| S1.UVPM | pm | 0.78247 |

**Figure 3.** Overall differences in colour between the two studied species. (*a*) Biplot representing results of RDA of summary variables derived from reflectance spectra of *Pseudemys concinna* and *Trachemys scripta elegans*. First axis (RDA1) is constrained by species (explains 17% of total variance). The first residual axis (PC1) explains 18% of total variance. B1: sum of the relative reflectance over the entire spectral range; H1: wavelength of maximum reflectance; relative contributions of a spectral ranges to the total brightness: 300–400 nm (S1.UV), 400–510 nm (S1.blue), 510–605 nm (S1.green), 550–625 nm (S1.yellow), 605–700 nm (S1.red). CBC: main median chin yellow stripe; DHC: dorsal head background coloration; FLBS: main bright stripe of the fore limb; PM: postorbital marking; ZP: yellow zygomatic patch. (*b*) Schema of the differences between species (distribution along RDA1 axis) in chroma (S1) and brightness (B1) of examined regions. Note that the DHC region aligned rather more with the first residual axis (PC1). (*c*) Table of scores of summary variables of all colour regions of both species on RDA1 axis. Thick line denotes zero.

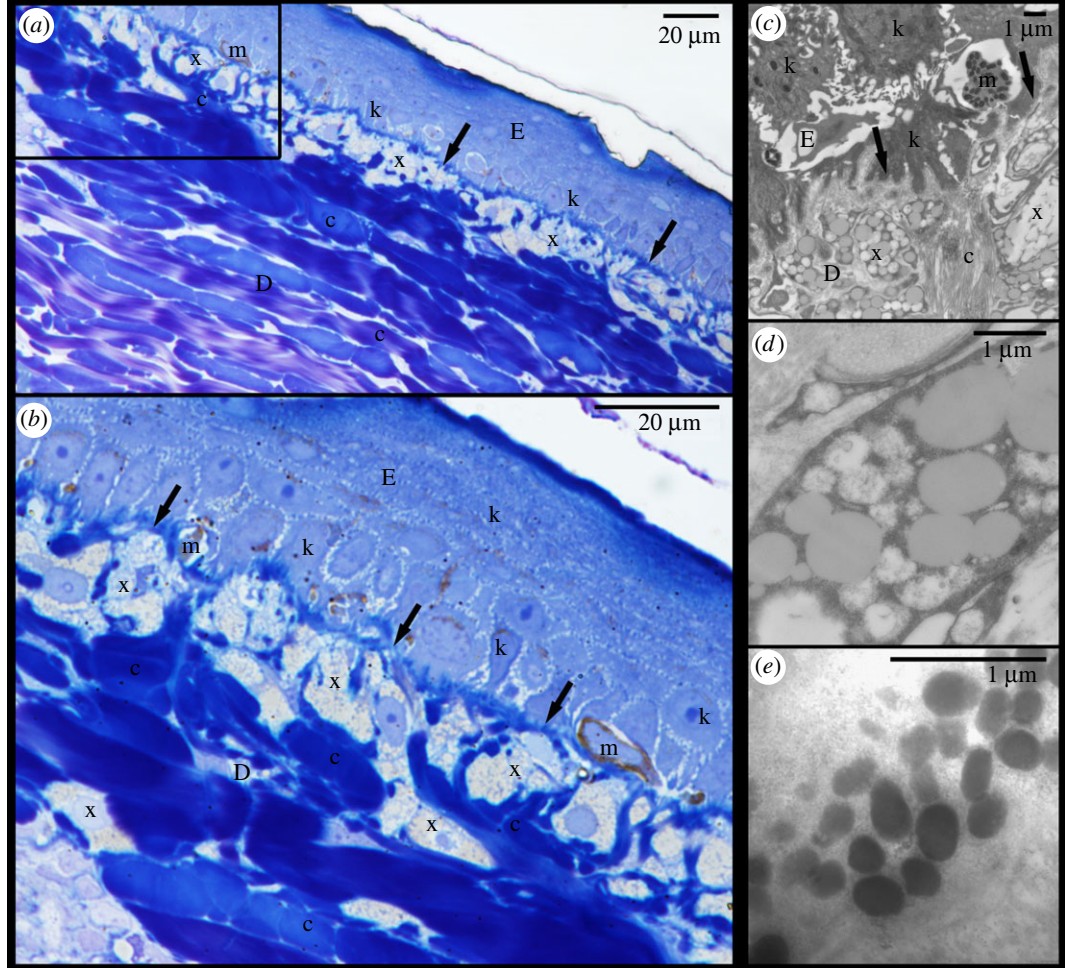

**Figure 4.** Red postorbital region (PM) of *Trachemys scripta elegans*. (*a*) Micrograph of semi-thin section stained with toluidine blue viewed under light microscope; (*b*) area denoted by rectangle in (*a*) viewed at higher magnification; (*c*) electromicrograph showing xanthophores residing in dermis under basal lamina (solid arrows); (*d*) closer view of xanthophore and its carotenoids vesicles; (*e*) detail of melanosomes of epidermal melanocyte. D: dermis; E: epidermis; c: collagen fibres; k: keratinocyte; m: melanocyte; x: xanthophore; arrows: basal lamina.

## 3.2. Microscopy

The epidermis is separated from the dermis by a conspicuous basal lamina (solid arrows in figures 4 and 5). The epidermis differs in thickness between regions and contains multiple layers of keratinocytes. The corneous layer of the epidermis is relatively thin in all regions except in the FLBS where it represents about one fifth of the thickness of the entire epidermis (of all the regions examined only the FLBS is covered by scales). Arrays of collagen fibres are highly abundant in the dermis of every region examined. Four basic pigment cells types are present in the integument of all studied turtles: epidermal melanocytes, and dermal iridophores, melanocytes, and xanthophores (figures 4, 5 and 6). We have not observed finger-like projections of dermal melanocytes that are characteristic of dermal melanophores with motile organelles [12]. Mosaic pigment cells were also found in the dermis of *T. s. elegans* (figure 6*d,g,h*). The pigment cell types present in each of the studied regions are summarized in table 2.

Epidermal melanocytes are present in the epidermis in the PM and DHC of *T. scripta* (figures 7*e* and 8*a–c*), and DHC of *P. concinna*. Epidermal melanocytes are located just above the basal lamina and are surrounded by keratinocytes. In samples with epidermal melanocytes the surrounding keratinocytes contain melanosomes. In some samples, however, the epidermal melanocytes are contracted and contain only a few melanosomes; those samples lack melanosomes deposited in keratinocytes. The PM of *T. s. elegans* contains mostly contracted epidermal melanocytes and almost no transferred melanosomes (figure 8*a*). In contrast, the PM of *T. s. scripta* contains enlarged epidermal melanocytes

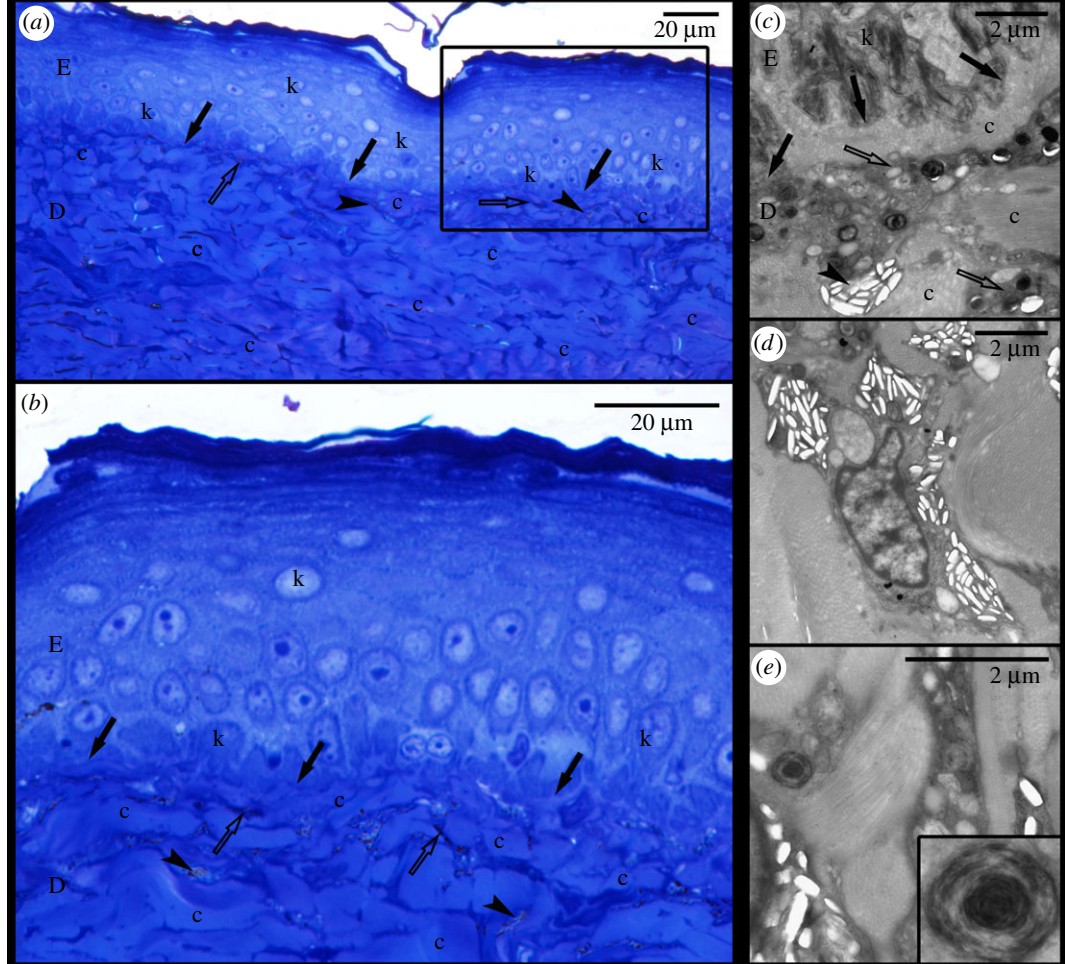

**Figure 5.** Yellow postorbital region (PM) of *Pseudemys concinna*. (*a*) Semi-thin section stained by toluidine blue viewed under light microscope; (*b*) area denoted by rectangle in (*a*) viewed at higher magnification; (*c*) Electromicrograph showing xanthophores and iridophores in the dermis; (*d*) closer view of iridophore; (*e*) closer look on xanthophore with detail of pterinosome in the inset. D: dermis; E: epidermis; c: collagen fibres; k: keratinocyte; solid arrows: basal lamina; clear arrows: xanthophore; arrowheads: iridophores.

and melanosomes distributed among many keratinocytes (figure 8*c*). *Pseudemys concinna* does not have epidermal melanocytes in the PM (figures 5 and 8*e,f*).

Iridophores are present in the dermis of the CBC, FLBS, and PM of *P. concinna*, and in the CBC, FLBS of *T. scripta*, but are missing in the PM and ZP of *T. scripta*. Iridophores are present also in the DHC, but these are very rare, small and seem haphazardly distributed (figure 7*e*). The iridophores of *P. concinna* do not form a continuous layer. They are rather scarce in the CBC and FLBS (figure 7*b,d*), and more abundant in the PM (figure 8*e*). In the CBC and FLBS of *T. scripta* iridophores are abundant and form an almost continuous layer several cells thick (figure 7*a,c*).

Dermal melanocytes are abundant in the DHC (figure 7*e,f*) and in the narrow black lines outlining the yellow-red regions (figure 6). In the DHC melanocytes are dispersed throughout the upper parts of dermis but do not generally form a continuous layer. In contrast, the dermal melanocytes in the narrow black lines form a continuous layer. Dermal melanocytes were present also in the PM of a melanistic male of *T. s. elegans* (figure 8*b*).

Xanthophores are present in the dermis of all regions examined just beneath the basal lamina and just above iridophores when those are present. In the CBC, FLBS (figure 7*b,d*) and PM (figure 8*e,f*) of *P. concinna* the xanthophores are sparse and do not form a continuous layer. The xanthophores form a thick continuous layer in the PM of *T. s. elegans* (figure 8*a*) and in the ZP of *T. s. scripta* (figure 8*d*). In the CBC and FLBS of *T. s. elegans* (figure 7*a,c*) and PM of *T. s. scripta* (figure 8*c*) the xanthophore layer is also continuous but its thickness is reduced compared to the PM of *T. s. elegans* or ZP of *T. s. scripta*. In the two melanistic males of *T. s. elegans* that we examined, the xanthophores of the PM

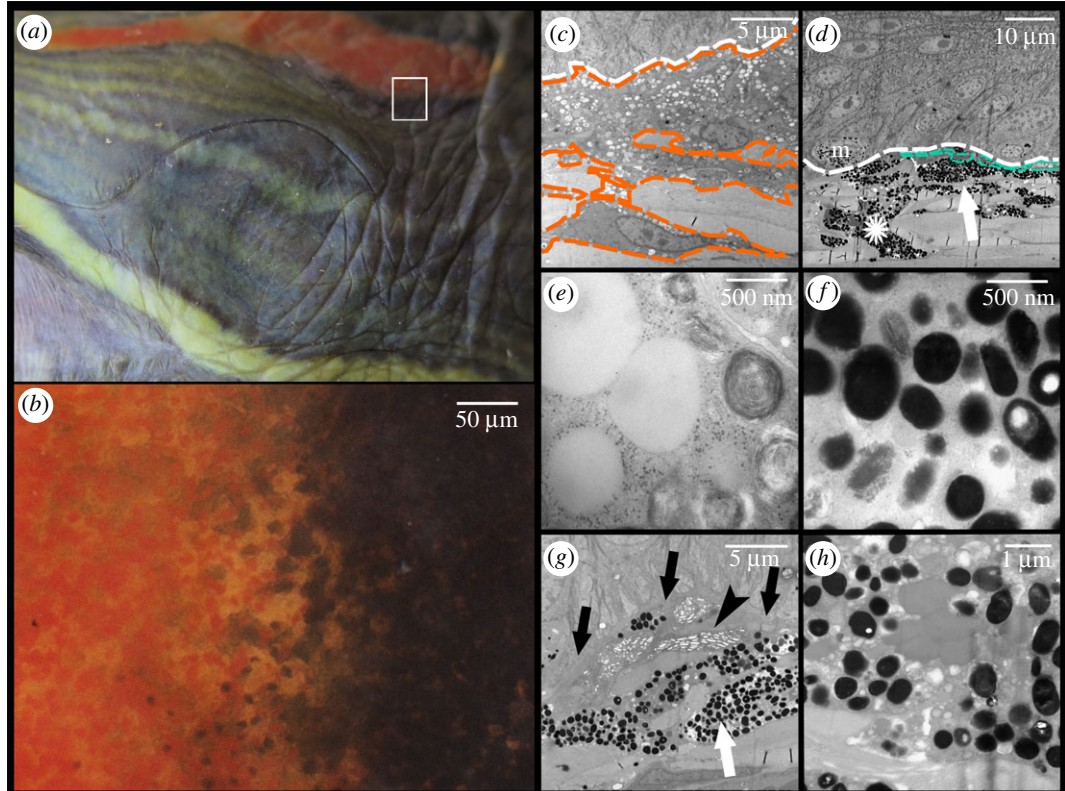

**Figure 6.** Boundary between the red postorbital region (PM) and its surrounding black line of *Trachemys scripta elegans*; (*a*) photography of the lateral view of the head, region of interest highlighted by white rectangle; (*b*) edge between two different regions viewed under binocular microscope; (*c*) electromicrograph of section of red postorbital region, orange line: xanthophores; white line: basal lamina; (*d*) edge of black region, turquoise line: iridophore; white line: basal lamina; m: epidermal melanocyte; white arrow: dermal melanocytes; asterisk: mosaic chromatophore; (*e*) detail of vesicles of xanthophores of the red postorbital region—carotenoid vesicles and pterinosomes; (*f*) detail of vesicles of dermal melanocytes of black region, melanosomes; (*g*) detail on small iridophore next to dermal melanocytes in the black region; arrowhead: iridophore, white arrow: dermal melanocytes, black arrows: basal lamina. White line: basal lamina; asterisk: mosaic chromatophore; (*h*) high magnification of vesicles of mosaic chromatophore, showing co-localization of carotenoid vesicles and melanosomes.

region are interspersed with dermal melanocytes and do not form a continuous layer (figure 8*b*). Xanthophores are also found in the DHC where they form aggregates of interconnected cells, but not a continuous layer (figure 7*e*). Xanthophores contain three types of pigment bearing organelles: two types of carotenoid vesicles with different electron density but without internal structure, and pterinosomes (figure 9). Unlike carotenoid vesicles, mature pterinosomes are characterized by the presence of concentric lamellae [73,74]. Mosaic pigment cells containing carotenoid vesicles, pterinosomes and melanosomes are present at the boundary between the PM and its black surrounding line in *T. s. elegans* (figure 6*g*,*h*).

*Pseudemys concinna* has the same pigment cell types in the ventral and dorsal sides of the head. In contrast, the dorsal and ventral regions of the head in the two subspecies of *T. scripta* differ in the distribution of colour-producing pigment cells (figure 10). The ventral CBC (figure 7*a*) and FLBS (figure 7*c*,*d*) of *P. concinna* and *T. scripta* contain iridophores together with xanthophores in the dermis. This organization is found also in the PM of *P. concinna* (figures 7*b* and 8*e*,*f*). However, the dorsal head regions of both subspecies of *T. scripta*, i.e. yellow YP of *T. s. scripta* (figure 8*d*), the dark PM of *T. s. scripta* (figure 8*c*), and the red PM of *T. s. elegans* (figure 8*a*) contain only xanthophores in the dermis. Another difference among the studied species is that, unlike *T. scripta*, the pigment cells of *P. concinna* do not form continuous layers.

Pigment cells form continuous layers whose composition differs between colour regions (figures 7 and 8). Bright colour regions contain both single cell type layers (xanthophores) and two cell type layers (xanthophores + iridophores) (figure 10). The dark stripes and the background coloration are characterized by the presence of all three cell types but with a predominance of dermal melanocytes. The dermal pigment cells of all studied turtles are embedded in abundant collagen fibres, which in

**Table 2.** Presence (✓ present/✗ absent) of pigment cell types in different regions of the integument in the studied taxa. CBC: main median chin yellow stripe; DHC: dorsal head background coloration; FLBS: main bright stripe of the fore limb; PM: postorbital marking; ZP: yellow zygomatic patch.

| | | pigment cell type | | | |
| --- | --- | --- | --- | --- | --- |
| | | epidermal | | dermal | |
| taxon | region | melanocytes | iridophores | melanocytes | xanthophores |
| P. concinna | CBC | ✗ | ✓ | ✗ | ✓ |
| | DHC | ✓ | ✓ | ✓ | ✓ |
| | FLBS | ✗ | ✓ | ✗ | ✓ |
| | PM | ✗ | ✓ | ✗ | ✓ |
| T. s. elegans | CBC | ✗ | ✓ | ✗ | ✓ |
| | DHC | ✓ | ✓ | ✓ | ✓ |
| | FLBS | ✗ | ✓ | ✗ | ✓ |
| | PM | ✓ | ✗ | ✗ | ✓ |
| T. s. scripta | CBC | ✗ | ✓ | ✗ | ✓ |
| | DHC | ✓ | ✓ | ✓ | ✓ |
| | FLBS | ✗ | ✓ | ✗ | ✓ |
| | ZP | ✗ | ✗ | ✗ | ✓ |

*P. concinna* make up most of the mass of the dermis adjacent to the epidermis (compare figure 4*a*,*b* and figure 5*a*,*b*).

## 3.3. Reflecting platelets of iridophores

The reflecting platelets present in the iridophores of all regions examined are roughly rectangular (figure 11). Characteristics and counts of intact reflecting platelets as well as empty holes are summarized in table 2. Only in the FLBS of *T. s. elegans* and the CBC of *P. concinna* reflecting platelets are mostly parallel to the skin surface (predominant angles of orientation are −1.98° and 2.49°, respectively), while the major axes of reflecting platelets in the CBC of *T. s. elegans* and in the PM of *P. concinna* have average angles differing from zero (30.86° and 68.3°, respectively). Moreover, the reflecting platelets of the FLBS of *T. s. elegans* are more aligned with the skin surface and regularly organized ($A/y_0 = 607.28$, FWHM = 59.19°) than other regions examined (CBC of *P. concinna*—$A/y_0 =$ 10.85, FWHM = 95.3°; CBC of *T. s. elegans*—$A/y_0 = 6.31$, FWHM = 106.24°; PM of *P. concinna*—$A/y_0 =$ 4.91, FWHM = 112.42°) (see electronic supplementary material, figure S3a–d for distributions of angles of reflecting platelets for different regions; see measurements of reflecting platelets in [56]).

Only the FLBS of *T. s. elegans* meets the full conditions of the thin layer interference model. Figure 12*a*,*b* shows the predicted reflectivity of iridophores of the FLBS of *T. s. elegans*. The peak of the highest density at the predicted reflected wavelength differs slightly depending on whether intact reflecting platelets or empty holes are measured (see electronic supplementary material, figure S3e–h for distributions of thickness of reflecting platelets for different regions based on the two methods of measurement): 618 nm based on measurements of intact platelets and 637 nm based on measurements of empty holes.

Due to the organization of their platelets, the iridophores in the CBC of *P. concinna* probably function as broadband reflectors. The same probably applies to the iridophores in the CBC of *T. s. elegans* and the PM of *P. concinna*, which in addition to disorganized reflecting platelets also possess platelets of variable size (figure 12*c*). Only the iridophores of the FLBS of *T. s. elegans* seem capable of intense narrowband colour production.

## 3.4. Fourier analysis of spatial distribution of dermal collagen arrays

The 2-D Fourier power spectra show that none of the single collagen fibre arrays examined (see examples of electromicrographs analysed in figure 13; see output files of Fourier analyses for each electromicrograph analysed in [56]) are organized randomly with respect to wavelengths of light. In

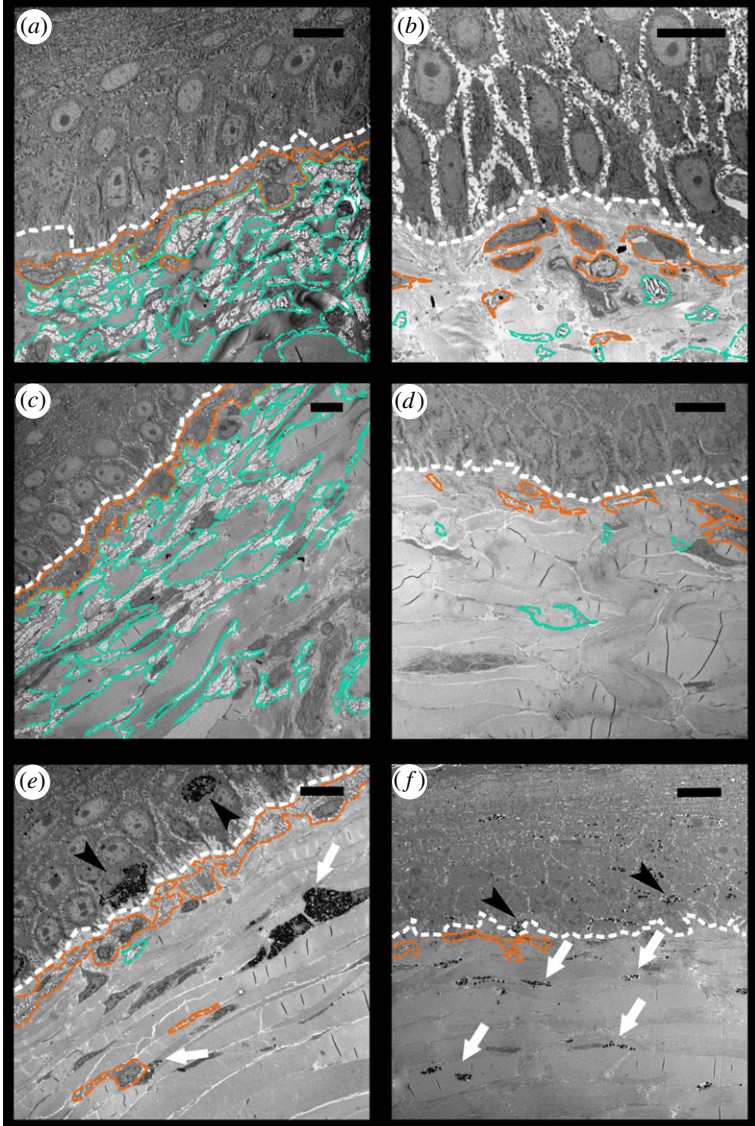

**Figure 7.** Electromicrographs of different regions of turtles' integument. (*a*) Yellow chin (CBC) region of *Trachemys scripta*; (*b*) CBC region of *Pseudemys concinna*; (*c*) yellow front limb (FLBS) region of *T. scripta*; (*d*) FLBS region of *P. concinna*; (*e*) Dorsal dark head (DHC) region of *T. s. elegans*; (*f*) DHC region of *T. s. scripta*. White line: basal lamina; orange line: xanthophores; turquoise line: iridophores; black arrowheads: epidermal melanocytes; white arrows: dermal melanocytes. Scale bar represents 10 μm. Note that magnification differs.

other words, the distances between collagen fibres are of the same order of magnitude as wavelengths of light [63]. However, the collagen arrays of *T. s. elegans* show variation in the spatial frequencies of the 2-D Fourier power spectra within regions (electronic supplementary material, figure S4*a–c*). In contrast, the power spectra of collagen arrays of *P. concinna* show reduced variation in organization within regions (electronic supplementary material, figure S4*d–f*). Collagen arrays of the PM of *P. concinna* are unambiguously organized in a way which results in a small disc pattern of 2-D Fourier power spectra at low spatial frequencies across all micrographs analysed. Collagen arrays of the FLBS of *P. concinna* show a two-sided symmetric pattern of 2-D Fourier power spectra at intermediate spatial frequencies rather than a disc pattern, suggesting predominant periodicity in one direction perpendicular to the fibres. In the CBC of *P. concinna* power spectra show a ring pattern at intermediate spatial frequencies, pointing to equivalent nanostructure in all directions perpendicular to the fibres.

Normalized average radial means of the Fourier power spectra for each region of *P. concinna* and *T. s. elegans* are shown in figure 14. Radial means of the Fourier power spectra of all examined regions of both species show increased power values in the lowest spatial frequencies (less than $0.0034\,\mathrm{nm}^{-1}$) compared to spatial frequencies relevant to coherent scattering in the visible light (figure 14*a–f*;

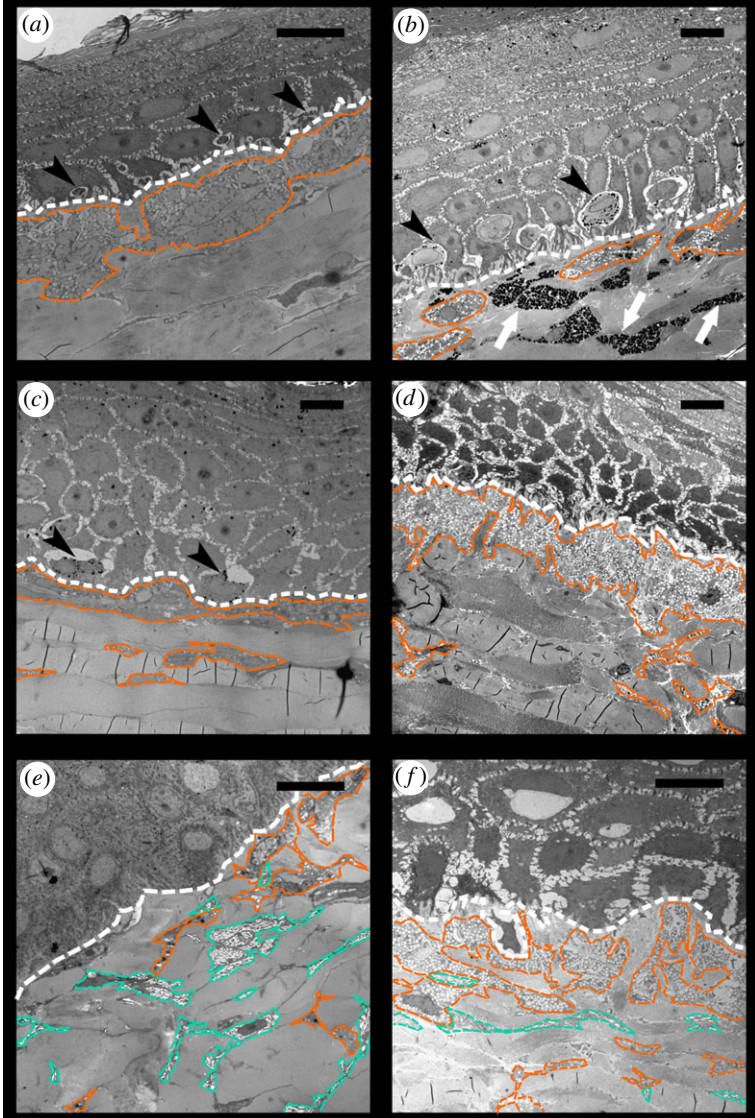

**Figure 8.** Electromicrographs of different regions of turtles' integument. (*a*) Red postorbital (PM) region of *Trachemys scripta elegans*, note small epidermal melanocytes and no melanosomes in keratinocytes; (*b*) red PM region of melanistic male of *T. s. elegans*, note enlarged epidermal melanocytes, but few melanosomes in keratinocytes; (*c*) dark PM region of *T. s. scripta*, note enlarged epidermal melanocytes and abundant melanosomes in keratinocytes; (*d*) yellow zygomatic patch (ZP) region of *T. s. scripta*, note that there are no epidermal melanocytes; (*e*) yellow PM region of *P. concinna*; (*f*) yellow PM region of different specimen of *P. concinna* than in (*e*), see the difference in abundance of xanthophores and iridophores, here xanthophores form almost continuous layer, it is possible that these differences are due to the different localization within the PM stripe on head, or these differences may be due to intraspecific variability. White line: basal lamina; orange line: xanthophores; turquoise line: iridophores; black arrowheads: epidermal melanocytes; white arrows: dermal melanocytes. Scale bar represents 10 μm. Note that magnification differs.

electronic supplementary material, figure S5a). These values of spatial frequencies indicate that the largest portion of light reflected by collagen fibre arrays in these regions belongs to the infrared part of the light spectrum. However, for the CBC of *P. concinna* this peak of low spatial frequencies lies immediately next to the segment of spatial frequencies relevant to coherent scattering in visible light (from 0.0034 to 0.0092 nm$^{-1}$) (figure 14*e*). Radial means of spatial frequencies of collagen fibres of the CBC of *P. concinna* relevant to coherent scattering in visible light are somewhat elevated compared to the radial means of all regions of *T. scripta* and the PM of *P. concinna* (figure 14*d*). In the FLBS of *P. concinna* collagen fibres are organized at appropriate spatial frequencies to produce coherent scattering in the visible spectrum (figure 14*f*).

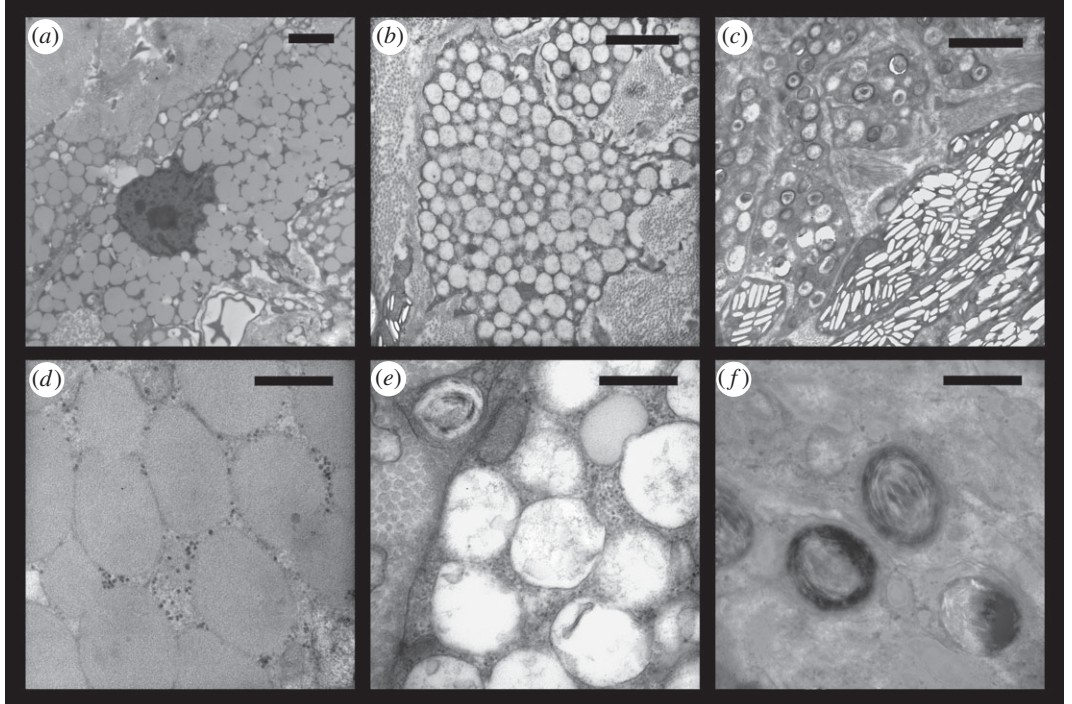

**Figure 9.** Examples of three different types of vesicles present in the xanthophores of examined turtles. (*a*) Electron dense vesicles present predominantly in the PM of *Trachemys scripta elegans*; (*b*) vesicles with lower electron density present predominantly in the ZP of *Trachemys scripta scripta*; (*c*) pterinosomes; (*d*) detail of electron dense vesicles from (*a*), these vesicles lack any internal structure and possibly contain ketocarotenoids; (*e*) detail of vesicles with lower electron density shown in (*b*); (*f*) detail of pterinosomes shown in (*c*), typical by concentric lamellae. Scalebar 2 µm (*a,b,c*) and 500 nm (*d,e,f*). Note that magnification of (*a*) differ.

Reflectivity of collagen fibre arrays of *P. concinna* and *T. s. elegans* predicted by Fourier analyses is shown in figure 15 together with normalized measured reflectance spectra. These predictions describe how the reflecting shield of collagen fibres could contribute to the colour of the skin of turtles through its interaction with incident light. All predicted normalized reflectivity curves have three distinct peaks (360–400, 450–500 and 560–570 nm) except for the YP of *T. s. scripta* which has four peaks (370, 450, 520 and 570 nm; electronic supplementary material, figure S5b). All predicted normalized reflectivity curves increase beyond 600 nm, but in the CBC of both *T. s. elegans* (figure 15*b*) and *P. concinna* (figure 15*e*) there is an additional peak/plateau at 660–680 nm. The overall shapes of the predicted normalized reflectivity of collagen fibre arrays are more similar among regions than among species. The PMs of both species (figure 15*a,d*) have few distinct predicted peaks with a pronounced peak at 560 nm in the PM of *P. concinna*. The CBC of both species (figure 15*b,e*) have four significant predicted peaks with the highest at 490–500 nm. The FLBS (figure 15*c,f*) are characterized by predicted major peaks in the UV spectrum at 360–380 nm. As most of the predicted reflectivity curves increase beyond 600 nm we conclude that the reflective shield of collagen fibres serves primarily to protect against overheating by reflecting infrared wavelengths.

Peaks of the predicted normalized reflectivity of collagen fibre arrays correspond roughly to peaks of normalized measured reflectance spectra. However, there are differences in the overall shapes of the predicted and measured curves. Differences between measured and predicted spectra probably arise due to interactions between colour-producing pigment cells and the intercellular matrix of collagen fibre arrays. Even though the radial means of the Fourier power spectra of the FLBS of *P. concinna* suggest appropriate spatial frequencies to produce colour by coherent scattering, the predicted reflectivity curve does not match the measured reflectance spectra. However, at least in the CBC of *P. concinna* the organization of collagen fibres, the radial means of the Fourier power spectra, and the congruence of the predicted reflectivity curve with the measured reflectance spectra all suggest a role of collagen fibre arrays in colour production by coherent scattering. The majority of the Fourier power falls in the infrared part of the light spectrum, the contribution of collagen fibre arrays to the overall reflectance being only minor in most other regions. This suggests that the main function of collagen

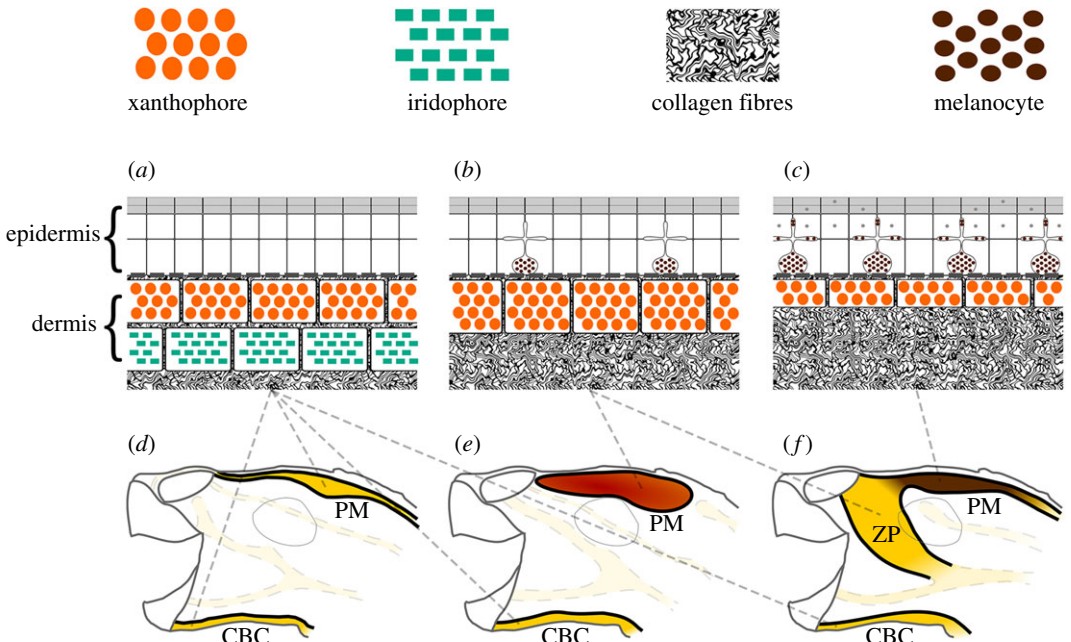

**Figure 10.** Schematic diagram of cellular composition of colourful region of Emydinae. Ventral chin stripe (CBC), and dorsal postorbital marking (PM) and zygomatic patch (ZP) of examined turtles composed of epidermal melanocytes and dermal xanthophores and iridophores. Both ventral and dorsal regions of *Pseudemys concinna* (*d*) consist of xanthophores and iridophores (*a*). However, integument of *Trachemys scripta elegans* (*e*) and *T. s. scripta* (*f*) contain xanthophores together with iridophores on ventral side. On dorsal side there are only xanthophores in the dermis (*b,c*) present in both subspecies of *T. scripta*. There are no epidermal melanocytes in neither PM nor CBC of *P. concinna* (*a*), but there are epidermal melanocytes in dorsal integument of *T. scripta*. These epidermal melanocytes are small, without apparent melanosome transferring activity (*b*) in red PM of *T. s. elegans* and yellow zygomatic region (ZP) of *T. s. scripta*. In dark PM region of *T. s. scripta* (*c*) epidermal melanosomes are enlarged and epidermal keratinocytes contain transferred melanosomes (grey dots). The dermis contains abundant collagen fibres.

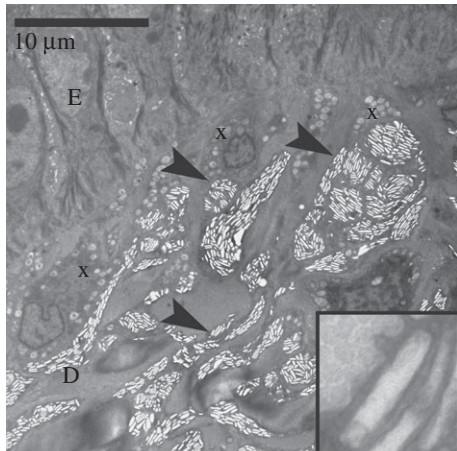

**Figure 11.** Electromicrograph of iridophores in the turtle integument. E: epidermis; D: dermis; x: xanthophore; arrowheads: iridophore. In the inset is detailed view of reflecting platelet.

fibres is to reflect infrared radiation and only in some instances collagen fibres have been co-opted to function in colour production.

## 3.5. Pigment types

Results of pigment diversity analyses for carotenoids, pterins and riboflavin are summarized in table 4. Examples of UPLC chromatograms resulting from analyses of carotenoids can be found in the

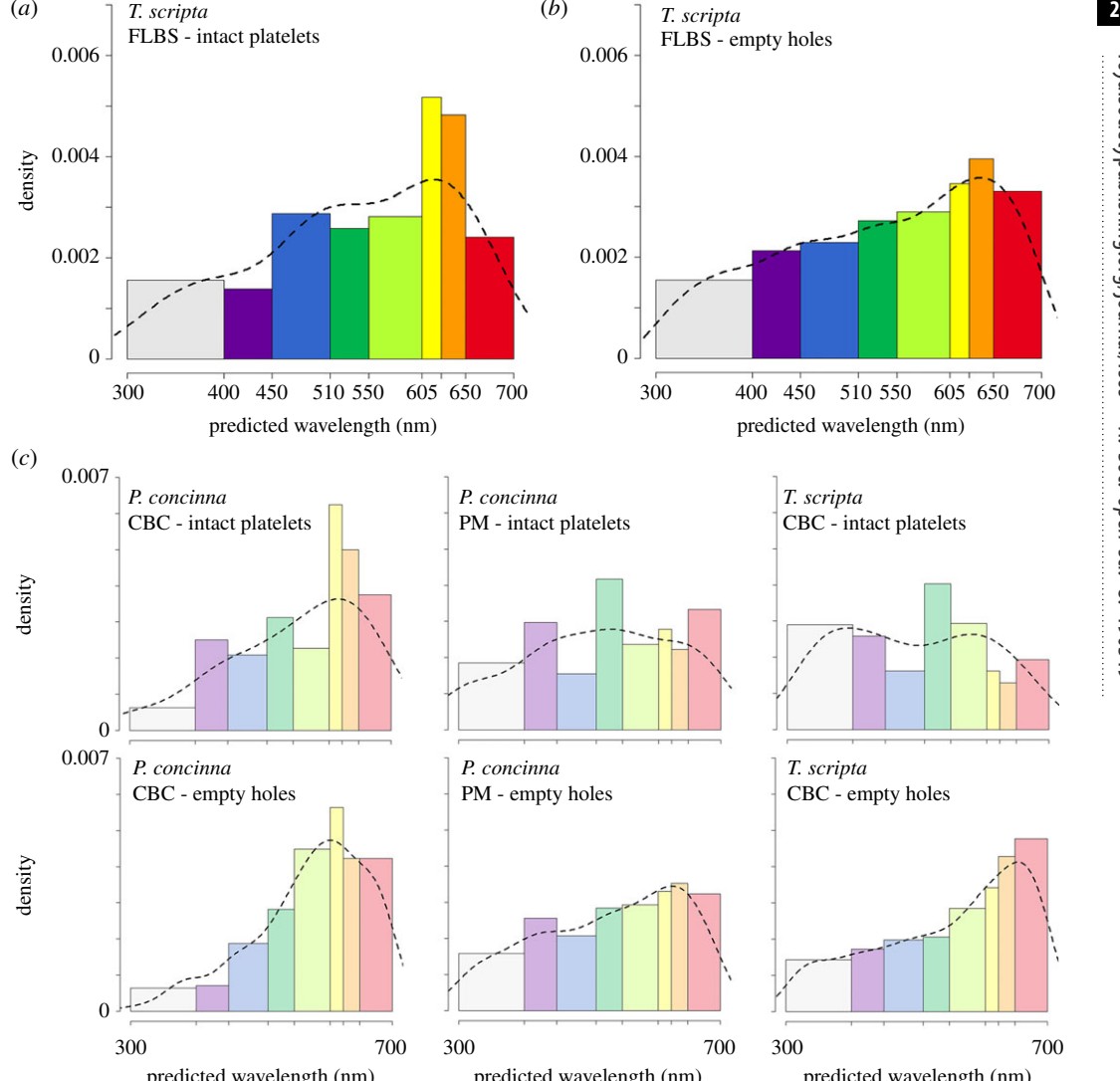

**Figure 12.** Distribution of predicted reflectivity of reflecting platelets of iridophores based on thin layer interference model. (a) Based on size of intact reflecting platelets of FLBS—yellow front limb region of *Trachemys scripta*. (b) Based on size of empty holes of FLBS region of *T. scripta* left after staining of EM sample. Dashed line in (a) and (b) represent shape of reflectance spectra predicted from reflecting platelets properties based on thin layer interference model. (c) Distribution of predicted reflectivity of reflecting platelets of iridophores in regions where assumption of thin layer interference model does not seem to be appropriate.

supplementary materials (electronic supplementary material, figure S1b,g,h,i,j). The red PM of *T. s. elegans* contains a rich mixture of various carotenoids which, according to UV/VIS absorbance spectra, are predominantly ketocarotenoids. The two largest retention peaks were determined as astaxanthin (RT = 11.8 min, $\lambda_{max} = 471$ nm, $[M + H]^+ = 597.3944$) and canthaxanthin (RT = 13.7 min, $\lambda_{max} = 466$, $[M]^+ = 565.4046$). Other regions (yellow CBC, yellow FLBS of both species, yellow PM of *P. concinna*, and dark PM and yellow ZP of *T. s. scripta*) had no ketocarotenoids, but contained a relatively complex mixture of hydroxylated xanthophylls, of which lutein (RT = 12.9 min, $\lambda_{max} = 444$, $[M]^{\bullet+} = 568.4254$) and zeaxanthin (RT = 13.1 min., $\lambda_{max} = 449$, $[M]^{\bullet+} = 568.4253$) was by far the most abundant.

Examples of SRM chromatograms of pterins are given in the supplementary material (electronic supplementary material, figure S6). Out of 10 pterins tested, seven were found in the integument of the turtles analysed. L-sepiapterin, erythropterin and drosopterin were not detected in any sample. All other pterins, i.e. 6-biopterin, pterin-6-COOH, isoxanthopterin, leucopterin, D-neopterin, pterin and xanthopterin, were found together only in the red PM of *T. s. elegans*. The yellow CBC and FLBS of both lineages of *T. scripta* and the ZP of *T. s. scripta* contain 6-biopterin, pterin-6-COOH, isoxanthopterin, leucopterin, D-neopterin, and xanthopterin, but not pterin. Only four types of pterins,

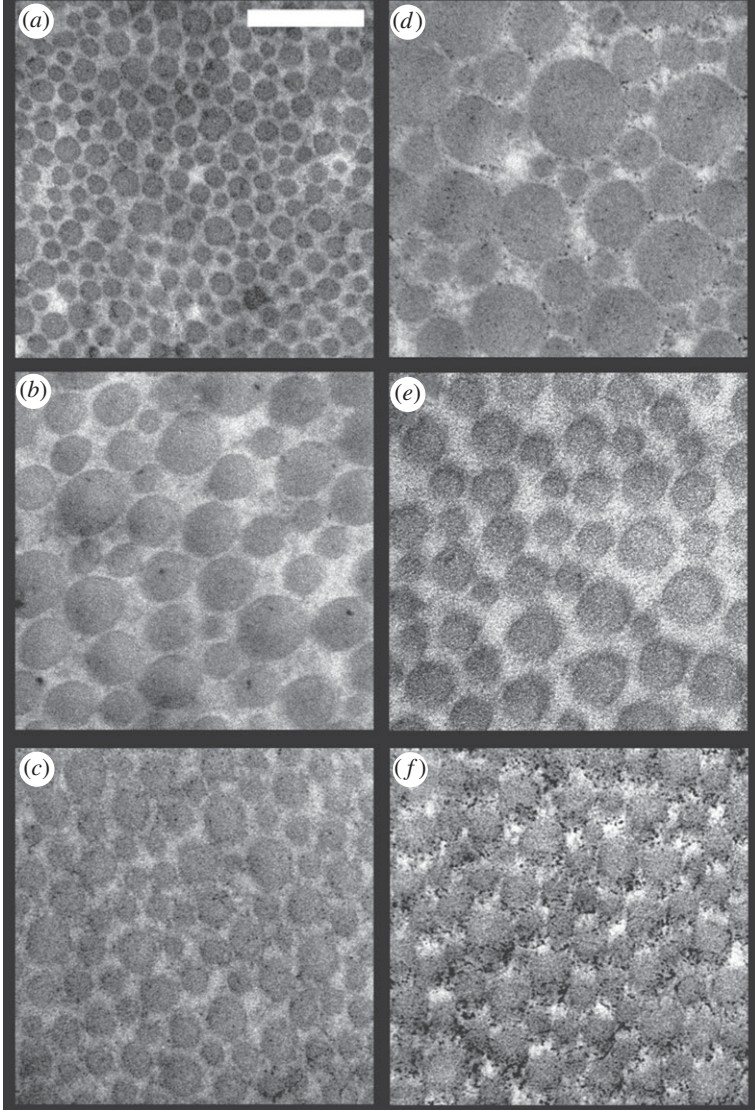

**Figure 13.** Electromicrographs of details of collagen fibre arrays. (*a*) Postorbital marking (PM) of the *Trachemys scripta*. (*b*) Main median chin yellow stripe (CBC) of *T. scripta*. (*c*) Main bright stripe forelimb stripe of *T. scripta*. (*d*) PM region of *Pseudemys concinna*. (*e*) CBC region of *P. concinna*. (*f*) FLBS region of *P. concinna*. Scalebar represent 200 nm.

namely 6-biopterin, pterin-6-COOH, leucopterin and D-neopterin, were detected in the CBC, FLBS and PM of *P. concinna*. Riboflavin was found in all yellow-red samples except the yellow PM of *P. concinna*.

## 4. Discussion

Although animal coloration research has traditionally focused on other groups, many turtles have bright, conspicuous colours [75]. Information on the mechanisms of turtle colour production is particularly scant [27,29,30]. In fact, there is more information on colour production in some fossil extinct taxa than in extant turtles [76–82]. Research on the functional role of turtle chromatic signals is also limited [42]. Given the revised phylogenetic position of turtles as part of archelosauria [22,83–85], studying turtle coloration and colour-producing mechanisms is crucial to understanding the evolution of colour and colour-producing mechanisms in vertebrates in general.

In this study we have analysed the pigment cell organization of the integument, the ultrastructure of colour-producing elements, and the chemical nature of pigments that produce skin colour in *Pseudemys concinna* and *Trachemys scripta*, two species of freshwater turtles with contrasting courtship behaviour. We found striking interspecific differences in the distribution of chromatophores in the dorsal and ventral head surfaces. *Trachemys scripta* shows a different chromatophore composition in the ventral and

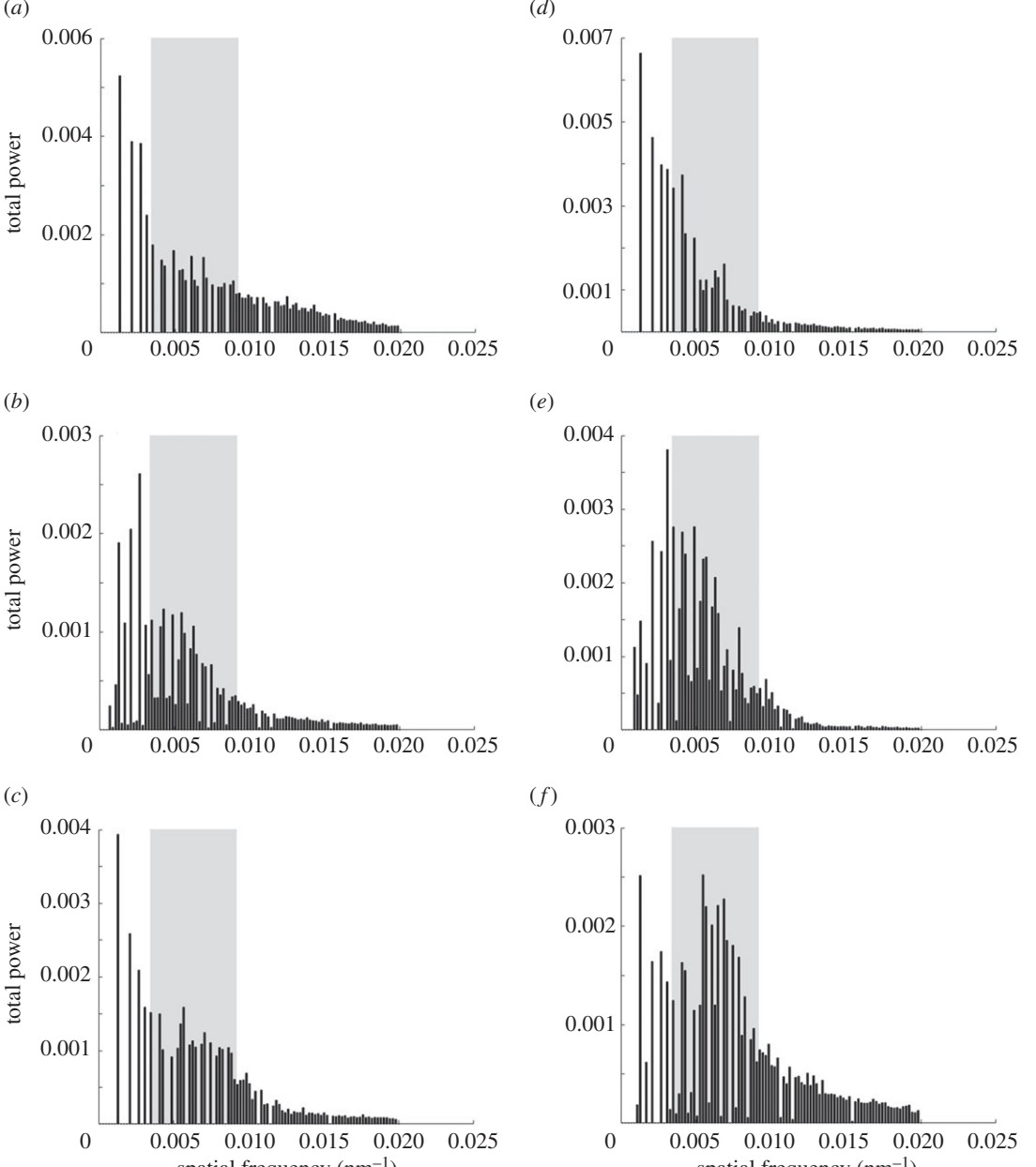

**Figure 14.** Radial means of the Fourier power spectra of electromicrographs of collagen fibre arrays. (*a*) Postorbital marking (PM) of the *Trachemys scripta*. (*b*) Main median chin yellow stripe (CBC) of *T. scripta*. (*c*) Main bright stripe forelimb stripe of *T. scripta*. (*d*) PM region of *Pseudemys concinna*. (*e*) CBC region of *P. concinna*. (*f*) FLBS region of *P. concinna*. Grey rectangle denotes spatial frequencies relevant to coherent scattering in visible light, on the other hand lower frequencies (to the left from the grey rectangle) are relevant to scattering in the infrared part of light spectrum.

dorsal sides of the head, whereas *P. concinna* shows similar chromatophore composition on both sides. Also, there are more pigment types present in the colour patches of *T. scripta* than in those of *P. concinna*.

Turtle skin coloration is produced by the combined action of different types of pigment cells in the dermis similarly to amphibians and lepidosaurs. In addition to xanthophores and melanophores, previously described in turtles [27] and also in crocodiles [86], we found abundant iridophores containing rectangular reflecting platelets in yellow skin of both species. To our knowledge, this is the first report of iridophores playing a role in integumental colour production in any archelosaurian species [86,87]. In addition, our pigment analyses suggest that both pterins and carotenoids are involved in the production of the yellow-red skin colours, which had not been clearly documented in the integument of turtles to this date.

We also show that abundant collagen fibre arrays in the skin may serve to prevent infrared light from penetrating into the body of turtles, but in some cases collagen fibre arrays may be capable of reflecting

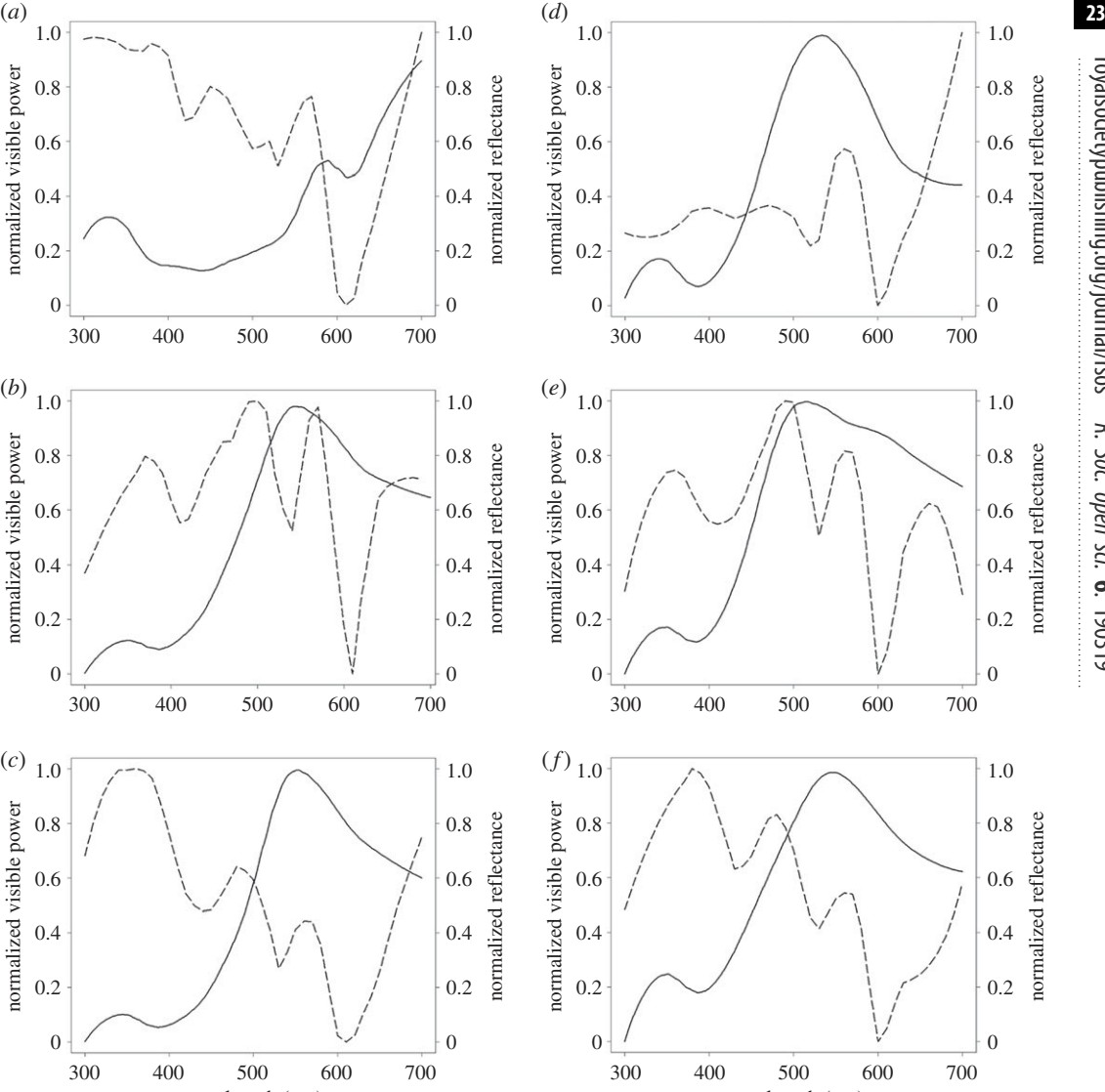

**Figure 15.** Measured reflectance spectra of the skin regions (solid line) and reflectance spectra predicted by Fourier analyses of dermal collagen fibres (dashed line). (a) Postorbital marking (PM) of the *Trachemys scripta*. (b) Main median chin yellow stripe (CBC) of *T. scripta*. (c) Main bright stripe forelimb stripe of *T. scripta*. (d) PM region of *Pseudemys concinna*. (e) CBC region of *P. concinna*. (f) FLBS region of *P. concinna*. Note that both measured reflectance and Fourier power are both normalized to have minimum zero and maximum one.

visible wavelengths and therefore affect colour. Taken together, these results suggest that turtle coloration and colour-producing mechanisms are more complex than previously thought.

## 4.1. The coloration of deirocheline turtles: interspecific and sex differences

Our results reveal interspecific differences in colour-producing mechanisms at multiple levels: differences can be found in the distribution of chromatophores along the dorsoventral body axis, in the ultrastructure of pigment-bearing vesicles of xanthophores, and in the chemical nature of pigments. As for the distribution of chromatophores, *T. scripta* has both xanthophores and iridophores in the yellow-red regions of the ventral side of the head, but only xanthophores on the dorsal side. Such dorsoventral patterning is lacking in *P. concinna* which has xanthophores and iridophores on both sides of the head (figure 10). As iridophores increase the overall reflectance of a patch of skin [4,20,88], their presence on the ventral side of the head may provide some sort of countershading to enhance crypsis [89–91] or may be related to signalling (see below). The fact that both species of turtles also have a light plastron, but a dark coloured carapace is consistent with the hypothesis that the increased reflectance

caused by the presence of iridophores on the ventral side of the head might serve a camouflage function. The background head coloration (DHC), which has been previously reported to change in response to substrate colour [37,92–94], is produced in both *T. scripta* and *P. concinna* by epidermal melanosomes and a combination of all three dermal pigment cell types (figure 7*e,f*). However, we found epidermal melanocytes in the PM of both lineages of *T. scripta* (figures 4 and 8*a–c*), but not in the PM of *P. concinna* (figures 5 and 8*e,f*) or in other yellow regions of the examined turtles (figure 7*b,d*).

Our results show that the conspicuous yellow stripes in the head and limbs of deirocheline turtles exhibit, in addition to a primary reflectance peak responsible for the human-perceived yellow hue, a small secondary peak of reflectance in the UV range of the spectrum. This confirms the results of several previous studies [30,95]. However, we did not find support for the spectral shapes reported by Wang *et al.* [96]; in fact, we contend that the small 372 nm blip that these authors found in all their reflectance spectra (see their fig. 1 [96]) is an artefact possibly caused by the illumination source. As turtles are capable of seeing into the UV range [97,98], the small peak of UV reflectance may account for chromatic variation among different individuals and contribute to the conspicuousness of the yellow stripes, at least during close-range interactions with conspecifics [30].

We found intersexual differences in the reflectance spectra of CBC and FLBS of *T. s. elegans* but not *P. concinna* (table 1). These results contrast with a previous study reporting no intersexual differences in coloration in *T. s. elegans* [84]. The small sample sizes may be responsible for our inability to statistically support the effect of sex on colour in *P. concinna*. Sexual dichromatism is apparently present in some deirocheline turtles. For example, males of the northern map turtle (*Graptemys geographica*) have brighter postorbital patches than females [36]. However, in other species, such as *Chrysemys picta* [37] and *Malaclemmys terrapin* [35], there is little or no sexual dichromatism. Sexual dichromatism has been hypothesized to result from females becoming duller as they grow and age [37]. Growth is known to affect the reflectance spectra of colourful regions in some turtles [99]. Moreover, males of some species of deirocheline turtles undergo ontogenetic colour change as they get larger and older [49,100]. As we have excluded melanistic males from the spectral measurements, the sexual dichromatism of *T. s. elegans* found here is not due to ontogenetic colour change of males. Nevertheless, it will be interesting in the future to examine the postnatal ontogenesis of colour-producing mechanisms in deirocheline turtles to see how sexual dichromatism arises as the turtles grow and age.

## 4.2. Mechanisms of colour production

The red and yellow regions of turtle skin contain at least three morphologically distinct types of vesicles in xanthophores. Two of these have been previously [72,74,101] identified as carotenoid vesicles, and differ in electron density, shape and size (figure 9*a,b,d,e*). The third type of xanthophore vesicles are pterinosomes, likely containing pterins (figure 9*c,f*). These are recognizable by their prominent concentric lamellae [73,74]. Xanthophores containing oval carotenoid vesicles with low electron density (figure 9*a,d*) are abundant only in the red PM of *T. scripta elegans*, while xanthophores containing round carotenoid vesicles with high electron density (figure 9*b,e*) and/or pterinosomes (figure 9*c,f*) are found in yellow regions of both species. We assume that differences in the ultrastructure of carotenoid vesicles reflect differences in their pigment contents, as in the case of red and yellow retinal oil droplets [102] known to differ in carotenoid content [31,103,104]. However, similar structures have been previously described as vesicles in different stages of maturation [105]. Further study using appropriate methods is necessary to unambiguously determine the types of pigments contained in vesicles with different ultrastructure.

Continuous layers of iridophores with organized reflecting platelets are found in *T. scripta* but not in *P. concinna*. The colour produced by the scattering of incident light on arrays of particles depends on the organization, size and the refractive index of particles [106]. Incoherent scattering occurs when the distances between individual particles are random (e.g. reflecting platelets are randomly arranged in iridophores) causing the phases of waves of light to nearly cancel out. In contrast, coherent scattering occurs when the distances between individual particles are similar (e.g. when reflecting platelets are organized in iridophores) causing the phases of waves of light either to completely cancel out or add up to produce relatively high reflectance. As the reflecting platelets of iridophores from the FLBS of *T. scripta* are highly organized they could produce colour by coherent scattering. Highly organized reflecting platelets in orange and yellow skin regions of the lizards *Uta stansburiana* [60], *Sceloporus undulatus* and *S. magister* [59] reflect orange and yellow wavelengths due to thin-film interference, which is a special case of coherent scattering. The thin-film interference model predicts the highest reflectivity of the FLBS of *T. scripta* to be between 618 and 637 nm (figure 12*a,b*). Other regions of

**Table 3.** Characteristics of reflective platelets of iridophores in CBC (main median chin yellow stripe), FLBS (main bright stripe of the fore limb) regions of *Pseudemys concinna*, and CBC, PM (postorbital marking) of *Trachemys scripta elegans*. The $A/y_0$ ratio represents the relative number of reflecting platelets parallel to the skin surface. A: amplitude of Gaussian curve above the background of randomly oriented platelets; $y_0$: background level of randomly oriented platelets; FWHM: full width of half maximum of the peak of Gaussian curve fitted to the data, i.e. width of angular distribution of reflecting platelets.

| variable | P. concinna | | T. s. elegans | |
| --- | --- | --- | --- | --- |
| | CBC | PM | CBC | FLBS |
| number of platelets measured—intact | 65 | 100 | 59 | 74 |
| number of platelets measured—empty holes | 456 | 625 | 2777 | 2366 |
| mean minor axis length—intact ± s.d. (nm) | 143.44 (64.52) | 91.5 (44.57) | 86.34 (30.65) | 85.85 (33.49) |
| mean minor axis length—empty holes ± s.d. (nm) | 124.63 (47.55) | 100.95 (47.55) | 104.43 (51.20) | 93.95 (42.65) |
| mean major axis length—intact ± s.d. (nm) | 457.51 (221.97) | 414.93 (222.31) | 370.4 (145.88) | 395.52 (179.60) |
| mean major axis length—empty holes ± s.d. (nm) | 420.30 (223.63) | 333.70 (208.98) | 364.25 (240.22) | 347.40 (221.04) |
| $A/y_0$ ($A;y_0$) | 10.85 (0.0090; 0.0008) | 4.91 (0.0069; 0.0014) | 6.31 (0.0076; 0.0012) | 607.28 (0.0158; 0.000026) |
| FWHM (°) | 95.3 | 112.42 | 106.24 | 59.19 |
| predominant orientation (°) | 2.49 | 68.3 | 30.86 | −1.98 |

turtle skin containing iridophores, i.e. the yellow CBC of *T. scripta* and all yellow regions of *P. concinna*, do not have organized reflecting platelets. Iridophores with randomly organized platelets in red and white skin of *Phelsuma grandis* produce broadband reflection which enhances the overall brightness of the corresponding skin patch [16]. The disorganized prismatic guanin crystals of some spiders produce matte-white colour by incoherent (diffuse) light scattering [107]. However, prismatic crystals of spiders are around one micrometre in size [108], while the disorganized reflecting platelets of fish [107], lizards [16] and turtles (table 3) are much smaller. The yellow regions containing iridophores in *T. scripta* have increased brightness relative to yellow regions without iridophores (figure 2). Thus even if they do not produce colour through thin-film interference, the iridophores most likely increase the brightness of colour patches and their overall conspicuousness to observers, thus contributing to colour [4,109]. Whether or not the scattering on disorganized reflecting platelets in turtles is coherent or incoherent remains to be answered.

All yellow regions of both species contain hydroxylated xanthophylls such as lutein and zeaxanthin, while the red PM of *T. s. elegans* contains the ketocarotenoids astaxanthin and canthaxanthin (table 4; electronic supplementary material, figure S1). Such difference is in partial agreement with previous analyses of carotenoid contents in yellow and red regions of *T. s. elegans* and *Chrysemys picta* [30]. Two major classes of carotenoids have been described in the integument of these turtles: short wavelength absorbing apocarotenoids and longer wavelength absorbing ketocarotenoids [30]. In the yellow chin of *C. picta* only apocarotenoids were described, whereas the orange neck and leg contained apocarotenoids and ketocarotenoids. A previous report indicated that apocarotenoids are abundant in tissue from yellow chin and neck stripes of *T. scripta*, its 'orange' postorbital region containing only ketocarotenoids [30]. Our results, however, suggest a major role of xanthophylls, rather than apocarotenoids, in yellow colour production in *T. scripta* and *P. concinna*. Xanthophylls have been shown to produce yellow coloration in some vertebrates [110–112]. In *C. picta*, increased ingestion of the xanthophyll carotenoid lutein results in an increase in the yellow and red chroma of yellow and red skin patches [28]. In contrast, it is unclear whether apocarotenoids participate in colour production in animals. Apocarotenoids are cleavage products of carotenoids [113–115]. Transparent

**Table 4.** Presence (✓ present/✗ absent) of different types of pterins, predominant carotenoids and riboflavin in different regions of the integument in the studied taxa (*Pseudemys concinna*, *Trachemys scripta elegans*, *Trachemys scripta scripta*). CBC: main median chin yellow stripe; FLBS: main bright stripe of the fore limb; PM: postorbital marking; ZP: yellow zygomatic patch.

| taxon | region | pterins | | | | | | | | | |
| --- | --- | --- | --- | --- | --- | --- | --- | --- | --- | --- | --- |
| | | 6-biopterin | D-neopterin | drosopterin | erythropterin | isoxanthopterin | L-sepiapterin | leucopterin | pterin | pterin-6-COOH | xanthopterin |
| P. concinna | CBC | ✓ | ✓ | ✗ | ✗ | ✗ | ✗ | ✓ | ✗ | ✓ | ✗ |
| | FLBS | ✓ | ✓ | ✗ | ✗ | ✗ | ✗ | ✓ | ✗ | ✓ | ✗ |
| | PM | ✓ | ✓ | ✗ | ✗ | ✗ | ✗ | ✓ | ✗ | ✓ | ✗ |
| T. s. elegans | CBC | ✓ | ✓ | ✗ | ✗ | ✓ | ✗ | ✓ | ✗ | ✓ | ✓ |
| | FLBS | ✓ | ✓ | ✗ | ✗ | ✓ | ✗ | ✓ | ✗ | ✓ | ✓ |
| | PM | ✓ | ✓ | ✗ | ✗ | ✓ | ✗ | ✓ | ✓ | ✓ | ✓ |
| T. s. scripta | CBC | ✓ | ✓ | ✗ | ✗ | ✓ | ✗ | ✓ | ✗ | ✓ | ✓ |
| | FLBS | ✓ | ✓ | ✗ | ✗ | ✓ | ✗ | ✓ | ✗ | ✓ | ✓ |
| | ZP | ✓ | ✓ | ✗ | ✗ | ✓ | ✗ | ✓ | ✗ | ✓ | ✓ |

| | | predominant carotenoids | | | | flavins |
| --- | --- | --- | --- | --- | --- | --- |
| | | astaxanthin | canthaxanthin | lutein | zeaxanthin | riboflavin |
| P. concinna | CBC | ✗ | ✗ | ✓ | ✓ | ✓ |
| | FLBS | ✗ | ✗ | ✓ | ✓ | ✓ |
| | PM | ✗ | ✗ | ✓ | ✓ | ✗ |
| T. s. elegans | CBC | ✗ | ✗ | ✓ | ✓ | ✓ |
| | FLBS | ✗ | ✗ | ✓ | ✓ | ✓ |
| | PM | ✓ | ✓ | ✗ | ✗ | ✓ |
| T. s. scripta | CBC | ✗ | ✗ | ✓ | ✓ | ✓ |
| | FLBS | ✗ | ✗ | ✓ | ✓ | ✓ |
| | ZP | ✗ | ✗ | ✓ | ✓ | ✓ |

and clear cone oil droplets in the retina of birds contain the apocarotenoid galloxanthin, but yellow oil droplets contain the xanthophyll zeaxanthin and red oil droplets contain the ketocarotenoid astaxanthin [116]. The unique yellow colour of the *macula lutea* in primates is due to the accumulation of xanthophylls in the cells of the retina via downregulation of their cleavage pathway to apocarotenoids [117]. Various carotenoid precursors give rise to a large diversity of apocarotenoids [113], but specific types of apocarotenoids have not been reported in previous studies of turtle coloration [30]. The role of apocarotenoids in turtle coloration will thus remain unclear until the relationship between precursor carotenoids in the skin and apocarotenoids is elucidated.

Steffen *et al*. [30] suggested that the yellow-red skin of *C. picta* and the yellow skin (but not the red skin) of *T. scripta* contain small amounts of pterins. Our results show that both yellow and red regions of turtle skin contain multiple types of pterins (table 4). Red and orange colours are produced in some lizards by drosopterin [118,119] which we have not found in turtles. However, yellowish isoxanthopterin and yellow xanthopterin are present in yellow-red regions of *T. scripta*. Xanthopterin has been previously reported in yellow and red skin of *Phelsuma* lizards [16]. The yellow and red skin regions of *Phelsuma* have identical pigment composition but differ in the cellular pH environment which determines the colour of the skin [16]. Thus, it is possible that xanthopterin and isoxanthopterin could contribute to both yellow and red colour production in the skin in turtles due to differences in the cellular pH among these regions. However, most of the pterins present in the skin of both species of turtles are colourless compounds. Colourless biopterin and 6-pterin-COOH have been previously described in the skin of *M. japonica* [32]. In lizards colourless pterins have been found in colourful body regions. For example, biopterin has been detected in the yellow-red xanthophores of *Phelsuma* spp. [16], and varying amounts of colourful and colourless pterins have been found in the yellow-orange-red ventral colour morphs of the European wall lizard (*Podarcis muralis*) [72]. Since many of these colourless pterins absorb in the UV range [16], which turtles can see [97,98], colourless pterins may in fact affect colour perception in turtles. Moreover, although some pterins may be colourless even to turtles, it may be difficult to separate their influence on colour production from that of colourful pterins, because the colourful pterins are derived from colourless ones and *vice versa*. For example, yellow sepiapterin is a precursor of colourless biopterin (the precursor of yellow 7-oxobiopterin and pterin) and colourless pterin, the precursor of yellowish isoxanthopterin and its isoform yellow xanthopterin [120]. The content of all individual types of pterins thus depends on the availability of their precursor pterins. The diversity of different pterin types may thus describe the differences between colourful skin regions better than the amount of any single pterin type. In addition to carotenoids and pterins, we found riboflavin in all yellow and red regions of *T. scripta*. In *P. concinna* we found riboflavin in the yellow CBC and FLBS but not in the yellow PM. Riboflavin is present in the yellow-red skin of various lizards and snakes but has never been reported in turtle skin [32,72,121]. Therefore, the yellow and red colours of the turtles examined here likely result from a complex interplay between carotenoids, pterins and riboflavin.

## 4.3. Functional significance of turtle coloration: relationship to courtship behaviour

Whether the differences in colour and colour-producing mechanisms among deirocheline turtles are of functional significance has yet to be determined. However, it is already well established that turtles are capable of visually discriminating different yellow and red colours [35,97,98] and it has been suggested that body coloration may play a role in mate choice in emydid turtles [30,35,36]. Furthermore, there is some evidence that the red and yellow skin regions of *T. scripta* could play a role in signalling individual quality based on the correlation of colour and induced immune response [122]. Moreover, the spectral properties of colour patches of *C. picta* are influenced by the amount of ingested carotenoids [28] which is one of the basic assumptions of individual quality signalling [123–125].

Is there a relationship between body coloration, colour-producing mechanisms and male-female position during courtship? Males of *T. scripta* court females in a face-to-face position with forelimbs outstretched, so that both the dorsal and ventral sides of the head and the limbs of the male are simultaneously exposed to the female. On the other hand, males of *P. concinna* swim above the female during the final stages of courtship [45]. In this position it may be more difficult for the female to perceive the dorsal side of the male's head. If coloration affects male sexual attractiveness, one would expect those skin regions that males display during courtship to be under selection for conspicuousness and/or condition dependence. This may result in skin patches that are particularly bright (i.e. high luminance), to facilitate detection by the female, or that have a complex chromatophore composition

allowing for condition-dependent signalling (e.g. xanthophores containing carotenoids). Some aspects of the coloration of *T. scripta* and *P. concinna* are consistent with this hypothesis while others clearly are not.

The red and yellow patches in the head and forelimbs of *T. scripta* contain a more diverse palette of pterins than the yellow patches of *P. concinna* (table 4). The face-to-face position may facilitate female assessment of male coloration in *T. scripta*, where the red and yellow patches could function as signals conveying information about individual male quality. Conversely, the relatively simple pigment composition of species that court using the swim-above position would suggest a reduced role for male head coloration in female mate assessment.

*Trachemys scripta* and *P. concinna* also differ in the distribution of chromatophores along the dorsoventral body axis (figure 10). *Pseudemys concinna* has no epidermal melanosomes in the PM, and xanthophores and iridophores are present both dorsally (PM) and ventrally (CBC) in the head. This should make *P. concinna* rather conspicuous when viewed from above. On the other hand, *T. scripta* has red-yellow xanthophores and iridophores in the ventral CBC, but not in the dorsal PM. Thus, the location of the potentially brighter, more conspicuous skin patches the head of the males of the two species does not match predictions based on the relative position of males and females during courtship.

Female mating preferences for colourful males often result in sexual dichromatism, with brightly coloured males and comparatively dull females [126]. Males of *T. scripta*, which expose their CBC, FLBS and PM to females during courtship, have brighter CBC and FLBS than females (table 1). Males of *T. scripta* also have increased red chroma in the PM region compared to females, although the difference is not statistically significant after correcting for multiple comparisons (figure 2; electronic supplementary material, table S4). In *P. concinna*, on the other hand, there are no sex-related differences in coloration (figure 2, table 1).

Other factors besides courtship position could contribute to interspecific variation in colour and colour-producing mechanisms in deirocheline turtles, including habitat type, predation and evolutionary history. The pattern of alternating dark and light stripes that is typical of emydid turtles could provide camouflage through background matching (e.g. when viewed against underwater vegetation) or as disruptive coloration creating false edges and visually breaking the outline of the body, thus hindering detection of turtles by predators [127]. The presence of iridophores ventrally but not dorsally in the head could enhance the cryptic appearance of *T. scripta* in aquatic environments by countershading (see above). It is not clear, however, why *P. concinna* does not possess a similar distribution of pigment cell types and instead has iridophores in the dorsal and ventral head colour patches. Because mating strategies are well-studied in this group, it may be relatively straightforward to test hypotheses about the role of natural and sexual selection in shaping the evolution of colour across the Deirochelinae when more data on colour and colour-producing mechanisms become available. Finally, it is important to note that proximate developmental processes may be responsible for differences in coloration between *T. scripta* and *P. concinna* which may not necessarily relate to visual signalling or crypsis, but rather to the intrinsic self-organizing properties of pigment cells [105,128–131]. Further studies are needed to clarify the development of pigment pattern formation in turtles [132,133].

## 4.4. The reflective shield of collagen fibres

Results of Fourier analyses show that turtle collagen fibres reflect in the infrared part of the spectrum in all regions examined (figure 14). Long wavelengths of light that pass through the xanthophores and iridophores may be absorbed by the melanophores, or they may be reflected back to the surface by the underlying collagenous connective tissue when melanophores are absent or scarce [134]. In many fish and amphibian larvae, the collagen fibres form subepidermal transparent collagenous lamella overlying the dermal chromatophores [20,135,136]. In lizards and snakes collagenous connective tissue is found in the *superficial fascia* between the integument and the underlying muscle [134], but in other animals such as frogs [137], mammals [17] or birds [18] collagen fibres are part of the dermis. Here we show highly abundant dermal collagen fibre arrays located below the chromatophore layers in turtles (figures 4 and 5). Unlike lizards [16,60,138], turtles lack a layer of dermal melanophores in the red-yellow skin regions. Turtles bask to thermoregulate and overheating may represent an immediate risk for them [139], but basking could also have a social function [140] for which it may be advantageous to accumulate heat at a slow rate. Therefore, the function of dermal collagen fibres may be to prevent infrared wavelengths from reaching the body core to avoid overheating. A thermoprotective function of colour-producing structures has been suggested for iridophores found in

frogs [141] or chameleons [58]. A thermoprotective function has also been suggested for the keratinous feathers of birds that reflect infrared wavelengths due to their microstructure [142,143]. The role of melanin in thermoregulation is well documented in a broad range of organisms [144]. Thus, it seems that colour-producing elements often acquire a thermoregulatory function and, conversely, skin structures that play a role in thermoregulation may in some instances be co-opted to produce visible colour.

Colour production by coherent scattering on nanostructured collagen fibre arrays has been previously reported in mammals [17] and birds [15,18]. In mammals the colour produced by scattering on collagen fibres is limited to the blue part of the light spectrum [17], but in birds other colours, such as orange in *Trogopan coboti*, are produced by coherent scattering on collagen fibre arrays [18]. The collagen fibres in the yellow FLBS of *P. concinna* show sufficient nanostructural organization to produce yellow by coherent scattering (figure 14*f*). However, in most other regions the collagen fibres are arranged randomly and thus do not produce bright colours by coherent scattering. Even though the light reflected by collagen fibre arrays is largely achromatic it could nonetheless affect the reflectance spectra, as suggested by [4]. As in the CBC and FLBS of *P. concinna* chromatophores are scarce, it is possible that collagen fibres play a major role in colour production in this species.

## 4.5. Significance for vertebrate skin colour evolution

Pigment cells are a defining synapomorphy of vertebrates and tunicates [10]. Vertebrates show two fundamental mechanisms of colour production by pigment cells: morphological and physiological. The former is based on the superposition of different dermal pigment cells (iridophores, melanocytes and xanthophores), and is found in ectotherms [145]. Physiological mechanism of colour production based solely on physiological actions of epidermal melanocytes [146] have evolved only in endothermic tetrapods, i.e. birds and mammals [145]. The archelosauria, comprising ectothermic turtles and crocodiles and endothermic birds [22], has undergone dramatic evolutionary changes affecting the physiology of melanocytes [147,148]. Also, iridophores and xanthophores have evolved differently in the main archelosaurian clades [145,149–151]. Skin pigment cells of crocodiles are not organized into stacked continuous layers and their putative iridophores contain amorphous material rather than reflecting platelets [86]. Cells with lipid vesicles suspected to contain carotenoids have been reported in both the epidermis and the dermis of non-feathered bare skin of some bird taxa [18,152], but this association probably evolved secondarily and does not involve pigment cells but other cell types [151]. Even though xanthophores, iridophores and a broad diversity of pigment cells with unusual pigment vesicles are present in the irises of birds, these cell types are not present in the skin of birds [151]. Here we show that turtles develop all the types of pigment cells (epidermal melanocytes, dermal xanthophores, iridophores and melanocytes) in the skin and organize them as continuous stacked layers as in other ectothermic vertebrates. Thus, early archelosaurs most likely shared with fish [153,154], amphibians [155,156], and lepidosaurs [138,157,158] a mechanism of skin colour production by superposition of pigment cells. It remains to be determined when the evolutionary shift between morphological and physiological mechanism of colour production by pigment cells occurred in the evolution of archelosaurs and whether these changes were parallel or convergent to the emergence of physiological mechanism of colour production in mammals.

Xanthophores are pigment cells that contain carotenoids and pterins [12]. Carotenoids are important to produce colour in the feathers and skin of birds [111,159]. Moreover, except for rare occurrences in feathers of penguins [160] or natal down feathers of domestic chickens [161], pterins have so far been reported only in the iris of birds [162]. However, there is no evidence of carotenoids and pterins acting together to produce colour in the skin of any archelosaur. Analyses of the pigments involved in the production of yellow-red colours of the xanthophore containing skin of turtles suggest that pterins contribute, together with carotenoids, to these colours, which was previously reported in fish [123], amphibians [163], lizards [110,121] and in the irises of birds [151,162,164,165]. Thus, colour production by interplay between carotenoids and pterins presumably represents the ancestral condition in vertebrates. It remains open question whether the loss of xanthophores in the skin of avian archelosaurs led to rare occurrence of pterins in the skin and feathers.

The evidence thus suggests that studies of turtle coloration are crucial to understanding the evolution of chromatic diversity in the archelosaurian clade and in vertebrates in general, because turtles seem to employ most of the mechanisms of colour production known in other groups of vertebrates (figure 16). Moreover, the field of paleo-colour, which aims to reconstruct the coloration of extinct taxa, has grown dramatically in recent years, providing some of the most remarkable breakthroughs in our understanding of the evolution of animal coloration [78,79,147,167–169] and turtles, given their key

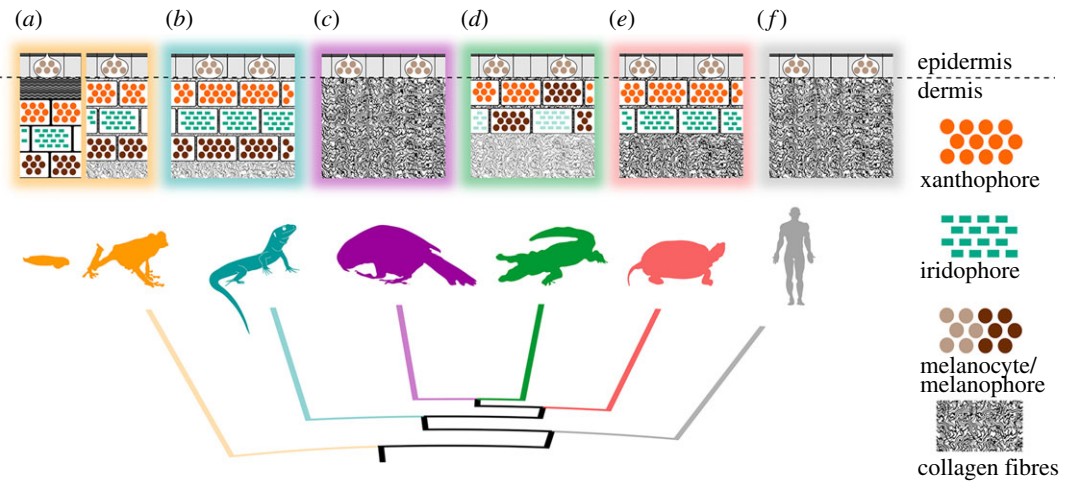

**Figure 16.** Phylogeny of extant tetrapods [166] illustrating distribution of colour-producing elements in the skin of tetrapods. All of the tetrapods may feature epidermal melanocytes in the epidermis. (*a*) Composition of colour-producing elements of amphibian larvae (on the left) is like that of fish. Basal lamina is underlined by layer of subepidermal collagen fibres beneath which are xanthophores, iridophores and melanophores/melanocytes located in the dermis. Adults of amphibians dermal xanthophores, iridophore and melanophores that may form dermal chromatophore units. (*b*) Lepidosaurs have colour-producing elements similar to those of adult amphibians. (*c*) Birds do not possess any pigment cell type besides epidermal melanocytes. Birds are known to produce colour by organized collagen fibre arrays. Means of colour production of plumage are not depicted. (*d*) Crocodiles have dermal pigment cells (xanthophores, iridophores, melanocytes) that do not form continuous layers. Iridophores of crocodiles have been reported to not contain rectangular reflecting platelets. (*e*) Turtles have a continuous layer of xanthophores in some regions; other regions may be characterized by presence of a continuous double layer of xanthophores together with iridophore. Dermal collagen fibres participate in colour production in turtles (*f*) and mammals. Mammals do not feature any additional pigment cell type in addition to the epidermal melanocytes.

phylogenetic position [83–85], may provide important insights for the proper interpretation of palaeontological data.

Ethics. No human tissues were used in this study. All the turtles used in this study were captured in the wild in accordance with Regional Decrees 14/2013 and 213/2009 on control measures on invasive alien species in the Valencian Community. All procedures performed were in accordance with the ethical standards of Direcció General de Medi Natural i d' Avaluació Ambiental, specially the 4/1994 Regional Act on Pets Protection. All applicable international, national and/or institutional guidelines for the care and use of animals were followed. No other permissions were required prior to conducting the research.

Data accessibility. Reflectance spectra data, reflectance spectra summary variables, R script of spectral analyses, reflecting platelets electromicrographs, reflecting platelets measurements and Fourier analyses output files: https://doi.org/10.5061/dryad.b68b048 [56].

Authors' contributions. J.B. conceived the study, designed the study, coordinated the study, measured reflectance spectra, collected sample tissues, carried out the histological laboratory work, carried out microscopical observation, carried out the statistical analyses, carried out reflective platelets measurements and analyses, carried out Fourier analyses, participated in carotenoid analyses, drafted manuscript; J.V.B. carried out the fieldwork, communicated with local authorities; Z.B. supervised and carried out the pterins analyses, helped to draft the manuscript; J.G. participated in Fourier analyses; M.H. carried out pterins analyses; K.K. supervised and financially supported J.B., supervised statistical analyses, helped to draft the manuscript; P.M. conceived the study, carried out the carotenoid analyses, helped to draft the manuscript; E.F. conceived the study, designed the study, financially supported the study, supervised histological and microscopical techniques, coordinated the study, drafted manuscript. All authors gave final approval for publication.

Competing interests. We have no competing interests.

Funding. J.B. was financially supported by the Charles University (SVV 260434/2018) by the Ministry of Culture of the Czech Republic (DKRVO 2019-2023/ 6.VII.a, 00023272). M.H. was financially supported by the Charles University, project GA UK No 760216 and the project of the Specific University Research (SVV260440).

Acknowledgements. We are thankful to Piscífatoria del Palmar for the exceptional hospitality. This study would have not been possible without Vicente Sancho and Ignacio Lacomba who pioneered turtle conservation in the Comunitat Valenciana and are responsible for J.B. and E.F. getting to know each other. This study would have not been possible without Alice Exnerová and Prof. Pavel Štys who put J.B. and Z.B. in contact. We are thankful to Centro de recuperación de fauna, Granja El Saler, for the veterinary expertise. We are thankful to

José Manuel Garcia-Verdugo, his laboratory personnel (Patricia García-Tárraga, Arantxa Cebrian-Silla, Mariana Fill), and Mario Soriano for support in electronic microscopy techniques. We are thankful to Jan Krajíček, who conducted the initial analyses of pterins. J.B. is thankful to Radek Šanda for his patience and support as Head of the Department of Zoology in the National museum. We are thankful to Jan Raška, Gerardo Antonio Cordero, and two anonymous reviewers for their fruitful comments on the manuscript. J.B. dedicates this study to his (growing) family.

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
