## [Reviewer comments · Royal Society Open Science]

Review History

RSOS-190319.R0 (Original submission)

Review form: Reviewer 1

Is the manuscript scientifically sound in its present form?

No

Are the interpretations and conclusions justified by the results?

Yes

Is the language acceptable?

No

Is it clear how to access all supporting data?

Yes

Do you have any ethical concerns with this paper?

No

Have you any concerns about statistical analyses in this paper?

No

Recommendation?

Major revision is needed (please make suggestions in comments)

Comments to the Author(s)

I have read the Royal Society Open Science manuscript (RSOS-190319) 'Body coloration and mechanisms of colour production in Archelosauria: The case of deirocheline turtles'. I find the paper interesting, and it contains a great deal of important information about color production in archelosaurian reptiles.

I have several general comments:

There is so much detailed information here that the writing and presentation have to be super tight. A few things might help tighten this story up:

Explain the coherent / incoherent scattering & broad band / narrow band reflector ideas more clearly in the discussion.

Use direct/active sentence structure as much as possible to get the sentences meanings across, especially in the discussion.

Describe the sexual dichromatic results in abstract, set it up briefly in intro, and describe in results & discussion a little. This seems to be pretty concrete evidence of sexual dichromatism so there is a story here for this topic.

I also have several specific comments:

Abstract

Line 6-7: 'Increasing knowledge of the functional role of animal coloration stresses...' Awkward sentence. Rephrase.

Line 10: ; and carotenoid/pteridine derivatives contents. Choose one of the words: derivative OR contents.

Line 12: Mention both species: '...we found abundant iridophores in each species, which...'

Line 13: Mention both species: 'abundant dermal collagen fibers in both species which may serve...'

Line 14: Mention both species: 'The colour of yellow-red skin patches in both species results from...'

Line 17: Now describe the results of colour production mechanisms by stripe locations & tell us if it matches predictions based on mating behavior differences between *T. scripta* & *P. concinna*.

Mention sexual dichromatic spectral results briefly after line 17 sentence.

Line 18: 'Our results indicate that archelosaurs share some colour production mechanisms with amphibians and lepidosaurs (mention which mechanisms), but also employ novel mechanisms (mention which mechanisms).'

Page 2: Introduction

First sentence should start off as 'Identifying the evolutionary origins and selective pressures...as is its missing a true subject.

'morphological traits in which developing (instead of composing) subunits evolve for various roles...'

Line 9: clarify: 'Colour-producing elements interact to enhance or reduce their respective contributions to observable colour' instead of 'reduce each other's contributions'

Line 24: insert new paragraph, and start with sentence 'Iridophores contain reflecting platelets of guanine...'

Line 38: 'patterns are common in all major lineages (for a comprehensive...'. Delete 2nd placement of 'turtles'

Throughout: make sure Colour is always spelled colour throughout and NOT color.

Page 3

Line 17: '...it IS possible to examine whether there is a relationship...'

Lines 25-29: 'In *T. scripta* two of three subspecies....All *Pseudemys* turtles...' These two sentences here don't belong in the intro but in Materials and Methods.

Rewrite line 28 'We particularly focus on...' Delete that beginning and instead say something like 'To determine if colour production mechanisms match our predictions based on mating behavior we describe the cellular and ultrastructural composition of those body regions that likely...'

Line 36: 'We discuss our results...' move this sentence somewhere in Materials and Methods...

Line 38: 'Our results represent the first...' delete sentence...not important to be first here. It might help sell your paper so you should say something like this in your introductory letter though...

End last paragraph of introduction by stating your predictions clearly: FLBS and PM will be brightest in *T. scripta*, and PM will be brightest in PC (or something to this effect).

Materials and Methods

2.2 Spectral reflectance measurements

Line 63: 'Measurements were taken in a darkened room at a distance of 5 mm...' I don't understand. What was taken at a distance of 5 mm?

Page 4

2.3. Processing and statistical analysis of reflectance spectra

Line 23-27: 'Redundancy analysis (RDA) was performed on...' Explain why it is necessary to do this first.

Page 5

2.6 Fourier analysis of spatial distribution of dermal collagen arrays

Line 48: End of 2.6 section: explain what an example of output looks like and what it tells us.

2.7. Pigment content analysis

Line 54: '...cleaned mechanically, washed...' How was it cleaned mechanically? Explain...with a brush, etc?

Page 6

Line 15-16: 'A chromatogram of the all carotenoid standards mixture...' awkward phrasing, revise.

Line 31-32: should be '...HPLC system (Agilent series 1290 coupled with...tandem mass spectrometry, Agilent Technologies, Waldbronn, Germany).

Line 46: '...Fig S2a; SRM chromatograms of the individuals...'

Results

Page 7

Line 6-9: 'Summary variables derived from reflectance spectra...' So? Remind us what this means...

Also before next paragraph mentioning ordination plots, describe numerically the sexually dichromatic differences in spectra here.

Page 8.

3.3 Reflecting paltelets of iridophores

Line 42-42: Unfortunately, measurements of iridophores from the FLBS...' Move this sentence to M&M?

Page 9.

Line 8: electromicrograph is misspelled.

Line 9: end sentence like this: '...with respect to wavelengths of light. In other words, the distances between collagen fibres...'

Lines 37-42: 'Predicted normalized reflectivity of collagen fibre arrays of P.c. and T.s....' OK, but what is all of this supposed to tell us? Explain.

Line 61-62: '...is to reflect infrared radiation and only in some instances collagen fibres have been co-opted to function...'

Page 10

Line 8: Specify colors of the regions described in this sentence to make the point that yellow

regions lack ketocarotenoids: 'Other regions (yellow CBC, yellow FLBS of *P. conccina*, red FLBS of *T. scripta*, yellow PM of *P. concinna*, etc) ...'

Line 13: Indent, new paragraph here: 'Examples of SRM chromatograms of pteridine...'

Line 17: '...were found together only in the red PM of *T.s. elegans*'.

Line 21: Indent for new paragraph: 'The red PM of *T.s.elegans* is unique among the examined...'

Discussion

I am not sure about the way you start the first sentence. The claim itself is contentious because many turtles ARE known for their color.

Line 40: ', studying turtle coloration and colour producing mechanisms is crucial'....

Line 43-45: long sentence, but still statement of contrasting courtship behavior should occur earlier in the sentence.

'...and the chemical nature of pigments that produce skin colour in two freshwater turtles with contrasting courtship behavior, *Pseudemys concinna* and *Trachemys scripta*.

Line 51: 'Turtles employ colour producing mechanisms found in amphibians...'

Line 53: '...and crociles [81]...' seems to contradict what was said in previous sentence (extant archosaurs (crocodiles and birds)...

Omit entire sentence placed in line 55-58: Carotenoids are involved in...feathers of birds, etc.

Next sentence should be 'In addition, our pigment analyses suggest that both pteridine derivatives and carotenoids...'

Line 62: '...penetrating into the body of turtles, but in some cases collagen fibre arrays may be capable of reflecting...'

Page 11

4.1. The coloration of Deirocheline turtles

Line 12: 'However, we did not find support for the spectral shapes reported by Wang et al [87]; in fact...'

Is this sentence important enough to include / present in this paper? We should expect there to be geographical / environmental variation in spectra of these stripes...the mechanistic causes to these spectra are what's really important.

Line 39, after info from the 2nd paragraph ('...examined turtles (Fig 7 b,d): write about intraspecific differences in spectra, e.g. *T. scripta scripta* vs *T. scripta elegans*. Then write about sexual dichromatism in spectra?

4th paragraph, Lines 55-62: better explain how reflecting platelet organization causes tissue to act as either a narrow band or broad band reflector.

Page 12

1st paragraph, 1st sentence: rewrite 1st sentence (Unlike all yellow regions of both species,...) as a direct-active sentence to make it more powerful.

Line 9: 'Previous reports indicated that...you cite only 1 previous report so this statement should be singular

4th paragraph

Line 44-Line 61, etc. this entire paragraph needs better logical flow.

Line 46: 'The presence of red? Yellow? Xanthophores together with iridophores...'

Line 52-56: Sentence: This should make *P. concinna* rather conspicuous when viewed... This long sentence also needs more clarity, better logical flow. Also the way this sentence is written seems to contradict what you expect for *P.c.*, no? Because *P.c.* males observe mates from above, should there be xanthophores and iridophores be distributed dorsally (in the PM) and ventrally (in the CBC)?

Page 13

Can you better explain iridophores as coherent or incoherent scattering mechanisms in this 4.1. section?

Section 4.2

Line 39 the collagen fibres form subepidermal collagenous lamella overlying the dermal chrotophores...Do you mean to say this? Wouldn't this obscure spectral reflectance?

2nd paragraph, line 55: '...but color production by coherent scattering on nanostructured

collagen fiber arrays has been previously reported in mammals (17). The 1st & second parts of this sentence don't seem logically rrelated. 1st part: stating achromatic nature of the fiber, 2nd part refers to color production by coherent scattering in mammals.

Page 14

Line 1: '...do not conform to the conditions of coherent scattering or predictions do not...' These need to be better explained then.

Section 4.3

Line 12-14: 'Most vertebrates do produce various colors by...' I don't understand what you're trying o say here.

Much of the rest of this section seems too speculative, and perhaps needs tightening if kept in future drafts.

4.4. Future directions. Is this heading part of RSOS format? If not, is it necessary and appropriate? Perhaps write a conclusion section instead?

Tables

Table 1. Couldn't table 1 be arranged / expressed to include sex differences and highlight significant differences with an asterisk? Of course the p-values would need to be Bonferoni correction to account for multiple comparisons.

Table 3. Spelling of word 'length' needs correction

Figures

Figure 2: seems to show sexual dichromatism in UV and Longwavelengths. Why not present this statistically in results and add to abstract, intro and discussion?

Review form: Reviewer 2

Is the manuscript scientifically sound in its present form?

Yes

Are the interpretations and conclusions justified by the results?

Yes

Is the language acceptable?

Yes

Is it clear how to access all supporting data?

Yes

Do you have any ethical concerns with this paper?

No

Have you any concerns about statistical analyses in this paper?

I do not feel qualified to assess the statistics

Recommendation?

Accept with minor revision (please list in comments)

Comments to the Author(s)

The authors undertook a thorough examination of the pigments and structures that underlie the integumentary pigmentation of two species of freshwater turtles, and discuss their findings in light of the species' different courtship behaviors and their phylogenetic relationships to other extant groups of vertebrates. The treatment is exhaustive, except perhaps for the lack of pigment

quantifications, and for the most part its findings are well supported.

I struggled a bit with the first part of the introduction and its very broad brush. The authors could probably have referenced the “interacting processes” of “functional integration of colour ornaments and morphological modularity of colour-producing elements” without actually naming them. To present them as “intriguing” relationships is a bit deceitful. Nothing is added by the “Future directions” section of the manuscript and it should probably be removed.

Throughout the manuscript, please refer to carotenoids and pteridines (or pterins) as such, not as “derivatives”. “Colour-producing” should be hyphenated throughout.

I have many other specific comments below.

Specific comments:

Page 2, line 21: five “types”, rather than “categories”.

Page 2, line 25: “mainly guanine” (hypoxanthine is also often present).

Page 2, line 57: “to this day” (not “date”).

Page 3, lines 3-5: last part of the sentence is not needed: “however sexual dichromatism is limited to particular regions of the body.”

Page 3, line 29: “ultrastructural make-up”, not “composition”.

Page 3, line 31: “chemical nature of pigments”, not “composition”.

Page 3, line 34: “the make-up (or structure) of the skin”, not “composition”.

Page 3, last sentence: did you use a probe holder to hold the probe? If not, then how did you achieve a fixed distance of 5 mm? If you used a probe holder, was 5 mm the distance of the probe to the skin, or to the probe holder?

Page 4, line 59: remove “similarly to”.

Page 6, line 1: the list of carotenoid standards you acquired is not exhaustive, though probably includes the most important ones.

Page 6, lines 26-27: reptiles (mainly lizards) are far more likely to use drospterins to produce red colours using pteridines than erythropterin. I am surprised that this was missed by the authors.

There are several examples of lizards with drospterins in their integument. To mention a few recent ones: Weiss, S.L., K. Foerster and J. Hudon. 2012. Pteridine, not carotenoid, pigments underlie the female-specific orange ornament of striped plateau lizard (*Sceloporus virgatus*). *Comp. Biochem. Physiol. B* 161: 117-123. Cuervo, J.J., J. Belliure, and J.J. Negro. 2016. Coloration reflects skin pterin concentration in a red-tailed lizard. *Comp. Biochem. Physiol. B* 193: 17-24.

Page 6, lines 42-43: check spelling: xanthopterin, leucopterin.

Page 6, last line: the peaks are not the only features of interest. The valleys point to absorbing compounds (pigments).

Page 7, line 51: concinna

Page 9, lines 33-34: “... are somewhat elevated compared to the radial means ...”, rather than “... are in comparison with ... relatively elevated”.

Page 10, lines 15-18: most of these pteridines are colourless, so are not likely to explain the colour unless they contributed structurally.

Page 10, line 40: “... studying turtle coloration and colour-producing mechanisms is crucial ...”

Page 10, line 56, lines 58-59: carotenoids are also present in the irises of species from many orders of birds (Anseriformes, Galliformes, Podicipediformes, Ciconiiformes, etc...). Ref: Oehme, V.H. (1969) Vergleichende Untersuchungen über die Färbung der Vogeliris. *Biol. Zentralblatt*. 88: 3-35. Both carotenoids and pteridines are present in the iris of some species. Ref: Oliphant, L.W. (1981) Crystalline pteridines in the stromal pigment cells of the iris of the Great Horned Owl. *Cell Tissue Res.*, 217:387-395; Oliphant, L.W. (1987) Pteridines and purines as major pigments of the avian iris. *Pigment Cell Res.*, 1:129-131; Oliphant, L.W. (1988) Cytology and pigments of non-melanophore chro-matophores in the avian iris. In: *Advances in Pigment Cell Research*. J.T. Bagnara, ed. Alan R. Liss, Inc., New York, pp. 65-82.

Page 12, line 1: though they may not produce colour through thin-film interference, iridophores most certainly “will” (not “may”) increase the brightness of colour regions and overall conspicuousness to observers, thus contribute to colour. The change should be made to the Abstract as well.

Page 12, lines 8-11: you sound like you agree with these results. You might want to change to "In the yellow chin of *C. picta* only apocarotenoids were postulated ..." or something of that sort.

Page 12, lines 26-27: your finding is not exactly the opposite of the previous sentence. Please state your finding instead.

Page 13, line 5: again many of these pteridines are colourless, so probably not all that relevant. Instead emphasize differences in the abundance of the colourful ones.

Page 13, line 50: there are no iridophores in the plumages of birds.

Page 14, line 12: "Most vertebrates produce ..." or "Most of the vertebrates produce .."

Page 14, line 13: true of integumentary coloration, not of bird feathers. Bird feathers can harbour many different types of pigments.

Page 14, line 25: "poses"?! Not the correct word.

Page 14, line 33: thus presumably the ancestral state (plesiomorphic).

Figure 6. It is pretty hard to discern some of the highlighted elements, like the iridophores in 6d. I wish the smaller images were made slightly larger.

Figure 9. I am not 100% sure that the vesicles in 9c are oil droplets. They do appear to show some internal structure. Can you see a membrane? Oil droplet should be membrane-free. Pterinosomes don't always show lamellae.

Figure 15. I don't see any grey rectangle on the graphs.

Page 41, line 24: A/y0 "represents" or "captures", not "determine"

Page 43, line 17: italicize *T. scripta*.

From page 43, line 56 to page 44, line 11: "pose" is not the right word. "present", "display" or "feature" might work.

Decision letter (RSOS-190319.R0)

26-Mar-2019

Dear Mr Brejcha,

The editors assigned to your paper ("Body coloration and mechanisms of colour production in Archelosauria: The case of deirocheline turtles") have now received comments from reviewers. We would like you to revise your paper in accordance with the referee and Associate Editor suggestions which can be found below (not including confidential reports to the Editor). Please note this decision does not guarantee eventual acceptance.

Please submit a copy of your revised paper before 18-Apr-2019. Please note that the revision deadline will expire at 00.00am on this date. If we do not hear from you within this time then it will be assumed that the paper has been withdrawn. In exceptional circumstances, extensions may be possible if agreed with the Editorial Office in advance. We do not allow multiple rounds of revision so we urge you to make every effort to fully address all of the comments at this stage. If deemed necessary by the Editors, your manuscript will be sent back to one or more of the original reviewers for assessment. If the original reviewers are not available, we may invite new reviewers.

When submitting your revised manuscript, you must respond to the comments made by the

referees and upload a file "Response to Referees" in "Section 6 - File Upload". Please use this to document how you have responded to the comments, and the adjustments you have made. In order to expedite the processing of the revised manuscript, please be as specific as possible in your response.

- Data accessibility

If you wish to submit your supporting data or code to Dryad (<http://datadryad.org/>), or modify your current submission to dryad, please use the following link:
<http://datadryad.org/submit?journalID=RSOS&manu=RSOS-190319>

- Competing interests

- Authors' contributions

- Acknowledgements

- Funding statement

on behalf of Dr Kristina Sefc (Associate Editor) and Professor Kevin Padian (Subject Editor)
 openscience@royalsociety.org

Associate Editor's comments (Dr Kristina Sefc):

The manuscript has been seen by two reviewers, who are impressed by the wealth of novel information which is provided by this study. Both of them also notice the difficulty of packaging so much information into a digestible format, and offer numerous suggestions to improve the presentation of the data. I'd like to ask the authors to follow their advice and am looking forward to a revision of the manuscript.

Comments to Author:

Reviewers' Comments to Author:

Reviewer: 1

Comments to the Author(s)

I have read the Royal Society Open Science manuscript (RSOS-190319) 'Body coloration and mechanisms of colour production in Archelosauria: The case of deirocheline turtles'. I find the paper interesting, and it contains a great deal of important information about color production in archelosaurian reptiles.

I have several general comments:

There is so much detailed information here that the writing and presentation have to be super tight. A few things might help tighten this story up:

Explain the coherent / incoherent scattering & broad band / narrow band reflector ideas more clearly in the discussion.

Use direct/active sentence structure as much as possible to get the sentences meanings across, especially in the discussion.

Describe the sexual dichromatic results in abstract, set it up briefly in intro, and describe in results & discussion a little. This seems to be pretty concrete evidence of sexual dichromatism so there is a story here for this topic.

I also have several specific comments:

Abstract

Line 6-7: 'Increasing knowledge of the functional role of animal coloration stresses...' Awkward sentence. Rephrase.

Line 10: ; and carotenoid/pteridine derivatives contents. Choose one of the words: derivative OR contents.

Line 12: Mention both species: '...we found abundant iridophores in each species, which...'

Line13: Mention both species: 'abundant dermal collagen fibers in both species which may serve...'

Line 14: Mention both species: 'The colour of yellow-red skin patches in both species results from...'

Line 17: Now describe the results of colour production mechanisms by stripe locations & tell us if it matches predictions based on mating behavior differences between *T. scripta* & *P. concinna*. Mention sexual dichromatic spectral results briefly after line 17 sentence.

Line 18: 'Our results indicate that archelosaurs share some colour production mechanisms with amphibians and lepidosaurs (mention which mechanisms), but also employ novel mechanisms (mention which mechanisms).

Page 2: Introduction

First sentence should start off as 'Identifying the evolutionary origins and selective pressures...as is its missing a true subject.

'morphological traits in which developing (instead of composing) subunits evolve for various roles...'

Line 9: clarify: 'Colour-producing elements interact to enhance or reduce their respective contributions to observable colour' instead of 'reduce each other's contributions'

Line 24: insert new paragraph, and start with sentence 'Iridophores contain reflecting platelets of guanine...'

Line 38: 'patterns are common in all major lineages (for a comprehensive....' Delete 2nd placement of 'turtles'

Throughout: make sure Colour is always spelled colour throughout and NOT color.

Page 3

Line 17: '...it IS possible to examine whether there is a relationship...'

Lines 25-29: 'In *T. scripta* two of three subspecies....All *Pseudemys* turtles...' These two sentences here don't belong in the intro but in Materials and Methods.

Rewrite line 28 'We particularly focus on...' Delete that beginning and instead say something like 'To determine if colour production mechanisms match our predictions based on mating behavior we describe the cellular and ultrastructural composition of those body regions that likely...'

Line 36: 'We discuss our results...' move this sentence somewhere in Materials and Methods...

Line 38: 'Our results represent the first...' delete sentence...not important to be first here. It might help sell your paper so you should say something like this in your introductory letter though...

End last paragraph of introduction by stating your predictions clearly: FLBS and PM will be brightest in *T. scripta*, and PM will be brightest in PC (or something to this effect).

Materials and Methods

2.2 Spectral reflectance measurements

Line 63: 'Measurements were taken in a darkened room at a distance of 5 mm...' I don't understand. What was taken at a distance of 5 mm?

Page 4

2.3. Processing and statistical analysis of reflectance spectra

Line 23-27: 'Redundancy analysis (RDA) was performed on...' Explain why it is necessary to do this first.

Page 5

2.6 Fourier analysis of spatial distribution of dermal collagen arrays

Line 48: End of 2.6 section: explain what an example of output looks like and what it tells us.

2.7. Pigment content analysis

Line 54: ', cleaned mechanically, washed...' How was it cleaned mechanically? Explain...with a brush, etc?

Page 6

Line 15-16: 'A chromatogram of the all carotenoid standards mixture...' awkward phrasing, revise.

Line 31-32: should be '...HPLC system (Agilent series 1290 coupled with...tandem mass spectrometry, Agilent Technologies, Waldbronn, Germany).

Line 46: '...Fig S2a; SRM chromatograms of the individuals...'

Results

Page 7

Line 6-9: 'Summary variables derived from reflectance spectra...' So? Remind us what this means...

Also before next paragraph mentioning ordination plots, describe numerically the sexually dichromatic differences in spectra here.

Page 8.

3.3 Reflecting paltelets of iridophores

Line 42-42: Unfortunately, measurements of iridophores from the FLBS...' Move this sentence to M&M?

Page 9.

Line 8: electromicrograph is misspelled.

Line 9: end sentence like this: '...with respect to wavelengths of light. In other words, the distances between collagen fibres...'

Lines 37-42: 'Predicted normalized reflectivity of collagen fibre arrays of P.c. and T.s....' OK, but what is all of this supposed to tell us? Explain.

Line 61-62: '...is to reflect infrared radiation and only in some instances collagen fibres have been co-opted to function...'

Page 10

Line 8: Specify colors of the regions described in this sentence to make the point that yellow regions lack ketocarotenoids: 'Other regions (yellow CBC, yellow FLBS of *P. concinna*, red FLBS of *T. scripta*, yellow PM of *P. concinna*, etc) ...'

Line 13: Indent, new paragraph here: 'Examples of SRM chromatograms of pteridine...'

Line 17: '...were found together only in the red PM of *T.s. elegans*'.

Line 21: Indent for new paragraph: 'The red PM of *T.s.elegans* is unique among the examined...'

Discussion

I am not sure about the way you start the first sentence. The claim itself is contentious because many turtles ARE known for their color.

Line 40: '...studying turtle coloration and colour producing mechanisms is crucial'....

Line 43-45: long sentence, but still statement of contrasting courtship behavior should occur earlier in the sentence.

'...and the chemical nature of pigments that produce skin colour in two freshwater turtles with contrasting courtship behavior, *Pseudemys concinna* and *Trachemys scripta*.

Line 51: 'Turtles employ colour producing mechanisms found in amphibians...'

Line 53: '...and crociles [81]...' seems to contradict what was said in previous sentence (extant archosaurs (crocodiles and birds)...

Omit entire sentence placed in line 55-58: Carotenoids are involved in....feathers of birds, etc.

Next sentence should be 'In addition, our pigment analyses suggest that both pteridine derivatives and carotenoids...'

Line 62: '...penetrating into the body of turtles, but in some cases collagen fibre arrays may be capable of reflecting...'

Page 11

4.1. The coloration of Deirocheline turtles

Line 12: 'However, we did not find support for the spectral shapes reported by Wang et al [87]; in fact...'

Is this sentence important enough to include / present in this paper? We should expect there to be geographical / environmental variation in spectra of these stripes...the mechanistic causes to these spectra are what's really important.

Line 39, after info from the 2nd paragraph ('...examined turtles (Fig 7 b,d).): write about intraspecific differences in spectra, e.g. *T. scripta scripta* vs *T. scripta elegans*. Then write about sexual dichromatism in spectra?

4th paragraph, Lines 55-62: better explain how reflecting platelet organization causes tissue to act as either a narrow band or broad band reflector.

Page 12

1st paragraph, 1st sentence: rewrite 1st sentence (Unlike all yellow regions of both species,...) as a direct-active sentence to make it more powerful.

Line 9: 'Previous reports indicated that...you cite only 1 previous report so this statement should be singular

4th paragraph

Line 44-Line 61, etc. this entire paragraph needs better logical flow.

Line 46: 'The presence of red? Yellow? Xanthophores together with iridophores...'

Line 52-56: Sentence: This should make *P. concinna* rather conspicuous when viewed... This long sentence also needs more clarity, better logical flow. Also the way this sentence is written seems to contradict what you expect for *P.c.*, no? Because *P.c.* males observe mates from above, should there be xanthophores and iridophores be distributed dorsally (in the PM) and ventrally (in the CBC)?

Page 13

Can you better explain iridophores as coherent or incoherent scattering mechanisms in this 4.1. section?

Section 4.2

Line 39 the collagen fibres form subepidermal collagenous lamella overlying the dermal chromatophores...Do you mean to say this? Wouldn't this obscure spectral reflectance?

2nd paragraph, line 55: '...but color production by coherent scattering on nanostructured collagen fiber arrays has been previously reported in mammals (17). The 1st & second parts of this sentence don't seem logically related. 1st part: stating achromatic nature of the fiber, 2nd part refers to color production by coherent scattering in mammals.

Page 14

Line 1: '...do not conform to the conditions of coherent scattering or predictions do not...' These need to be better explained then.

Section 4.3

Line 12-14: 'Most vertebrates do produce various colors by...' I don't understand what you're trying to say here.

Much of the rest of this section seems too speculative, and perhaps needs tightening if kept in future drafts.

4.4. Future directions. Is this heading part of RSOS format? If not, is it necessary and appropriate? Perhaps write a conclusion section instead?

Tables

Table 1. Couldn't table 1 be arranged / expressed to include sex differences and highlight significant differences with an asterisk? Of course the p-values would need to be Bonferroni correction to account for multiple comparisons.

Table 3. Spelling of word 'length' needs correction

Figures

Figure 2: seems to show sexual dichromatism in UV and Longwavelengths. Why not present this statistically in results and add to abstract, intro and discussion?

Reviewer: 2

Comments to the Author(s)

The authors undertook a thorough examination of the pigments and structures that underlie the integumentary pigmentation of two species of freshwater turtles, and discuss their findings in light of the species' different courtship behaviors and their phylogenetic relationships to other

extant groups of vertebrates. The treatment is exhaustive, except perhaps for the lack of pigment quantifications, and for the most part its findings are well supported.

I struggled a bit with the first part of the introduction and its very broad brush. The authors could probably have referenced the “interacting processes” of “functional integration of colour ornaments and morphological modularity of colour-producing elements” without actually naming them. To present them as “intriguing” relationships is a bit deceitful. Nothing is added by the “Future directions” section of the manuscript and it should probably be removed.

Throughout the manuscript, please refer to carotenoids and pteridines (or pterins) as such, not as “derivatives”. “Colour-producing” should be hyphenated throughout.

I have many other specific comments below.

Specific comments:

Page 2, line 21: five “types”, rather than “categories”.

Page 2, line 25: “mainly guanine” (hypoxanthine is also often present).

Page 2, line 57: “to this day” (not “date”).

Page 3, lines 3-5: last part of the sentence is not needed: “however sexual dichromatism is limited to particular regions of the body.”

Page 3, line 29: “ultrastructural make-up”, not “composition”.

Page 3, line 31: “chemical nature of pigments”, not “composition”.

Page 3, line 34: “the make-up (or structure) of the skin”, not “composition”.

Page 3, last sentence: did you use a probe holder to hold the probe? If not, then how did you achieve a fixed distance of 5 mm? If you used a probe holder, was 5 mm the distance of the probe to the skin, or to the probe holder?

Page 4, line 59: remove “similarly to”.

Page 6, line 1: the list of carotenoid standards you acquired is not exhaustive, though probably includes the most important ones.

Page 6, lines 26-27: reptiles (mainly lizards) are far more likely to use drospterins to produce red colours using pteridines than erythropterin. I am surprised that this was missed by the authors.

There are several examples of lizards with drospterins in their integument. To mention a few recent ones: Weiss, S.L., K. Foerster and J. Hudon. 2012. Pteridine, not carotenoid, pigments underlie the female-specific orange ornament of striped plateau lizard (*Sceloporus virgatus*).

Comp. Biochem. Physiol. B 161: 117-123. Cuervo, J.J., J. Belliure, and J.J. Negro. 2016. Coloration reflects skin pterin concentration in a red-tailed lizard. Comp. Biochem. Physiol B 193: 17-24.

Page 6, lines 42-43: check spelling: xanthopterin, leucopterin.

Page 6, last line: the peaks are not the only features of interest. The valleys point to absorbing compounds (pigments).

Page 7, line 51: concinna

Page 9, lines 33-34: “... are somewhat elevated compared to the radial means ...”, rather than “... are in comparison with ... relatively elevated”.

Page 10, lines 15-18: most of these pteridines are colourless, so are not likely to explain the colour unless they contributed structurally.

Page 10, line 40: “... studying turtle coloration and colour-producing mechanisms is crucial ...”

Page 10, line 56, lines 58-59: carotenoids are also present in the irises of species from many orders of birds (Anseriformes, Galliformes, Podicipediformes, Ciconiiformes, etc...). Ref: Oehme, V.H.

(1969) Vergleichende Untersuchungen über die Färbung der Vogeliris. Biol. Zentralblatt. 88: 3-35.

Both carotenoids and pteridines are present in the iris of some species. Ref: Oliphant, L.W. (1981)

Crystalline pteridines in the stromal pigment cells of the iris of the Great Horned Owl. Cell Tissue Res., 217:387-395; Oliphant, L.W. (1987) Pteridines and purines as major pigments of the avian iris. Pigment Cell Res., 1:129-131; Oliphant, L.W. (1988) Cytology and pigments of non-

melanophore chro- matophores in the avian iris. In: Advances in Pigment Cell Research. J.T. Bagnara, ed. Alan R. Liss, Inc., New York, pp. 65-82.

Page 12, line 1: though they may not produce colour through thin-film interference, iridophores

most certainly “will” (not “may”) increase the brightness of colour regions and overall

conspicuousness to observers, thus contribute to colour. The change should be made to the Abstract as well.

Page 12, lines 8-11: you sound like you agree with these results. You might want to change to "In the yellow chin of *C. picta* only apocarotenoids were postulated ..." or something of that sort.

Page 12, lines 26-27: your finding is not exactly the opposite of the previous sentence. Please state your finding instead.

Page 13, line 5: again many of these pteridines are colourless, so probably not all that relevant. Instead emphasize differences in the abundance of the colourful ones.

Page 13, line 50: there are no iridophores in the plumages of birds.

Page 14, line 12: "Most vertebrates produce ..." or "Most of the vertebrates produce .."

Page 14, line 13: true of integumentary coloration, not of bird feathers. Bird feathers can harbour many different types of pigments.

Page 14, line 25: "poses"?! Not the correct word.

Page 14, line 33: thus presumably the ancestral state (plesiomorphic).

Figure 6. It is pretty hard to discern some of the highlighted elements, like the iridophores in 6d. I wish the smaller images were made slightly larger.

Figure 9. I am not 100% sure that the vesicles in 9c are oil droplets. They do appear to show some internal structure. Can you see a membrane? Oil droplet should be membrane-free. Pterinosomes don't always show lamellae.

Figure 15. I don't see any grey rectangle on the graphs.

Page 41, line 24: A/y0 "represents" or "captures", not "determine"

Page 43, line 17: italicize *T. scripta*.

From page 43, line 56 to page 44, line 11: "pose" is not the right word. "present", "display" or "feature" might work.

Author's Response to Decision Letter for (RSOS-190319.R0)

See Appendix A.

RSOS-190319.R1 (Revision)

Review form: Reviewer 1

Is the manuscript scientifically sound in its present form?

Yes

Are the interpretations and conclusions justified by the results?

Yes

Is the language acceptable?

Yes

Is it clear how to access all supporting data?

Yes

Do you have any ethical concerns with this paper?

No

Have you any concerns about statistical analyses in this paper?

No

Recommendation?

Accept with minor revision (please list in comments)

Comments to the Author(s)

I have read the Royal Society Open Science manuscript (RSOS-190319 rev 1) 'Body coloration and mechanisms of colour production in Archelosauria: The case of deirocheline turtles'. The authors attended to the previous comments of reviewers and the findings and implications of this interesting work are more understandable. I believe it is almost ready for publication, but I have a few specific suggestions:

Make sure various words are consistently spelled with UK-style spelling rules throughout paper. E.g. Colour (not color), behavior (not behavior), etc. The title still has the 'US' spelling for color.

Abstract

Line 6: should write as 'colour production still remain unknown.'

Line 9: rewrite these few sentences to describe the perspective more logically: We found xanthophores, melanophores, abundant iridophores and dermal collagen fibers in stripes of both species. Moreover, in both species, the xanthophores coloring the yellow-red skin contain carotenoids, pterins, and riboflavin, but the two species differ in the distribution of pigment cell types and pigment diversity. Both *Pseudemys concinna*'s and *T. scripta*'s yellow chin and forelimb stripes contain xanthophores and iridophores, but *P. concinna* and *T. scripta*'s post-orbital region's differ in cell-type distribution. *T. scripta*'s post-orbital stripes contain xanthophores and iridophores, while *T. scripta*'s yellow-red post-orbital / zygomatic regions contain xanthophores only.

Introduction

Line 38: We predict that regions of the skin that males expose to females during courtship should be richer in pigment contents than regions not exposed during courtship. Omit the word 'bright' from these predictions because in tri-stimulus theory brightness means the amount of total reflectance coming off the integument and correlates inversely with increased pigment concentrations.

Materials and Methods

2.2 Spectral reflectance measurements

Line 51: necessary to include & describe the Cumberland slider? Seems to suggest you're going to present research on it, and of course you aren't.

Page 4, line 1: Make sure your abbreviations are formatted consistently throughout. E.g. if using bold-face for them make sure they are all bold face (in later sections, etc). I don't think you need to use bold-face on these abbreviations.

Results

Page 10, Line 29: 1st sentence of 2nd full paragraph: Normalized averaged radial means of the Fourier power spectra... This is a long, run-on sentence and should be broken into 2-3 sentences. Why is the zygomatic region of *P. concinna* abbreviated as YP? Wouldn't it be more intuitive if it were abbreviated 'ZP' (for zygomatic patch)? No other abbreviation has Y in it to mention color, and in general the letters represent anatomical locations instead of color.

Discussion

Page 13 line 6: Is 'wavelets' the correct word?

Line 10: 'As the reflecting platelets of iridophores from the CBC of *T. scripta* are highly ____, they produce...' Are you missing an adjective here?

Line 28: 'All yellow regions of both species contain hydroxylated xanthophylls such as lutein and

zeaxanthin...'

Page 15, line 2: Sentence meaning is unclear. Do you mean 'As mating strategies are better understood?' or 'become better studied'? Or should the sentence read 'Because mating strategies are well-studied in this group, it may be relatively straightforward to test hypotheses about sexual selection'?

4.3 Significance for vertebrate skin colour evolution

Line 65: 'Physiological mechanisms of colour production...'

Line 69: 'Iridophore and xanthophore differentiation has been lost...'

Page 16, 1st full paragraph:

Line 9: How about 'Xanthophores are pigment cells that contain carotenoids and pterins in their vesicles.'?

Line 11: 'Except a rare occurrence in feathers of penguins...'

Tables and Figures

Figure 3. Minor detail, but should the B1-and H1-based spectral variables (i.e. brightness and hue) be written with the same format as the S1 variables? E.g., since S1.blueCBC, shouldn't B1.CBC? H1.CBC?

Figure 10: overlay abbreviations for names of colored regions above the regions on these images (e.g. CBC, DHC, PM, & YP)?

Figure 12: x-axes (wavelength) spelled wrong.

Fig 10 caption: why is zygomatic patch abbreviated YP?

Fig 12. 1st sentence of caption should read: 'Distribution of predicted reflectivity of reflecting platelets of iridophores based on Fourier Analysis. (a)...'

Review form: Reviewer 2

Is the manuscript scientifically sound in its present form?

Yes

Are the interpretations and conclusions justified by the results?

Yes

Is the language acceptable?

Yes

Is it clear how to access all supporting data?

Yes

Do you have any ethical concerns with this paper?

No

Have you any concerns about statistical analyses in this paper?

No

Recommendation?

Accept with minor revision (please list in comments)

Comments to the Author(s)

I previously reviewed this paper. I commend the authors for resampling the skins for evidence of riboflavin and drosopterin, a pteridine often found in lepidosaurians with red tones. I think that it adds perspective to the paper.

I must qualify a comment I made in my past review. I pointed out that many of the pteridines the authors found in the brightly-coloured surfaces of turtles were colourless and might not contribute directly to skin colours we (humans) perceive as yellow or red. This led the authors to revise their manuscript to try to explain the presence of colourless pteridines, which produced a treatment that was not evenly convincing (see specific comment below). I will add another consideration to the mix: since many of these “colourless” pteridines absorb in the UV and turtles can see in the UV (as pointed out in the manuscript: p. 34, line 10), it is probably reasonable to assume that they would produce/alter colours that turtles can see (unlike us).

My main concern this time around is with the way the authors characterize birds in the last section of the manuscript (“Significance for vertebrate skin colour evolution”). It is inaccurate to state that birds lack pigment cells other than melanocytes. Birds may have largely lost pigment cells from their integument as a result of the development of an outer covering of feathers, but they retained most pigment cell types (e.g., xanthophores and iridophores) in their irises (Oliphant et al. 1992 Pigment Cell Res 5:367-371). I note that turtles probably have similar pigment cells in their irises than they do in their integuments, since the color patterns on the face of turtles often continue uninterrupted across the iris. I think that making a distinction between the pigment cells of the integument and those of the iris is inappropriate, since BOTH derive equally from neural crest cells (see review on the origin of cells that form the eye in Williams and Bohnsack 2015 Birth Defects Research 105:87-95). The pigment cells in the irises of birds may no longer have dendritic projections and motile pigment organelles (and thus might be more appropriately called xanthocytes and iridocytes), but that is beside the point.

Specific comments:

Page 23, line 39: I suggest “pigment types” rather than “pigment contents”, as the latter implies quantification (which did not take place).

Page 23, line 42-44: Awkward sentence. Maybe something like: “The chin, forelimb, and postorbital yellow stripes of *Pseudemys concinna* contains xanthophores ...”

Page 24, lines 8 and 9: “colour-producing elements” is sometimes hyphenated, but not always (as here). Please make sure that “colour-producing” is hyphenated throughout.

Page 24, line 55: Remove extra “in”: “playing a role in in colour production”.

Page 25, lines 30, 33: I suggest rich in “pigment types”? or “pigments”?

Page 28, line 41: you are missing micron in: 2.6 μ m

Page 28, line 42 “tert-butyl methyl ether”

Page 29, lines 7-10: Awkward sentence. I suggest something like “Separation of all studied compounds with the exception of drosopterin was achieved through isocratic elution at a flow rate of 0.5 ml/min with a mobile phase of ...”

Page 32 last line: Again “pigment types” would be preferable to “pigment contents”.

Page 33, lines 16 and 18: This is confusing. Some pigments were not detected in any samples. But then *T. s. scripta* contained all types except pterin. I suspect that the latter sample did not contain the pigments not detected in any sample, but this should be made clearer. Please revise the text.

Page 33, lines 22-29: This is largely a repeat of results from the previous two paragraphs, but framed differently. It is hard for me to justify repeating information when the paper is already so long.

Page 33, line 57: You are playing with words here. Pterins and carotenoids actually are involved in the production of yellow-red colours in birds, maybe not in their integument, but in the iris.

Page 35, line 28: “both species contain the hydroxylated xanthophylls ...”.

Page 35, line 35: “whereas the orange neck and leg contained apocarotenoids and ketocarotenoids.”

Page 35, last paragraph: Since turtles can see in the ultraviolet what look like colourless pterins to us may actually be visible (and coloured) to turtles.

Page 36 lines 1-2: I find this suggestion highly speculative and a bit far-fetched.

Page 37, last line. It is untrue that birds do not develop iridophores and xanthophores. They do in the iris.

Page 38, line 13. A period is missing after “[160, 162,163]”.

Page 38, line 14: “contain pigments similar to those in xanthophores”.

Page 38, lines 17-18: carotenoids and pterins do act together to produce colour in the iris of some birds (as pointed out earlier in the paper).

Page 38, line 20: Birds should to be listed here as well, as there are examples of pterins contributing with carotenoids to yellow-red colours, again in the iris.

Page 38, line 22: This is incorrect. Some birds possess xanthophores in their irises (unless you want to be technical and call those xanthocytes).

Figure 9. I am still not 100% sure that the vesicles in 9e (not 9c in my previous review, my mistake) are oil droplets. They do appear to show some internal structure.

Decision letter (RSOS-190319.R1)

28-May-2019

Dear Mr Brejcha:

On behalf of the Editors, I am pleased to inform you that your Manuscript RSOS-190319.R1 entitled "Body coloration and mechanisms of colour production in Archelosauria: The case of deirocheline turtles" has been accepted for publication in Royal Society Open Science subject to minor revision in accordance with the referee suggestions. Please find the referees' comments at the end of this email.

The reviewers and Subject Editor have recommended publication, but also suggest some minor revisions to your manuscript. Therefore, I invite you to respond to the comments and revise your manuscript.

- Ethics statement

- Data accessibility

<http://datadryad.org/submit?journalID=RSOS&manu=RSOS-190319.R1>

- **Competing interests**

- **Authors' contributions**

- **Acknowledgements**

- **Funding statement**

Because the schedule for publication is very tight, it is a condition of publication that you submit the revised version of your manuscript before 06-Jun-2019. Please note that the revision deadline will expire at 00.00am on this date. If you do not think you will be able to meet this date please let me know immediately.

on behalf of Dr Kristina Sefc (Associate Editor) and Kevin Padian (Subject Editor)
openscience@royalsociety.org

Associate Editor Comments to Author (Dr Kristina Sefc):

Dear authors,
Both reviewers agree with the changes that were made to the manuscript and offer specific suggestions to improve the wording and clarity. Reviewer 2 also requests a revision in regard to the interpretation of colorless pteridines and a correction of the information on avian pigment cells. Please follow their suggestions closely. Additionally, please complete the figure caption for Figure S1 (G – J). Do you have any citable source for drosoplerin extraction with DMSO?
Sincerely, Kristina Sefc

Associate Editor: 2
Comments to the Author:
(There are no comments.)

Reviewer comments to Author:

Reviewer: 1

Comments to the Author(s)

I have read the Royal Society Open Science manuscript (RSOS-190319 rev 1) 'Body coloration and mechanisms of colour production in Archelosauria: The case of deirocheline turtles'. The authors attended to the previous comments of reviewers and the findings and implications of this interesting work are more understandable. I believe it is almost ready for publication, but I have a few specific suggestions:

Make sure various words are consistently spelled with UK-style spelling rules throughout paper. E.g. Colour (not color), behavior (not behavior), etc. The title still has the 'US' spelling for color.

Abstract

Line 6: should write as 'colour production still remain unknown.'

Line 9: rewrite these few sentences to describe the perspective more logically: We found xanthophores, melanophores, abundant iridophores and dermal collagen fibers in stripes of both species. Moreover, in both species, the xanthophores coloring the yellow-red skin contain carotenoids, pterins, and riboflavin, but the two species differ in the distribution of pigment cell types and pigment diversity. Both *Pseudemys concinna*'s and *T. scripta*'s yellow chin and forelimb stripes contain xanthophores and iridophores, but *P. concinna* and *T. scripta*'s post-orbital region's differ in cell-type distribution. *T. scripta*'s post-orbital stripes contain xanthophores and iridophores, while *T. scripta*'s yellow-red post-orbital / zygomatic regions contain xanthophores only.

Introduction

Line 38: We predict that regions of the skin that males expose to females during courtship should be richer in pigment contents than regions not exposed during courtship. Omit the word 'bright' from these predictions because in tri-stimulus theory brightness means the amount of total reflectance coming off the integument and correlates inversely with increased pigment concentrations.

Materials and Methods

2.2 Spectral reflectance measurements

Line 51: necessary to include & describe the Cumberland slider? Seems to suggest you're going to present research on it, and of course you aren't.

Page 4, line 1: Make sure your abbreviations are formatted consistently throughout. E.g. if using bold-face for them make sure they are all bold face (in later sections, etc). I don't think you need to use bold-face on these abbreviations.

Results

Page 10, Line 29: 1st sentence of 2nd full paragraph: Normalized averaged radial means of the Fourier power spectra... This is a long, run-on sentence and should be broken into 2-3 sentences. Why is the zygomatic region of *P. concinna* abbreviated as YP? Wouldn't it be more intuitive if it were abbreviated 'ZP' (for zygomatic patch)? No other abbreviation has Y in it to mention color, and in general the letters represent anatomical locations instead of color.

Discussion

Page 13 line 6: Is 'wavelets' the correct word?

Line 10: 'As the reflecting platelets of iridophores from the CBC of *T. scripta* are highly ____, they produce...' Are you missing an adjective here?

Line 28: 'All yellow regions of both species contain hydroxylated xanthophylls such as lutein and zeaxanthin...'

Page 15, line 2: Sentence meaning is unclear. Do you mean 'As mating strategies are better understood?' or 'become better studied' ? Or should the sentence read 'Because mating strategies are well-studied in this group, it may be relatively straightforward to test hypotheses about sexual selection'?

4.3 Significance for vertebrate skin colour evolution

Line 65: 'Physiological mechanisms of colour production...'

Line 69: 'Iridophore and xanthophore differentiation has been lost...'

Page 16, 1st full paragraph:

Line 9: How about 'Xanthophores are pigment cells that contain carotenoids and pterins in their vesicles.'?

Line 11: 'Except a rare occurrence in feathers of penguins...'

Tables and Figures

Figure 3. Minor detail, but should the B1-and H1-based spectral variables (i.e. brightness and hue) be written with the same format as the S1 variables? E.g., since S1.blueCBC, shouldn't B1.CBC? H1.CBC?

Figure 10: overlay abbreviations for names of colored regions above the regions on these images (e.g. CBC, DHC, PM, & YP)?

Figure 12: x-axes (wavelength) spelled wrong.

Fig 10 caption: why is zygomatic patch abbreviated YP?

Fig 12. 1st sentence of caption should read: 'Distribution of predicted reflectivity of reflecting platelets of iridophores based on Fourier Analysis. (a)...'

Reviewer: 2

Comments to the Author(s)

I previously reviewed this paper. I commend the authors for resampling the skins for evidence of riboflavin and drosopterin, a pteridine often found in lepidosaurians with red tones. I think that it adds perspective to the paper.

I must qualify a comment I made in my past review. I pointed out that many of the pteridines the authors found in the brightly-coloured surfaces of turtles were colourless and might not contribute directly to skin colours we (humans) perceive as yellow or red. This led the authors to revise their manuscript to try to explain the presence of colourless pteridines, which produced a treatment that was not evenly convincing (see specific comment below). I will add another consideration to the mix: since many of these "colourless" pteridines absorb in the UV and turtles can see in the UV (as pointed out in the manuscript: p. 34, line 10), it is probably reasonable to assume that they would produce/alter colours that turtles can see (unlike us).

My main concern this time around is with the way the authors characterize birds in the last section of the manuscript ("Significance for vertebrate skin colour evolution"). It is inaccurate to state that birds lack pigment cells other than melanocytes. Birds may have largely lost pigment cells from their integument as a result of the development of an outer covering of feathers, but they retained most pigment cell types (e.g., xanthophores and iridophores) in their irises (Oliphant et al. 1992 *Pigment Cell Res* 5:367-371). I note that turtles probably have similar pigment cells in their irises than they do in their integuments, since the color patterns on the face of turtles often continue uninterrupted across the iris. I think that making a distinction between the pigment cells of the integument and those of the iris is inappropriate, since BOTH derive equally from neural crest cells (see review on the origin of cells that form the eye in Williams and Bohnsack 2015 *Birth Defects Research* 105:87-95). The pigment cells in the irises of birds may no longer have dendritic projections and motile pigment organelles (and thus might be more appropriately called xanthocytes and iridocytes), but that is beside the point.

Specific comments:

Page 23, line 39: I suggest "pigment types" rather than "pigment contents", as the latter implies quantification (which did not take place).

Page 23, line 42-44: Awkward sentence. Maybe something like: "The chin, forelimb, and postorbital yellow stripes of *Pseudemys concinna* contains xanthophores ..."

Page 24, lines 8 and 9: "colour-producing elements" is sometimes hyphenated, but not always (as here). Please make sure that "colour-producing" is hyphenated throughout.

Page 24, line 55: Remove extra "in": "playing a role in colour production".

Page 25, lines 30, 33: I suggest rich in "pigment types"? or "pigments"?

Page 28, line 41: you are missing micron in: 2.6 μ m

Page 28, line 42 "tert-butyl methyl ether"

Page 29, lines 7-10: Awkward sentence. I suggest something like "Separation of all studied compounds with the exception of drosoplerin was achieved through isocratic elution at a flow rate of 0.5 ml/min with a mobile phase of ..."

Page 32 last line: Again "pigment types" would be preferable to "pigment contents".

Page 33, lines 16 and 18: This is confusing. Some pigments were not detected in any samples. But then *T. s. scripta* contained all types except pterin. I suspect that the latter sample did not contain the pigments not detected in any sample, but this should be made clearer. Please revise the text.

Page 33, lines 22-29: This is largely a repeat of results from the previous two paragraphs, but framed differently. It is hard for me to justify repeating information when the paper is already so long.

Page 33, line 57: You are playing with words here. Pterins and carotenoids actually are involved in the production of yellow-red colours in birds, maybe not in their integument, but in the iris.

Page 35, line 28: "both species contain the hydroxylated xanthophylls ...".

Page 35, line 35: "whereas the orange neck and leg contained apocarotenoids and ketocarotenoids."

Page 35, last paragraph: Since turtles can see in the ultraviolet what look like colourless pterins to us may actually be visible (and coloured) to turtles.

Page 36 lines 1-2: I find this suggestion highly speculative and a bit far-fetched.

Page 37, last line. It is untrue that birds do not develop iridophores and xanthophores. They do in the iris.

Page 38, line 13. A period is missing after "[160, 162,163]".

Page 38, line 14: "contain pigments similar to those in xanthophores".

Page 38, lines 17-18: carotenoids and pterins do act together to produce colour in the iris of some birds (as pointed out earlier in the paper).

Page 38, line 20: Birds should to be listed here as well, as there are examples of pterins contributing with carotenoids to yellow-red colours, again in the iris.

Page 38, line 22: This is incorrect. Some birds possess xanthophores in their irises (unless you want to be technical and call those xanthocytes).

Figure 9. I am still not 100% sure that the vesicles in 9e (not 9c in my previous review, my mistake) are oil droplets. They do appear to show some internal structure.

Author's Response to Decision Letter for (RSOS-190319.R1)

See Appendix B.

Decision letter (RSOS-190319.R2)

28-Jun-2019

Dear Mr Brejcha,

I am pleased to inform you that your manuscript entitled "Body coloration and mechanisms of colour production in Archelosauria: The case of deirocheline turtles" is now accepted for publication in Royal Society Open Science.

Kind regards,

on behalf of Dr Kristina Sefc (Associate Editor) and Kevin Padian (Subject Editor)
openscience@royalsociety.org

CHARLES UNIVERSITY
Faculty of science

Appendix A

Prague, 25th April 2019

To: Editorial board of *Royal Society Open Science*

Dear Prof. Padian, Dr. Sefc, and Reviewers

Please find attached the revised manuscript entitled “Body coloration and mechanisms of colour production in Archelosauria: The case of deirocheline turtles”. We have carefully followed editor’s and reviewers’ suggestions that helped us to improve the manuscript. Our response to each point is presented below in italics.

We have made suggested changes by reviewers and answered all the questions. We have amended manuscript by results on sexual dichromatism in turtles. We have also analysed our samples for two additional pteridine derivatives, drosopterin and riboflavin.

We hope you will find this revised version of the manuscript suitable for publication in Royal Society Open Science.

On behalf of authors,
Yours faithfully,

Jindřich Brejcha

Address:

Dpt. Philosophy and History of Science, Faculty of Science, Charles University, Viničná 7, Prague 2, 128 00, Czech Republic

Phone:

+420 606 381 557

e-mail:

brejcha@natur.cuni.cz

Responses to review

(Author's responses are written in italics)

Associate Editor's comments (Dr Kristina Sefc):

The manuscript has been seen by two reviewers, who are impressed by the wealth of novel information which is provided by this study. Both of them also notice the difficulty of packaging so much information into a digestible format, and offer numerous suggestions to improve the presentation of the data. I'd like to ask the authors to follow their advice and am looking forward to a revision of the manuscript.

Reviewers' Comments to Author:

Reviewer: 1

I have read the Royal Society Open Science manuscript (RSOS-190319) 'Body coloration and mechanisms of colour production in Archelosauria: The case of deirocheline turtles. I find the paper interesting, and it contains a great deal of important information about colour production in archelosaurian reptiles. I have several **general comments**:

There is so much detailed information here that the writing and presentation have to be super tight. A few things might help tighten this story up:

We are impressed by the great effort devoted by Reviewer 1 to our manuscript. Thank you very much for all the comments and suggestions.

Explain the coherent / incoherent scattering & broad band / narrow band reflector ideas more clearly in the discussion.

We explain the coherent/incoherent scattering and broadband/narrowband reflectors in the 5th paragraph on reflecting platelets of iridophores in section 4.1 now.

Use direct/active sentence structure as much as possible to get the sentences meanings across, especially in the discussion.

Revised

Describe the sexual dichromatic results in abstract, set it up briefly in intro, and describe in results & discussion a little. This seems to be pretty concrete evidence of sexual dichromatism so there is a story here for this topic.

Done

Address:

Dpt. Philosophy and History of Science, Faculty of Science, Charles University, Viničná 7, Prague 2, 128 00, Czech Republic

Phone:

+420 606 381 557

e-mail:

brejcha@natur.cuni.cz

I also have several **specific comments**:

Abstract

Line 6-7: 'Increasing knowledge of the functional role of animal coloration stresses...' Awkward sentence. Rephrase.

We have replaced "Increasing knowledge of the functional role of animal coloration stresses the need to study the proximate causes of colour production." by "While many studies address the functions of animal coloration, the mechanisms of colour production remain still unknown in most taxa."

Line 10: ; and carotenoid/pteridine derivatives contents. Choose one of the words: derivative OR contents.

Revised.

Line 12: Mention both species: '...we found abundant iridophores in each species, which...'

Revised as suggested.

Line 13: Mention both species: 'abundant dermal collagen fibers in both species which may serve...'

Revised as suggested.

Line 14: Mention both species: 'The colour of yellow-red skin patches in both species results from...'

Revised as suggested.

Line 17: Now describe the results of colour production mechanisms by stripe locations & tell us if it matches predictions based on mating behaviour differences between *T. scripta* & *P. concinna*. Mention sexual dichromatic spectral results briefly after line 17 sentence.

*We have added "Pseudemys concinna has chin, forelimb, and postorbital yellow stripes containing xanthophores and iridophores, whereas *T. scripta* has xanthophores and iridophores in chin and forelimb yellow stripes but only xanthophores in yellow-red postorbital/zygomatic regions. *T. s. elegans* is sexually dichromatic. The distribution of pigment cell types across body regions may be related to visual signalling but does not match predictions based on courtship position differences between the two species."*

Address:

Dpt. Philosophy and History of Science, Faculty of Science, Charles University, Viničná 7, Prague 2, 128 00, Czech Republic

Phone:

+420 606 381 557

e-mail:

brejcha@natur.cuni.cz

Line 18: 'Our results indicate that archelosaurs share some colour production mechanisms with amphibians and lepidosaurs (mention which mechanisms), but also employ novel mechanisms (mention which mechanisms).'

We have replaced "Our results indicate that archelosaurs share some colour production mechanisms with amphibians and lepidosaurs, but also employ novel mechanisms based on the nano-organization of the extracellular protein matrix that they share with mammals." by "Our results indicate that archelosaurs share some colour production mechanisms with amphibians and lepidosaurs (superposition of different pigment cell types and interplay of carotenoids with pterins), but also employ novel mechanisms (nano-organization of dermal collagen) shared with mammals.."

Page 2: Introduction

First sentence should start off as 'Identifying the evolutionary origins and selective pressures...as is its missing a true subject.
'morphological traits in which developing (instead of composing) subunits evolve for various roles...'

Revised as suggested.

Line 9: clarify: 'Colour-producing elements interact to enhance or reduce their respective contributions to observable colour' instead of 'reduce each other's contributions'

Revised as suggested.

Line 24: insert new paragraph, and start with sentence 'Iridophores contain reflecting platelets of guanine...'

Revised as suggested.

Line 38: 'patterns are common in all major lineages (for a comprehensive....' Delete 2nd placement of 'turtles'

Revised as suggested.

Throughout: make sure Colour is always spelled colour throughout and NOT color.

We checked carefully the spelling of "colour" throughout the manuscript and have not found any misspelling. We spell "coloration" following McFarland, D. (2014). A dictionary of animal behaviour. Oxford University Press.

Address:

Dpt. Philosophy and History of Science, Faculty of Science, Charles University, Viničná 7, Prague 2, 128 00, Czech Republic

Phone:

+420 606 381 557

e-mail:

brejcha@natur.cuni.cz

Page 3

Line 17: '...it IS possible to examine whether there is a relationship...'

Revised as suggested.

Lines 25-29: 'In T. scripta two of three subspecies....All Pseudemys turtles...'. These two sentences here don't belong in the intro but in Materials and Methods.

Revised as suggested. We have moved these two sentences at the beginning of section 2.2 Spectral reflectance measurements.

Rewrite line 28 'We particularly focus on...'. Delete that beginning and instead say something like To determine if colour production mechanisms match our predictions based on mating behavior we describe the cellular and ultrastructural composition of those body regions that likely...'

We have replaced "...We particularly focus on a comprehensive description of..." by "To determine if colour production mechanisms differ depending on the species-specific male position during courtship, we describe..."

Line 36: 'We discuss our results...' move this sentence somewhere in Materials and Methods...

We have removed the sentence.

Line 38: 'Our results represent the first...' delete sentence...not important to be first here. It might help sell your paper so you should say something like this in your introductory letter though...

We have removed the sentence.

End last paragraph of introduction by stating your predictions clearly: FLBS and PM will be brightest in T. scripta, and PM will be brightest in PC (or something to this effect).

We have added following: "We predict that regions of the skin that males expose during courtship to females should be brighter in colour and richer in pigment contents than regions not exposed during courtship. Specifically, in P. concinna we expect the yellow chin to be sexually dichromatic, bright, and rich in pigment contents compared to other skin surfaces. In T. scripta we expect all regions exposed during courtship, i. e. dorsal and ventral head and limbs, to be sexually dichromatic, bright, and rich in pigment contents. "

Materials and Methods

2.2 Spectral reflectance measurements

Line 63: 'Measurements were taken in a darkened room at 5 mm...' I don't understand. What was taken at a distance of 5 mm?

Address:

Dpt. Philosophy and History of Science, Faculty of Science, Charles University, Viničná 7, Prague 2, 128 00, Czech Republic

Phone:

+420 606 381 557

e-mail:

brejcha@natur.cuni.cz

We have clarified by “Reflectance spectra were taken in a darkened room ...”

Page 4

2.3. Processing and statistical analysis of reflectance spectra

Line 23-27: ‘Redundancy analysis (RDA) was performed on...’ Explain why it is necessary to do this first.

We have replaced “Redundancy analysis (RDA) was performed on summary variables scaled to zero mean and unit variance using the package vegan in R” by “To analyse the contribution of summary variables to differences among species as well as sexes, we performed redundancy analyses (RDA) using the package vegan in R [54]. To overcome difficulties with different scale and variance of summary variables we standardized the data by scaling to zero mean and unit variance [55].”

Page 5

2.6 Fourier analysis of spatial distribution of dermal collagen arrays

Line 48: End of 2.6 section: explain what an example of output looks like and what it tells us.

We have revised section 2.6. Please see the manuscript.

2.7. Pigment content analysis

Line 54: ‘, cleaned mechanically, washed...’ How was it cleaned mechanically? Explain...with a brush, etc?

Revised. We have clarified the procedure by “..., cleaned mechanically with scalpel and tweezers, ...”

Page 6

Line 15-16: ‘A chromatogram of the all carotenoid standards mixture...’ awkward phrasing, revise.

Revised. We have replaced the sentence by “A chromatogram of the mixture of all carotenoid standards (1000 ng of each standard dissolved in 1 ml EtOAc) is shown as supplementary Fig. S1a. Chromatograms of the individual carotenoid standards with their absorbance spectra are shown in supplementary Fig. S1 c – f.”

Line 31-32: should be ‘...HPLC system (Agilent series 1290 coupled with...tandem mass spectrometry, Agilent Technologies, Waldbronn, Germany).

Revised as suggested.

Line 46: ‘...Fig S2a; SRM chromatograms of the individuals...’

Address:

Dpt. Philosophy and History of Science, Faculty of Science, Charles University, Viničná 7, Prague 2, 128 00, Czech Republic

Phone:

+420 606 381 557

e-mail:

brejcha@natur.cuni.cz

Revised as suggested.

Results

Page 7

Line 6-9: 'Summary variables derived from reflectance spectra...' So? Remind us what this means... Also before next paragraph mentioning ordination plots, describe numerically the sexually dichromatic differences in spectra here.

We have added "Therefore the colour of the different regions varies more between species than between subspecies of the same species. For most of the regions T. s. elegans and T. s. scripta can be treated as a single taxon T. scripta. "

Moreover, now we describe sexual dichromatism in the next paragraphs of section 3.1

Page 8.

3.3 Reflecting paltelets of iridophores

Line 42-42: Unfortunately, measurements of iridophores from the FLBS...' Move this sentence to M&M?

Revised as suggested.

Page 9.

Line 8: electromicrograph is misspelled.

Revised.

Line 9: end sentence like this: '...with respect to wavelengths of light. In other words, the distances between collagen fibres...'

Revised as suggested.

Lines 37-42: 'Predicted normalized reflectivity of collagen fibre arrays of P.c. and T.s....' OK, but what is all of this supposed to tell us? Explain.

We have clarified the meaning of the paragraph at the end by "These predictions show how the reflecting shield of collagen fibres could contribute to the colour of the skin of turtles through its interaction with incident light. As most of the predicted reflectivity curves increase beyond 600 nm we conclude that the reflective shield of collagen fibres serves primarily to protect against overheating by reflecting infrared wavelengths. "

Line 61-62: '..is to reflect infrared radiation and only in some instances collagen fibres have been co-

Address:

Dpt. Philosophy and History of Science, Faculty of Science, Charles University, Viničná 7, Prague 2, 128 00, Czech Republic

Phone:

+420 606 381 557

e-mail:

brejcha@natur.cuni.cz

opted to function...'

Revised as suggested.

Page 10

Line 8: Specify colors of the regions described in this sentence to make the point that yellow regions lack ketocarotenoids: 'Other regions (yellow CBC, yellow FLBS of *P. concinna*, red FLBS of *T. scripta*, yellow PM of *P. concinna*, etc) ...'

Revised as suggested.

Line 13: Indent, new paragraph here: 'Examples of SRM chromatograms of pteridine...'

Revised as suggested.

Line 17: '...were found together only in the red PM of *T.s. elegans*'.

Revised as suggested.

Line 21: Indent for new paragraph: 'The red PM of *T.s.elegans* is unique among the examined...'

Revised as suggested.

Discussion

I am not sure about the way you start the first sentence. The claim itself is contentious because many turtles ARE known for their color.

We have changed the first sentence "Although they are not known for their vivid, conspicuous body coloration, many turtles are in fact as colourful as other vertebrates." to "Although animal coloration research has traditionally focused on other groups, many turtles have bright, conspicuous colours [75]."

Line 40: ', studying turtle coloration and colour producing mechanisms is crucial'....

Revised.

Line 43-45: long sentence, but still statement of contrasting courtship behavior should occur earlier in the sentence.

'...and the chemical nature of pigments that produce skin colour in two freshwater turtles with contrasting courtship behavior, *Pseudemys concinna* and *Trachemys scripta*.

Address:

Dpt. Philosophy and History of Science, Faculty of Science, Charles University, Viničná 7, Prague 2, 128 00, Czech Republic

Phone:

+420 606 381 557

e-mail:

brejcha@natur.cuni.cz

*We have replaced “In this study we analysed the pigment cell organization of the integument, the ultrastructure of colour-producing elements, and the chemical nature of pigments that produce skin colour in two freshwater turtles, *Pseudemys concinna* and *Trachemys scripta*, with contrasting courtship behaviour.” by “*Pseudemys concinna* and *Trachemys scripta*, are freshwater turtles with contrasting courtship behaviour [44–46]. In this study we analyse the pigment cell organization of the integument, the ultrastructure of colour-producing elements, and the chemical nature of pigments that produce skin colour in these two species.”*

Line 51: ‘Turtles employ colour producing mechanisms found in amphibians...’

Revised as suggested.

Line 53: ‘...and crocodiles [81]...’ seems to contradict what was said in previous sentence (extant archosaurs (crocodiles and birds)...

We have changed to first sentence of the paragraph which now reads as “Turtle skin coloration is produced by the combined action of different types of pigment cells in the dermis similarly to amphibians and lepidosaurs. In addition to xanthophores and melanophores, that were previously described in turtles [27] and also in crocodiles [86], we found abundant iridophores containing rectangular reflecting platelets in yellow skin of both species. “

Omit entire sentence placed in line 55-58: Carotenoids are involved in...feathers of birds, etc. Next sentence should be ‘In addition, our pigment analyses suggest that both pteridine derivatives and carotenoids...’

Revised as suggested.

Line 62: ‘...penetrating into the body of turtles, but in some cases collagen fibre arrays may be capable of reflecting...’

Revised as suggested.

Page 11

4.1. The coloration of Deirocheline turtles

Line 12: ‘However, we did not find support for the spectral shapes reported by Wang et al [87]; in fact...’

Is this sentence important enough to include / present in this paper? We should expect there to be geographical / environmental variation in spectra of these stripes...the mechanistic causes to these spectra are what’s really important.

Address:

Dpt. Philosophy and History of Science, Faculty of Science, Charles University, Viničná 7, Prague 2, 128 00, Czech Republic

Phone:

+420 606 381 557

e-mail:

brejcha@natur.cuni.cz

We would like to keep the sentence if possible. The reported peaks at 372 nm are artifacts and we think this should be stated here. See Figure 1 in Wang et al. 2013. In the results the peak at 372 nm is reported to be female specific “在 300~400 nm 的紫外线区段，雌性个体的各测量部位均在~372 nm 具一高峰。/In the UV light (300 – 400 nm) part of spectra, females have peak at ~372 nm in every measured site (see arrow).” The study is relevant to the topic of this paper which has been often cited.

Line 39, after info from the 2nd paragraph ('...examined turtles (Fig 7 b,d).): write about intraspecific differences in spectra, e.g. *T. scripta scripta* vs *T. scripta elegans*. Then write about sexual dichromatism in spectra?

We have added new paragraph on sexual dichromatism

*“We found intersexual differences in the reflectance spectra of CBC and FLBS of *T. s. elegans* but not *P. concinna* (Table 1). These results contrast with a previous study reporting no intersexual differences in coloration in *T. s. elegans* [84]. The small sample sizes may be responsible for our inability to statistically support the effect of sex on colour in *P. concinna*. Sexual dichromatism is apparently present in some deirocheline turtles. For example, males of the northern map turtle (*Graptemys geographica*) have brighter postorbital patches than females [36]. However, in other species, such as *Chrysemys picta* [37] and *Malaclemmys terrapin* [35], there is little or no sexual dichromatism. Sexual dichromatism has been hypothesized to result from females becoming duller as they grow and age [37]. Growth is known to affect the reflectance spectra of colourful regions in some turtles [100]. Moreover, males of some species of deirocheline turtles undergo ontogenetic colour change as they get larger and older [49,101]. As we have excluded melanistic males from the spectral measurements, the sexual dichromatism of *T. s. elegans* found here is not due to ontogenetic colour change of males. Nevertheless, it will be interesting in the future to examine the postnatal ontogenesis of colour-producing mechanisms in deirocheline turtles to see how sexual dichromatism arises as the turtles grow and age. “*

4th paragraph, Lines 55-62: better explain how reflecting platelet organization causes tissue to act as either a narrow band or broad band reflector.

We have revised the paragraph which now reads as follows

*“Continuous layers of iridophores with organized reflecting platelets are found in *T. scripta* but not in *P. concinna*. The colour produced by the scattering of incident light on arrays of particles depends on the organization, size, and the refractive index of particles [108]. Incoherent scattering occurs when the distances between individual particles are random (e. g. reflecting platelets are randomly arranged in iridophores) causing the phases of wavelets of light to nearly cancel out. In contrast, coherent scattering occurs when the dis-*

Address:

Dpt. Philosophy and History of Science, Faculty of Science, Charles University, Viničná 7, Prague 2, 128 00, Czech Republic

Phone:

+420 606 381 557

e-mail:

brejcha@natur.cuni.cz

tances between individual particles are similar (e. g. when reflecting platelets are organized in iridophores) causing the phases of wavelets of light either to completely cancel out or add up to produce relatively high reflectance. As the reflecting platelets of iridophores from the CBC of T. scripta are highly they produce colour by coherent scattering. Highly organized reflecting platelets in orange and yellow skin regions of the lizards Uta stansburiana [60], Sceloporus undulatus, and S. magister [59] reflect orange and yellow wavelengths due to thin-film interference, which is a special case of coherent scattering. The thin-film interference model predicts the highest reflectivity of the CBC of T. scripta to be between 618 and 637 nm (Fig. 12 a, b). Other regions of turtle skin containing iridophores, i. e. yellow FLBS of T. scripta and all yellow regions of P. concinna, do not have organized reflecting platelets. Iridophores with randomly organized platelets in red and white skin of Phelsuma grandis produce broadband reflection which enhance the overall brightness of the corresponding skin patch [16]. The disorganized prismatic guanin crystals of some spiders produce matte-white colour by incoherent (diffuse) light scattering [109]. However, prismatic crystals of spiders are around one micrometre in size [110], while disorganized reflecting platelets of fish [109], lizards [16] and turtles (Table 3) are much smaller. The yellow regions containing iridophores in T. scripta have increased brightness relative to yellow regions without iridophores (Figure 2). Thus even if they do not produce colour through thin-film interference, the iridophores most likely increase the brightness of colour patches and their overall conspicuousness to observers, thus contributing to colour [4,111]. Whether or not the scattering on disorganized reflecting platelets in turtles is coherent or incoherent remains to be answered.”

Page 12

1st paragraph, 1st sentence: rewrite 1st sentence (Unlike all yellow regions of both species,) as a direct-active sentence to make it more powerful.

We have replaced “Unlike all yellow regions of both species, which contain xanthophylls lutein and zeaxanthin, the red PM of T. s. elegans contains the ketocarotenoids astaxanthin and canthaxanthin (Table 4, Fig. S1).” by “All yellow regions of both species contain hydroxylated xanthophylls, lutein and zeaxanthin, whilst the red PM of T. s. elegans contains the ketocarotenoids astaxanthin and canthaxanthin (Table 4, Fig. S1).”

Line 9: ‘Previous reports indicated that...you cite only 1 previous report so this statement should be singular

Revised.

4th paragraph

Line 44-Line 61, etc. this entire paragraph needs better logical flow.

Address:

Dpt. Philosophy and History of Science, Faculty of Science, Charles University, Viničná 7, Prague 2, 128 00, Czech Republic

Phone:

+420 606 381 557

e-mail:

brejcha@natur.cuni.cz

We have revised the section 4.2

Line 46: 'The presence of red? Yellow? Xanthophores together with iridophores...'

Revised as "The presence of red-yellow xanthophores together..." Because the xanthophores in T.s.e are red whereas in T.s.s. are yellow.

Line 52-56: Sentence: This should make P. concinna rather conspicuous when viewed... This long sentence also needs more clarity, better logical flow. Also the way this sentence is written seems to contradict what you expect for P.c., no? Because P.c. males observe mates from above, should there be xanthophores and iridophores be distributed dorsally (in the PM) and ventrally (in the CBC)?

It is true. We have revised the section 4.2. Please see the manuscript for the changes we have made.

Page 13

Can you better explain iridophores as coherent or incoherent scattering mechanisms in this 4.1. section?

Please see response to the comment above "4th paragraph, Lines 55-62: better explain how..."

Section 4.2

Line 39 the collagen fibres form subepidermal collagenous lamella overlying the dermal chromatophores...Do you mean to say this? Wouldn't this obscure spectral reflectance?

Yes, we really mean to say this. Specific organization of collagen fibres may result in transparent clear tissue as in cornea (Benedek 1971 Theory of transparency of the eye. Applied optics, 10(3), 459-473). Thus, in fish and tadpoles, the subepidermal collagenous lamella must maintain specific order of collagen fibres in order to not obscure spectral reflectance. Indeed, there are examples of completely transparent fish and tadpoles. We have clarified by "In many fish and amphibian larvae, the collagen fibres form transparent subepidermal collagenous lamella overlying the dermal chromatophores [20,127,128] ."

2nd paragraph, line 55: '...,but color production by coherent scattering on nanostructured collagen fiber arrays has been previously reported in mammals (17). The 1st & second parts of this sentence don't seem logically related. 1st part: stating achromatic nature of the fiber, 2nd part refers to color production by coherent scattering in mammals.

Address:

Dpt. Philosophy and History of Science, Faculty of Science, Charles University, Viničná 7, Prague 2, 128 00, Czech Republic

Phone:

+420 606 381 557

e-mail:

brejcha@natur.cuni.cz

We have removed the first part of the sentence which now reads as follows “Colour production by coherent scattering on nanostructured collagen fibre arrays has been previously reported in mammals [17] and birds [15,18].”

Page 14

Line 1: ‘...do not conform to the conditions of coherent scattering or predictions do not...’ These need to be better explained then.

We explain the coherent/incoherent scattering in paragraph on reflecting platelets of iridophores (see above). We have revised the sentence which now reads as follows “However, in most other regions the collagen fibres are arranged randomly and thus do not produce bright colours by coherent scattering.”

Section 4.3

Line 12-14: ‘Most vertebrates do produce various colors by...’ I don’t understand what you’re trying to say here.

Much of the rest of this section seems too speculative, and perhaps needs tightening if kept in future drafts.

We have revised section 4.3

4.4. Future directions. Is this heading part of RSOS format? If not, is it necessary and appropriate? Perhaps write a conclusion section instead?

Section was removed.

Tables

Table 1. Couldn’t table 1 be arranged / expressed to include sex differences and highlight significant differences with an asterisk? Of course the p-values would need to be Bonferoni correction to account for multiple comparisons.

Revised.

Table 3. Spelling of word ‘length’ needs correction

Revised.

Figures

Figure 2: seems to show sexual dichromatism in UV and Longwavelengths. Why not present this statistically in results and add to abstract, intro and discussion?

Revised, please see the manuscript

Address:

Dpt. Philosophy and History of Science, Faculty of Science, Charles University, Viničná 7, Prague 2, 128 00, Czech Republic

Phone:

+420 606 381 557

e-mail:

brejcha@natur.cuni.cz

Reviewer: 2

The authors undertook a thorough examination of the pigments and structures that underlie the integumentary pigmentation of two species of freshwater turtles, and discuss their findings in light of the species' different courtship behaviors and their phylogenetic relationships to other extant groups of vertebrates. The treatment is exhaustive, except perhaps for the lack of pigment quantifications, and for the most part its findings are well supported.

We are very thankful for all the comments and for the extraordinary effort Reviewer 2 has made.

*It is true that we do not report any quantitative results for pigment analyses. The main reason is the small sample size of individuals analyzed to provide representative quantitative results. We find that the main differences observed between *Trachemys scripta* and *Pseudemys concinna* are qualitative rather than quantitative. This is in contrast with our previous experience with analyzing lizard coloration where quantification was essential and necessary to interpret the results (Andrade et al. 2019 Regulatory changes in pterin and carotenoid genes underlie balanced color polymorphisms in the wall lizard. PNAS, 116(12), 5633-5642.)*

I struggled a bit with the first part of the introduction and its very broad brush. The authors could probably have referenced the "interacting processes" of "functional integration of colour ornaments and morphological modularity of colour-producing elements" without actually naming them.

We have revised the first paragraph which now reads as follows

"Identifying the evolutionary origins and the selective processes responsible for the maintenance of complex morphological traits remains a persistent challenge in biology. Animal coloration is a complex morphological trait resulting from the interaction of multiple elements that evolve for various roles and only later are co-opted to produce colour [1,2]. Variation in body coloration during evolution is the result of two processes [3]: integration of colour producing elements to produce the observable colour [4], and independent development and evolution of colour producing elements [5]. The intricate relationships between colour-producing elements and their functional role stress the need to study the proximate causes of colour production [6]. While there has been progress in our understanding of the functions of animal coloration [7,8], our knowledge of colour production mechanisms in most groups of animals is still scarce. "

To present them as "intriguing" relationships is a bit deceitful.

Address:

Dpt. Philosophy and History of Science, Faculty of Science, Charles University, Viničná 7, Prague 2, 128 00, Czech Republic

Phone:

+420 606 381 557

e-mail:

brejcha@natur.cuni.cz

We have replaced “intriguing” by “intricate”.

Nothing is added by the “Future directions” section of the manuscript and it should probably be removed.

The section was removed.

Throughout the manuscript, please refer to carotenoids and pteridines (or pterins) as such, not as “derivatives”.

Initially we thought the term “pteridine derivative” would be the best to avoid confusion with Pterin as specific compound. To refer to the compounds we have analyzed as “pteridines” may lead to confusion as flavins also contain a pteridine ring. We now refer to “pteridine derivatives” with keto and amino groups as “pterins”.

“Colour-producing” should be hyphenated throughout.

Revised.

I have many other specific comments below.

Specific comments:

Page 2, line 21: five “types”, rather than “categories”.

Revised as suggested.

Page 2, line 25: “mainly guanine” (hypoxanthine is also often present).

Revised as suggested.

Page 2, line 57: “to this day” (not “date”).

Revised as suggested.

Page 3, lines 3-5: last part of the sentence is not needed: “however sexual dichromatism is limited to particular regions of the body.”

Revised as suggested.

Page 3, line 29: “ultrastructural make-up”, not “composition”.

Address:

Dpt. Philosophy and History of Science, Faculty of Science, Charles University, Viničná 7, Prague 2, 128 00, Czech Republic

Phone:

+420 606 381 557

e-mail:

brejcha@natur.cuni.cz

Revised as suggested.

Page 3, line 31: “chemical nature of pigments”, not “composition”.

Revised as suggested.

Page 3, line 34: “the make-up (or structure) of the skin”, not “composition”.

Revised as suggested.

Page 3, last sentence: did you use a probe holder to hold the probe? If not, then how did you achieve a fixed distance of 5 mm? If you used a probe holder, was 5 mm the distance of the probe to the skin, or to the probe holder?

We achieved constant distance between the probe and turtle skin by means of an entomological pin attached head down to the probe. We have clarified this now in the manuscript: “Reflectance spectra were taken in a darkened room using a hand-held probe oriented perpendicular to the skin surface (i.e. coincident normal measuring geometry [48]). An entomological pin (with acrylic head down) attached to the side of the probe allowed us to maintain a fixed 5 mm distance between the probe and the skin surface “

Page 4, line 59: remove “similarly to”.

Revised as suggested.

Page 6, line 1: the list of carotenoid standards you acquired is not exhaustive, though probably includes the most important ones.

It is true that the list of standards represents only a narrow subset of carotenoids. However, 1) the standards characterize the most abundant carotenoids present in the samples; 2) the results of analyses show additional peaks on chromatogram presented as Figure S1 in supplement. Detailed analytical study would be required to disentangle every specific carotenoid and carotenoid derivative in the samples. We would be very happy if we could acquire as many standards as possible and analyse the turtle pigments in more detail in further study.

Page 6, lines 26-27: reptiles (mainly lizards) are far more likely to use drosopterins to produce red colours using pteridines than erythropterin. I am surprised that this was missed by the authors. There are several examples of lizards with drosopterins in their integument. To mention a few recent ones: Weiss, S.L., K. Foerster and J. Hudon. 2012. Pteridine, not carotenoid, pigments underlie the female-specific orange ornament of striped plateau lizard (*Sceloporus virgatus*). *Comp. Biochem.*

Address:

Dpt. Philosophy and History of Science, Faculty of Science, Charles University, Viničná 7, Prague 2, 128 00, Czech Republic

Phone:

+420 606 381 557

e-mail:

brejcha@natur.cuni.cz

Physiol. B 161: 117-123. Cuervo, J.J., J. Belliure, and J.J. Negro. 2016. Coloration reflects skin pterin concentration in a red-tailed lizard. Comp. Biochem. Physiol B 193: 17-24.

Thank you very much for this comment. Also thank you for providing references.

We have re-analysed the samples for drosopterin and riboflavin. Please see revised manuscript for details.

We have revised section 4.1 in the discussion

Page 6, lines 42-43: check spelling: xanthopterin, leucopterin.

Revised as suggested.

Page 6, last line: the peaks are not the only features of interest. The valleys point to absorbing compounds (pigments).

Thank you for this point. It is true that valleys are important. However, we feel that the Figure 2 itself is representative enough to inform reader about the shape of reflectance spectra. We think that additional description would be redundant here.

Page 7, line 51: concinna

Revised.

Page 9, lines 33-34: “ ... are somewhat elevated compared to the radial means ...”, rather than “... are in comparison with ... relatively elevated”.

Revised as suggested.

Page 10, lines 15-18: most of these pteridines are colourless, so are not likely to explain the colour unless they contributed structurally.

It is true that many pterins reported here are colourless, but that does not imply that these compounds do not contribute to the colour. We discuss this question in revised section 4.1 now. Please see detailed answer below.

Page 10, line 40: “... studying turtle coloration and colour-producing mechanisms is crucial ...”

Revised.

Page 10, line 56, lines 58-59: carotenoids are also present in the irises of species from many orders

Address:

Dpt. Philosophy and History of Science, Faculty of Science, Charles University, Viničná 7, Prague 2, 128 00, Czech Republic

Phone:

+420 606 381 557

e-mail:

brejcha@natur.cuni.cz

of birds (Anseriformes, Galliformes, Podicipediformes, Ciconiiformes, etc...). Ref: Oehme, V.H. (1969) Vergleichende Untersuchungen über die Färbung der Vogeliris. Biol. Zentralblatt. 88: 3-35. Both carotenoids and pteridines are present in the iris of some species. Ref: Oliphant, L.W. (1981) Crystalline pteridines in the stromal pigment cells of the iris of the Great Horned Owl. Cell Tissue Res., 217:387-395; Oliphant, L.W. (1987) Pteridines and purines as major pigments of the avian iris. Pigment Cell Res., 1:129-131; Oliphant, L.W. (1988) Cytology and pigments of non-melanophore chromatophores in the avian iris. In: Advances in Pigment Cell Research. J.T. Bagnara, ed. Alan R. Liss, Inc., New York, pp. 65-82.

Thank you for this comment and all the references you provide.

The sentence "Carotenoids are involved in colour production in the feathers...." was removed as suggested by Reviewer 1. We have also revised following sentence "However, our pigment analyses suggest that both pteridine derivatives and carotenoids are involved in the production of the yellow-red skin colours of turtles, which had not been clearly documented in any archelosaur to this date." Which now reads as „In addition, our pigment analyses suggest that both pterins and carotenoids are involved in the production of the yellow-red skin colours of turtles, which had not been clearly documented in integument of any archelosaur to this date“.

We cite the suggested references in revised section 4.3

Page 12, line 1: though they may not produce colour through thin-film interference, iridophores most certainly "will" (not "may") increase the brightness of colour regions and overall conspicuousness to observers, thus contribute to colour. The change should be made to the Abstract as well.

Revised as suggested.

Page 12, lines 8-11: you sound like you agree with these results. You might want to change to "In the yellow chin of *C. picta* only apocarotenoids were postulated ..." or something of that sort.

*Revised as suggested. Now the sentence reads as follows "In the yellow chin of *C. picta* only apocarotenoids were described, whereas the orange neck and leg contain apocarotenoids and ketocarotenoids. A previous report indicated that apocarotenoids are abundant in tissue from yellow chin and yellow neck stripes of *T. scripta*, its "orange" postorbital region containing only ketocarotenoids [30]. "*

Page 12, lines 26-27: your finding is not exactly the opposite of the previous sentence. Please state your finding instead.

Address:

Dpt. Philosophy and History of Science, Faculty of Science, Charles University, Viničná 7, Prague 2, 128 00, Czech Republic

Phone:

+420 606 381 557

e-mail:

brejcha@natur.cuni.cz

We have replaced “Our results show the opposite is true (Table 4).” By “Our results show that both yellow and red regions of turtle skin contain multiple types of pterins (Table 4).”

Page 13, line 5: again many of these pteridines are colourless, so probably not all that relevant. Instead emphasize differences in the abundance of the colourful ones.

We have revised and amended the paragraph describing differences in pterin content in section 4.1. Please see the revised manuscript.

Page 13, line 50: there are no iridophores in the plumages of birds.

The sentence was not clear. We have clarified by following: “ A thermoprotective function of colour-producing structures has been suggested for iridophores found in frogs [141] or chameleons [58]. A thermoprotective function has also been suggested for the keratinous feathers of birds that reflect infrared wavelengths due to their microstructure[142,143]. “

Page 14, line 12: “Most vertebrates produce ...” or “Most of the vertebrates produce ..”

We have revised section 4.3

Page 14, line 13: true of integumentary coloration, not of bird feathers. Bird feathers can harbour many different types of pigments.

We meant that there are no pigment cell types other than melanocytes in birds. The other pigments in feathers are not of pigment cell origin, but rather result from physiological actions of keratinocytes (Vanhoutteghemet al. (2004) Serial cultivation of chicken keratinocytes, a composite cell type that accumulates lipids and synthesizes a novel β -keratin. Differentiation, 72(4), 123-137; McGraw (2004) Colorful songbirds metabolize carotenoids at the integument. Journal of Avian Biology, 35(6), 471-476). We have revised section 4.3

Page 14, line 25: “poses”?! Not the correct word.

“poses” replaced by “contain”.

Page 14, line 33: thus presumably the ancestral state (plesiomorphic).

We have added new sentence: “...has been reported in fish [113], amphibians [148], and lizards [105,149]. Thus colour-production by interplay between carotenoids and pterins presumably represents the ancestral state in vertebrates. “

Address:

Dpt. Philosophy and History of Science, Faculty of Science, Charles University, Viničná 7, Prague 2, 128 00, Czech Republic

Phone:

+420 606 381 557

e-mail:

brejcha@natur.cuni.cz

Figure 6. It is pretty hard to discern some of the highlighted elements, like the iridophores in 6d. I wish the smaller images were made slightly larger.

The 6d is to show context of all elements in the dark stripe. Iridophores there are very small, and the highlighting is supposed to illustrate their presence, even at this magnification are hard to discern. We have included electromicrograph of higher magnification of iridophore next to the dermal melanosome in the dark stripe 6g. The figure 6g has arrows instead of white line to not obscure area of basal lamina and arrowhead instead of turquoise line to not obscure details of iridophore.

Figure 9. I am not 100% sure that the vesicles in 9c are oil droplets. They do appear to show some internal structure. Can you see a membrane? Oil droplet should be membrane-free. Pterinosomes don't always show lamellae.

There are no oil droplets in Figure 9. All of the structures in Figure 9. have a visible membrane. Vesicles in Figure 9 c have prominent concentric lamellae that are characteristic of pterinosomes. Vesicles in Figure 9 a, b are similar to the carotenoid vesicles in other studies. It is true however, that vesicles of similar ultrastructure as 9 b have been reported previously reported as immature vesicles (possibly pterinosomes). We do point to this fact in section 4.1. The resemblance of the vesicles in Figure 9 a, b in electron density to carotenoid containing oil droplets as well as the results of the pigment analyses in these same regions suggests that these structures are carotenoid vesicles with different pigment contents.

Figure 15. I don't see any grey rectangle on the graphs.

There is no grey rectangle in Figure 15. We apologize for the mistake. The sentence has been removed.

Page 41, line 24: A/y0 "represents" or "captures", not "determine"

We have replaced "determine" by "represents".

Page 43, line 17: italicize T. scripta.

Revised.

From page 43, line 56 to page 44, line 11: "pose" is not the right word. "present", "display" or "feature" might work

Revised.

Address:

Dpt. Philosophy and History of Science, Faculty of Science, Charles University, Viničná 7, Prague 2, 128 00, Czech Republic

Phone:

+420 606 381 557

e-mail:

brejcha@natur.cuni.cz

CHARLES UNIVERSITY
Faculty of science

Appendix B

Prague, 14th June 2019

To: Editorial board of *Royal Society Open Science*

Dear Prof. Padian, Dr. Sefc, and Reviewers

Please find attached the revised manuscript entitled “Body coloration and mechanisms of colour production in Archelosauria: The case of deirocheline turtles”. We have closely followed all the suggestions. Our response to each point is presented below in italics.

We are thankful for all the suggestions you and reviewers made during the process. We hope you will find current version of the manuscript suitable for publication in Royal Society Open Science.

On behalf of authors,
Yours faithfully,

Jindřich Brejcha

Address:

Dpt. Philosophy and History of Science, Faculty of Science, Charles University, Viničná 7, Prague 2, 128 00, Czech Republic

Phone:

+420 606 381 557

e-mail:

brejcha@natur.cuni.cz

Responses to review

(Author's responses are written in italics)

Associate Editor's comments (Dr Kristina Sefc):

Dear authors,

Both reviewers agree with the changes that were made to the manuscript and offer specific suggestions to improve the wording and clarity. Reviewer 2 also requests a revision in regard to the interpretation of colorless pteridines and a correction of the Information on avian pigment cells. Please follow their suggestions closely. Additionally, please complete the figure caption for Figure S1 (G – J). Do you have any citable source for drosopterin extraction with DMSO?

Sincerely, Kristina Sefc

Drosopterin was not extracted with DMSO. Drosopterin was extracted from frozen fruit flies with acidic methanol following the procedure described in Ferré et al. [70]. The extract was later diluted with DMSO and used as a working solution. As far as turtle skin extracts, these were extracted with DMSO as we did in the analyses of skin of lizard [72], where we have detected drosopterin.

Associate Editor: 2

Comments to the Author:

(There are no comments.)

Reviewer comments to Author:

Reviewer: 1

I have read the Royal Society Open Science manuscript (RSOS-190319 rev 1) 'Body coloration and mechanisms of colour production in Archelosauria: The case of deirocheline turtles'. The authors attended to the previous comments of reviewers and the findings and implications of this interesting work are more understandable. I believe it is almost ready for publication, but I have a few **specific suggestions**:

Make sure various words are consistently spelled with UK-style spelling rules throughout paper. E.g. Colour (not color), behavior (not behavior), etc. The title still has the 'US' spelling for color.

We checked carefully the spelling of "colour" throughout the manuscript and have not found any misspelling. We spell "coloration" following McFarland, D. (2014). A dictionary of animal behaviour. Oxford University Press.

Abstract

Line 6: should write as 'colour production still remain unknown.'

Revised.

Address:

Dpt. Philosophy and History of Science, Faculty of Science, Charles University, Viničná 7, Prague 2, 128 00, Czech Republic

Phone:

+420 606 381 557

e-mail:

brejcha@natur.cuni.cz

Line 9: rewrite these few sentences to describe the perspective more logically: We found xanthophores, melanophres, abundant iridophores and dermal collagen fibers is stripes of both species. Moreover, in both species, the xanthophores coloring the yellow-red skin contain carotenoids, pterins, and riboflavin, but the two species differ in the distribution of pigment cell types and pigment diversity. Both *Pseudemys concinna*'s and *T. scripta*'s yellow chin and forelimb stripes contain xanthophores and iridophores, but *P. concinna* and *T. scripta*'s post-orbital region's differ in cell-type distribution. *T. scripta*'s post-orbital stripes contain xanthophores and iridiophores, while *T. scripta*'s yellow-red post-orbital / zygomatic regions contain xanthophores only.

Revised as "Here we compare reflectance spectra, cellular, ultra- and nano- structure of colour-producing elements, and pigment types in the freshwater turtles with contrasting courtship behaviour, Trachemys scripta and Pseudemys concinna. The two species differ in the distribution of pigment cell-types and in pigment diversity. We found xanthophores, melanocytes, abundant iridophores and dermal collagen fibres in stripes of both species. The yellow chin and forelimb stripes of both P. concinna and T. scripta contain xanthophores and iridophores, but the post-orbital regions of the two species differ in cell-type distribution. The yellow post-orbital region of P. concinna contains both xanthophores and iridophores, while T. scripta has only xanthophores in the yellow-red postorbital/zygomatic regions. Moreover, in both species, the xanthophores colouring the yellow-red skin contain carotenoids, pterins, and riboflavin, but T.scripta has a higher diversity of pigments than P. concinna."

Introduction

Line 38: We predict that regions of the skin that males expose to females during courtship should richer in pigment contents than regions not exposed during courtship. Omit the word 'bright' from these predictions because in tri-stimulus theory brightness means the amount of total reflectance coming off the integument and correlates inversely with increased pigment concentrations.

Revised as "We predict that regions of the skin that males expose during courtship to females should be more conspicuous in colour and richer in pigment types than regions not exposed during courtship. Specifically, in P. concinna we expect the yellow chin to be sexually dichromatic, conspicuous, and rich in pigment types compared to other skin surfaces. In T. scripta, on the other hand, we expect all regions exposed during courtship, i. e. dorsal and ventral head and limbs, to be sexually dichromatic, conspicuous, and rich in pigment types. "

Materials and Methods

2.2 Spectral reflectance measurements

Line 51: necessary to include & describe the Cumberland slider? Seems to suggest you're going to present research on it, and of course you aren't.

Revised.

Page 4, line 1: Make sure your abbreviations are formatted consistently throughout. E.g. if using

Address:

Dpt. Philosophy and History of Science, Faculty of Science, Charles University, Viničná 7, Prague 2, 128 00, Czech Republic

Phone:

+420 606 381 557

e-mail:

brejcha@natur.cuni.cz

bold-face for them make sure they are all bold face (in later sections, etc). I don't think you need to use bold-face on these abbreviations.

Revised.

Results

Page 10, Line 29: 1st sentence of 2nd full paragraph: Normalized averaged radial means of the Fourier power spectra... This is a long, run-on sentence and should be broken into 2-3 sentences. Why is the zygomatic region of *P. concinna* abbreviated as YP? Wouldn't it be more intuitive if it were abbreviated 'ZP' (for zygomatic patch)? No other abbreviation has Y in it to mention color, and in general the letters represent anatomical locations instead of color.

*1)The long sentence – Revised. First two sentences in the paragraph now read as:
“Normalized average radial means of the Fourier power spectra for each region of P. concinna and T. s. elegans are shown in Fig 14. Radial means of the Fourier power spectra of all examined regions of both species show increased power values in the lowest spatial frequencies (< 0.0034 nm⁻¹) compared to spatial frequencies relevant to coherent scattering in the visible light (Fig. 14 a-f; supplementary Fig. S5 a).”*

2)The abbreviation – Revised as suggested.

Discussion

Page 13 line 6: Is 'wavelets' the correct word?

We have replaced “wavelets” by “waves”.

Line 10: 'As the reflecting platelets of iridophores from the CBC of *T. scripta* are highly _____, they produce...' Are you missing an adjective here?

We have added “organized”.

Line 28: 'All yellow regions of both species contain hydroxylated xanthophylls such as lutein and zeaxanthin...'

Revised.

Page 15, line 2: Sentence meaning is unclear. Do you mean 'As mating strategies are better understood?' or 'become better studied'? Or should the sentence read 'Because mating strategies are well-studied in this group, it may be relatively straightforward to test hypotheses about sexual selection'?

Revised as: “Because mating strategies are well-studied in this group, it may be relatively straightforward to test hypotheses about the role of natural and sexual selection in shaping the evolution of

Address:

Dpt. Philosophy and History of Science, Faculty of Science, Charles University, Viničná 7, Prague 2, 128 00, Czech Republic

Phone:

+420 606 381 557

e-mail:

brejcha@natur.cuni.cz

colour across the Deirochelinae once more data on color and colour-producing mechanisms become available.”

4.3 Significance for vertebrate skin colour evolution

Line 65: ‘Physiological mechanisms of colour production...’

Revised.

Line 69: ‘Iridophore and xanthophore differentiation has been lost...’

Revised.

Page 16, 1st full paragraph:

Line 9: How about ‘Xanthophores are pigment cells that contain carotenoids and pterins in their vesicles.’?

Thank you. Revised.

Line 11: ‘Except a rare occurrence in feathers of penguins...’

Revised.

Tables and Figures

Figure 3. Minor detail, but should the B1- and H1-based spectral variables (i.e. brightness and hue) be written with the same format as the S1 variables? E.g., since S1.blueCBC, shouldn’t B1.CBC? H1.CBC?

The S1.blueCBC means saturation in blue segment of visible wavelengths of the CBC region. Likewise, hue of the CBC region is abbreviated H1CBC.

Figure 10: overlay abbreviations for names of colored regions above the regions on these images (e.g. CBC, DHC, PM, & YP)?

Revised.

Figure 12: x-axes (wavelength) spelled wrong.

Revised.

Fig 10 caption: why is zygomatic patch abbreviated YP?

Address:

Dpt. Philosophy and History of Science, Faculty of Science, Charles University, Viničná 7, Prague 2, 128 00, Czech Republic

Phone:

+420 606 381 557

e-mail:

brejcha@natur.cuni.cz

Revised.

Fig 12. 1st sentence of caption should read: 'Distribution of predicted reflectivity of reflecting platelets of iridophores based on Fourier Analysis. (a)...'

Revised, however we have not used Fourier Analysis to predict reflectivity of reflecting platelets but thin layer interference model. The caption now reads as: "Distribution of predicted reflectivity of reflecting platelets of iridophores based on thin layer interference model."

Reviewer: 2

I previously reviewed this paper. I commend the authors for resampling the skins for evidence of riboflavin and drosopterin, a pteridine often found in lepidosaurians with red tones. I think that it adds perspective to the paper.

I must qualify a comment I made in my past review. I pointed out that many of the pteridines the authors found in the brightly-coloured surfaces of turtles were colourless and might not contribute directly to skin colours we (humans) perceive as yellow or red. This led the authors to revise their manuscript to try to explain the presence of colourless pteridines, which produced a treatment that was not evenly convincing (see specific comment below). I will add another consideration to the mix: since many of these "colourless" pteridines absorb in the UV and turtles can see in the UV (as pointed out in the manuscript: p. 34, line 10), it is probably reasonable to assume that they would produce/alter colours that turtles can see (unlike us).

We have revised the paragraph (according to comments below, please see specific responses there) which now reads as follows:

*"Steffen et al. [30] suggested that the yellow-red skin of *C. picta* and the yellow skin (but not the red skin) of *T. scripta* contain small amounts of pterins. Our results show that both yellow and red regions of turtle skin contain multiple types of pterins (Table 4). Red and orange colours are produced in some lizards by drosopterin [119,120] which we have not found in turtles. However, yellowish isoxanthopterin and yellow xanthopterin are present in yellow-red regions of *T. scripta*. Xanthopterin has been previously reported in yellow and red skin of *Phelsuma* lizards [16]. The yellow and red skin regions of *Phelsuma* have identical pigment composition but differ in the cellular pH environment which determines the colour of the skin [16]. Thus, it is possible that xanthopterin and isoxanthopterin could contribute to both yellow and red colour production in the skin in turtles due to differences in the cellular pH among these regions. However, most of the pterins present in the skin of both species of turtles are colourless compounds. Colourless biopterin and 6-pterin-COOH have been previously described in the skin of *M. japonica* [32]. In lizards colourless pterins have been found in colourful body regions. For example, biopterin has been detected in the yellow-red xanthophores of *Phelsuma* spp. [16], and varying amounts of colourful and colourless pterins have been found in the yellow-orange-red ventral colour morphs of the European wall lizard (*Podarcis muralis*) [72]. Since many of these colourless pterins absorb in the UV range [16], which turtles can see [90,91], colourless pterins*

Address:

Dpt. Philosophy and History of Science, Faculty of Science, Charles University, Viničná 7, Prague 2, 128 00, Czech Republic

Phone:

+420 606 381 557

e-mail:

brejcha@natur.cuni.cz

*may in fact affect colour perception in turtles. Moreover, although some pterins may be colourless even to turtles, it may be difficult to separate their influence on colour production from that of colourful pterins, because the colourful pterins are derived from colourless ones and vice versa. For example, yellow sepiapterin is precursor of colourless biopterin (the precursor of yellow 7-oxobiopterin and pterin) and colourless pterin, the precursor of yellowish isoxanthopterin and its isoform yellow xanthopterin [121]. The content of all individual types of pterins thus depends on the availability of their precursor pterins. The diversity of different pterin types may thus describe the differences between colourful skin regions better than the amount of any single pterin type. In addition to carotenoids and pterins, we found riboflavin in all yellow and red regions of *T. scripta*. In *P. concinna* we found riboflavin in the yellow CBC and FLBS but not in the yellow PM. Riboflavin is present in the yellow-red skin of various lizards and snakes but has never been reported in turtle skin [32,72,122]. Therefore, the yellow and red colours of the turtles examined here likely result from a complex interplay between carotenoids, pterins, and riboflavin."*

My main concern this time around is with the way the authors characterize birds in the last section of the manuscript ("Significance for vertebrate skin colour evolution"). It is inaccurate to state that birds lack pigment cells other than melanocytes. Birds may have largely lost pigments cells from their integument as a result of the development of an outer covering of feathers, but they retained most pigment cell types (e.g., xanthophores and iridophores) in their irises (Oliphant et al. 1992 Pigment Cell Res 5:367-371). I note that turtles probably have similar pigments cells in their irises than they do in their integuments, since the color patterns on the face of turtles often continue uninterrupted across the iris. I think that making a distinction between the pigment cells of the integument and those of the iris is inappropriate, since BOTH derive equally from neural crest cells (see review on the origin of cells that form the eye in Williams and Bohnsack 2015 Birth Defects Research 105:87-95). The pigment cells in the irises of birds may no longer have dendritic projections and motile pigment organelles (and thus might be more appropriately called xanthocytes and iridocytes), but that is beside the point.

Thank you very much for emphasizing this to us.

I (JB) have read the suggested paper of Oliphant and colleagues carefully and indeed you are right and it is important to notice. I have always considered it as rather special case of few species. It is very interesting problem. Of course, birds seem to maintain some regulatory mechanisms leading to differentiation of xanthophores contrary to mammals (Kimura, T., Nagao, Y., Hashimoto, H., Yamamoto-Shiraishi, Y. I., Yamamoto, S., Yabe, T., ... & Naruse, K. (2014). Leucophores are similar to xanthophores in their specification and differentiation processes in medaka. Proceedings of the National Academy of Sciences, 111(20), 7343-7348.). Yet it seems reasonable to assume that the integumental and iridial pigment cells will differ to some extent (not only in morphology, but also in developmental origin – different parts of neural crest, and possibly in regulatory mechanisms) and it thus make sense to me to discuss the integumental pigmentation as such.

We have revised the last section. Please see the MS for details.

Address:

Dpt. Philosophy and History of Science, Faculty of Science, Charles University, Viničná 7, Prague 2, 128 00, Czech Republic

Phone:

+420 606 381 557

e-mail:

brejcha@natur.cuni.cz

Specific comments:

Page 23, line 39: I suggest “pigment types” rather than “pigment contents”, as the latter implies quantification (which did not take place).

Revised.

Page 23, line 42-44: Awkward sentence. Maybe something like: “The chin, forelimb, and postorbital yellow stripes of *Pseudemys concinna* contains xanthophores ...”

Revised.

Page 24, lines 8 and 9: “colour-producing elements” is sometimes hyphenated, but not always (as here). Please make sure that “colour-producing” is hyphenated throughout.

Revised.

Page 24, line 55: Remove extra “in”: “playing a role in in colour production”.

Revised.

Page 25, lines 30, 33: I suggest rich in “pigment types”? or “pigments”?

Revised.

Page 28, line 41: you are missing micron in: 2.6 μm

Revised.

Page 28, line 42 “tert-butyl methyl ether”

Revised.

Page 29, lines 7-10: Awkward sentence. I suggest something like “Separation of all studied compounds with the exception of drosopterin was achieved through isocratic elution at a flow rate of 0.5 ml/min with a mobile phase of ...”

Revised as suggested.

Page 32 last line: Again “pigment types” would be preferable to “pigment contents”.

Revised.

Page 33, lines 16 and 18: This is confusing. Some pigments were not detected in any samples. But

Address:

Dpt. Philosophy and History of Science, Faculty of Science, Charles University, Viničná 7, Prague 2, 128 00, Czech Republic

Phone:

+420 606 381 557

e-mail:

brejcha@natur.cuni.cz

then *T. s. scripta* contained all types except pterin. I suspect that the latter sample did not contain the pigments not detected in any sample, but this should be made clearer. Please revise the text.

Revised as: "The yellow CBC and FLBS of both lineages of T. scripta and the ZP of T. s. scripta contain 6-biopterin, pterin-6-COOH, isoxanthopterin, leucopterin, d-neopterin, and xanthopterin, but not pterin. "

Page 33, lines 22-29: This is largely a repeat of results from the previous two paragraphs, but framed differently. It is hard for me to justify repeating information when the paper is already so long.

This paragraph was removed.

Page 33, line 57: You are playing with words here. Pterins and carotenoids actually are involved in the production of yellow-red colours in birds, maybe not in their integument, but in the iris.

Now the sentence reads as: "In addition, our pigment analyses suggest that both pterins and carotenoids are involved in the production of the yellow-red skin colours, which had not been clearly documented in the integument of turtles to this date. "

Page 35, line 28: "both species contain the hydroxylated xanthophylls ...".

Revised.

Page 35, line 35: "whereas the orange neck and leg contained apocarotenoids and ketocarotenoids."

Revised.

Page 35, last paragraph: Since turtles can see in the ultraviolet what look like colourless pterins to us may actually be visible (and coloured) to turtles.

We have added sentence: "Since many of these colourless pterins absorb in the UV range [16], which turtles can see [90.91], colourless pterins may be in fact colourful to turtles."

Page 36 lines 1-2: I find this suggestion highly speculative and a bit far-fetched.

We have removed the sentence: "It is thus possible that colourless pterins influence the pH environment inside the cells and thus affect the colour of the skin".

Page 37, last line. It is untrue that birds do not develop iridophores and xanthophores. They do in the iris.

We have clarified the statement which now reads as follows: "Also, iridophores and xanthophores have evolved differently in the main archelosaurian clades [145,149–151]. Skin pigment cells of croc-

Address:

Dpt. Philosophy and History of Science, Faculty of Science, Charles University, Viničná 7, Prague 2, 128 00, Czech Republic

Phone:

+420 606 381 557

e-mail:

brejcha@natur.cuni.cz

odiles are not organized into stacked continuous layers and their putative iridophores contain amorphous material rather than reflecting platelets [86]. Cells with lipid vesicles suspected to contain carotenoids have been reported in both the epidermis and the dermis of non-feathered bare skin of some bird taxa [18,152], but this association probably evolved secondarily and does not involve pigment cells but other cell types [151]. Even though xanthophores, iridophores, and a broad diversity of pigment cells with unusual pigment vesicles are present in the irises of birds, these cell types are not present in the skin of birds [151]."

Page 38, line 13. A period is missing after "[160, 162,163]".

Revised.

Page 38, line 14: "contain pigments similar to those in xanthophores".

Revised.

Page 38, lines 17-18: carotenoids and pterins do act together to produce colour in the iris of some birds (as pointed out earlier in the paper).

Revised. The sentence now reads as follows: "However, there is no evidence of carotenoids and pterins acting together to produce colour in the skin of any archelosaur."

Page 38, line 20: Birds should be listed here as well, as there are examples of pterins contributing with carotenoids to yellow-red colours, again in the iris.

Revised.

Page 38, line 22: This is incorrect. Some birds possess xanthophores in their irises (unless you want to be technical and call those xanthocytes).

Revised. The sentence now reads as follows "It remains open question whether the loss of xanthophores in the skin of avian archelosaurs led to rare occurrence of pterins in the skin and feathers."

Figure 9. I am still not 100% sure that the vesicles in 9e (not 9c in my previous review, my mistake) are oil droplets. They do appear to show some internal structure.

It is true that vesicles in 9c seem to have internal structure compared to vesicles in 9d (there are no oil droplets as these have apparent plasmatic membrane). We have removed ", there is no pronounced structure visible inside these vesicles. In context of chemical analysis, it seems these vesicles contain predominantly yellow carotenoids" from the figure caption.

Address:

Dpt. Philosophy and History of Science, Faculty of Science, Charles University, Viničná 7, Prague 2, 128 00, Czech Republic

Phone:

+420 606 381 557

e-mail:

brejcha@natur.cuni.cz